



# The Eulerian urban dispersion model EPISODE. Part II: Extensions to the source dispersion and photochemistry for EPISODE-CityChem v1.2 and its application to the city of Hamburg

Matthias Karl[1], Sam-Erik Walker[2], Sverre Solberg[2], and Martin O. P. Ramacher[1]

[1]Chemistry Transport Modelling, Helmholtz-Zentrum Geesthacht, Geesthacht, Germany
[2]Norwegian Institute for Air Research (NILU), Kjeller, Norway.

*Correspondence to:* M. Karl (matthias.karl@hzg.de)

**Abstract.** This paper describes the CityChem extension of the Eulerian urban dispersion model EPISODE. The development of the CityChem extension was driven by the need to apply the model in lower latitude cities with higher insolation than in northern European cities. The CityChem extension offers a more advanced treatment of the photochemistry in urban areas and entails specific developments within the sub-grid components for a more accurate representation of the dispersion in the proximity of urban emission sources. The WMPP (WORM Meteorological Pre-Processor) is used in the point source sub-grid model to calculate the wind speed at plume height. The simplified street canyon model (SSCM) is used in the line source sub-grid model to calculate pollutant dispersion in street canyons. The EPISODE-CityChem model integrates the CityChem extension in EPISODE, with the capability of simulating photochemistry and dispersion of multiple reactive pollutants within urban areas. The main focus of the model is the simulation of the complex atmospheric chemistry involved in the photochemical production of ozone in urban areas. EPISODE-CityChem was evaluated with a series of tests and with a first application to the air quality situation in the city of Hamburg, Germany. A performance analysis with the FAIRMODE DELTA Tool for the air quality in Hamburg showed that the model fulfils the model performance objectives for $NO_2$ (hourly), $O_3$ (daily max. of the 8-h running mean) and $PM_{10}$ (daily mean) set forth in the Air Quality Directive, qualifying the model for use in policy applications. Observed levels of annual mean ozone at the five urban background stations in Hamburg are captured by the model within ±15 %. Envisaged applications of the EPISODE-CityChem model are urban air quality studies, emission control scenarios in relation to traffic restrictions and the source attribution of sector-specific emissions to observed levels of air pollutants at urban monitoring stations.

## 1 Introduction

Air quality (AQ) modelling plays an important role by assessing the air pollution situation in urban areas and by supporting the development of guidelines for efficient air quality planning, as highlighted in the current Air Quality Directive (AQD) of the European Commission (EC, 2008). Main air pollution issues in Europe are the human health impacts of exposure to particulate matter (PM) and ozone ($O_3$); while the effects of air pollution due to nitrogen dioxide ($NO_2$), sulphur dioxide ($SO_2$), carbon monoxide (CO), lead (Pb) and benzene have been reduced during the last two decades due to emission abatement measures.



Ozone is a secondary pollutant, generated in the troposphere involving two classes of precursor compounds, i.e. nitrogen oxides and volatile organic compounds (VOCs), in photochemical reaction cycles, initiated by the reaction of the hydroxyl (OH) radical with organic molecules. For health protection, a maximum daily 8-hour mean threshold for ozone ($120\,\mu g\,m^{-3}$) is specified as a target value in the European Union, which should not be exceeded at any AQ monitoring station on more than

25 days per year. However, about 15 % of the population living in urban areas is exposed to ozone concentrations above the European Union (EU) target value (EEA, 2015).

Eulerian chemistry-transport model (CTM) systems using numerical methods for solving photochemistry (including chemical reaction schemes with varying degree of detail) have mainly been used for regional-scale air quality studies. Recent nested model approaches using regional CTM systems have been applied to capture pollution processes from the continental scale to

the local scale, using between 1-km to 5-km resolution and a temporal resolution of 1 hour for the innermost domain (e.g. Borge et al., 2014; Karl et al., 2015; Petetin et al., 2015; Valverde et al., 2016). Regional AQ models can give a reliable representation of $O_3$ concentrations in the urban background, but due to their limitation in resolving the near-field dispersion of emission sources and photochemistry at sub-kilometre scale, i.e. in street canyons, around industrial stacks and on neighbourhood level, they cannot provide the information needed by urban policymakers for population exposure mapping, city planning and the

assessment of abatement measures.

Urban scale AQ models overcome the limitation inherent in regional scale models by taking into account details of the urban topography, wind flow field characteristics, land use information and the geometry of local pollution sources. The urban AQ model EPISODE developed at the Norwegian Institute for Air Research (NILU) is a 3-D Eulerian grid model that operates as a CTM, offline coupled with a numerical weather prediction (NWP) model. EPISODE is typically applied with

a horizontal resolution of $1 \times 1\,km^2$ over an entire city with domains of up to $150\,km^2$ in size. The Eulerian grid component of EPISODE simulates advection, vertical/horizontal diffusion, background transport across the model domain boundaries, and photochemistry. Several sub-grid scale modules are embedded in EPISODE to represent emissions (line source and point sources), Gaussian dispersion and local photochemistry. In particular, the model allows the user to retrieve concentrations at the sub-grid scale in specified locations of the urban area. Moreover, the EPISODE model is an integral part of the operational

Air Quality Information System AirQUIS 2006 (Slørdal et al., 2008).

Part one (Hamer et al., 2019) of this two-part article series provides a detailed description of the EPISODE model system including physical processes for the atmospheric pollutant transport, photochemistry involving nitrogen oxide (NO), $NO_2$ and $O_3$ based on the photo-stationary state (PSS) approximation, the sub-grid components and the interaction between the Eulerian grid and the sub-grid processing of pollutant concentrations. Part one examines the application of EPISODE to air

quality scenarios in the Nordic winter setting. During wintertime in Northern Europe, the PSS assumption is a rather good approximation of the photochemical conversion occurring close to the emission sources. However, when the solar ultraviolet (UV) radiation is stronger, in particular during summer months or at more southerly locations, net ozone formation may take place in urban areas at a certain distance from the main local emission sources (Baklanov et al., 2007). The intention to achieve a wider applicability of the model to cities in lower latitudes drove the development of the CityChem extension. In this part, the

features of the CityChem extension for treating the atmospheric chemistry in urban areas and specific developments within the





sub-grid components for a more accurate representation of the near-field dispersion in the proximity of urban emission sources are described.

Atmospheric chemistry on an urban scale is complex due to the large spatial variations of input from anthropogenic emissions. VOCs related to emissions from traffic are involved in the chemical conversion in cities at lower latitudes. Therefore

it becomes necessary to simulate a large number of chemical interactions involving nitrogen oxides ($NO_X = NO_2 + NO$), $O_3$, VOCs, $SO_2$ and secondary pollutants. In order to use comprehensive photochemical schemes in urban AQ models involving VOC interactions, the highest priority for the initial development was to reduce the number of compounds and reactions to a minimum, while maintaining the essential and most important aspects of chemical reactions taking place in the urban atmosphere on the relevant space and time scales. From this initial development work at NILU, the EMEP45 chemical mech-

anism (Walker et al., 2003) resulted from an appropriate reduction of the former EMEP chemistry scheme (Simpson, 1995). In recent years, the need to update and expand the reaction equations and coefficients of EMEP45 led to the development of EmChem03-mod, using a chemical pre-processor for the automated set-up of differential equations of the chemistry solver in Fortran 90 code. The latest development is the EmChem09-mod scheme, which is similar to the current EMEP chemistry mechanism (EmChem09; Simpson et al., 2012). EmChem09-mod enables the simulation of biogenic VOCs such as isoprene

and monoterpenes, emitted from urban vegetation.

CityChem offers a more advanced treatment for the dispersion close to point emission sources (e.g. industrial stacks) and line emissions sources (open roads and streets):

1. Modifications to the plume rise from elevated point sources allow for a more accurate computation of the plume trajectories. The WMPP (WORM Meteorological Pre-Processor) is utilized in the CityChem extension to calculate the wind
speed at the actual plume height.

2. Modifications of the line source emission model to compute receptor point concentrations in street canyons. A simplified street canyon model (SSCM) is implemented to account for pollutant transfer along streets, including a parameterization of the mass transfer within a simplified building geometry at street level.

3. Modifications of the photochemistry in the sub-grid components, replacing the PSS assumption with the EP10-Plume
scheme, a compact scheme including inorganic reactions and the photochemical degradation of formaldehyde, using a numerical solver.

Although computational fluid dynamics (CFD) models can also be used to solve for the above local-scale phenomena, they are limited to localized applications and are not appropriate for the simulation of dispersion across complex urban areas. In addition, the simulation of chemical conversions of reactive pollutants using CFD models requires a large amount of computa-

tional time (Sanchez et al., 2016).

The EPISODE-CityChem model, which is based on the core of the EPISODE model, integrates the CityChem extension into an urban CTM system. This paper gives a model description of EPISODE-CityChem version 1.2. EPISODE-CityChem has the capability of simulating the photochemical transformation of multiple reactive pollutants along with atmospheric diffusion to



produce concentration fields for the entire city on a horizontal resolution of 100 m or even finer and a vertical resolution of 24 layers up to 4000 m height. The possibility to get a complete picture of the urban area with respect to reactive pollutant concentrations, but also information enabling exposure calculations in highly populated areas close to road traffic line sources and industrial point sources with high spatial resolution, turns EPISODE-CityChem into a valuable tool for urban air qual-

ity studies, health risk assessment, sensitivity analysis of sector-specific emissions and the assessment of local and regional emission abatement policy options.

The paper is organised as follows: Sect. 2 gives an overview of EPISODE-CityChem and a detailed description of the photochemical reaction schemes and the modifications of the near-source dispersion in the sub-grid components. Sect. 3 presents tests of the various modules of the CityChem extension. Sect. 4 describes the application of EPISODE-CityChem within a

nested model chain for simulating the air quality and atmospheric chemistry in the city of Hamburg. Plans for the future development of the EPISODE model, addressing the need for more sophisticated photochemistry, treatment of aerosol formation on an urban scale, and further improvements of the source dispersion are outlined in Sect. 5.

## 2   Development and description of EPISODE-CityChem model extensions

EPISODE consists of a 3-D Eulerian grid CTM model that interacts with a sub-grid Gaussian dispersion model for the disper-

sion of pollutants emitted from both line and point sources. We refer to part one (Hamer et al., 2019) for a technical description of the model. The standard EPISODE model simulates the emission and transport of $NO_X$, fine particulate matter with ($PM_{2.5}$; particles with diameter less than 2.5 μm) and $PM_{10}$ (particles with diameter less than 10 μm) in urban areas, and treats the photochemistry involving $O_3/NO/NO_2$ with the specific aim to predict concentrations of $NO_2$; the major pollutant in many cities of northern Europe.

Figure 1 illustrates the modules and processes of the EPISODE model. Modules that belong to the CityChem extension are shown in boxes with a magenta frame. Currently, three different mechanisms are available in EPISODE-CityChem for photochemistry on the Eulerian grid: (1) the EMEP45 chemistry scheme developed at NILU, (2) the EmChem03-mod scheme and (3) the EmChem09-mod scheme.

For the numerical solution of the stiff non-linear system of ordinary differential equations resulting from the chemical

reaction systems the TWOSTEP solver (Verwer and Simpson, 1995; Verwer et al., 1996) is applied in CityChem.

In the sub-grid components, the PSS assumption involving $O_3/NO/NO_2$ is replaced by the EP10-Plume scheme. The dispersion close to point and line sources is modified in the sub-grid component. In the point source sub-grid model the WMPP (WORM Meteorological Pre-Processor) is integrated to calculate the wind speed at the plume height. In the line source sub-grid model, the simplified street canyon model (SSCM) is integrated to calculate pollutant dispersion in street canyons.

In EPISODE-CityChem, a regular receptor grid is defined, for which time-dependent surface concentrations of the pollutants at receptor points are calculated by summation of the Eulerian grid concentration of the corresponding grid cell (i.e. the background concentration), and the concentration contributions from the Gaussian sub-grid models due to line source and point source emissions. This way, surface concentration fields of pollutants for the entire city on a horizontal resolution of (currently)



100 m are obtained. The modules of the CityChem extension for photochemistry and source dispersion are described in detail in the remainder of this section.

## 2.1 Extensions to the photochemistry

Atmospheric gas-phase chemical reactions are described by ordinary differential equations (ODEs). The ODE set of reactions
is considered stiff because the chemical e-folding lifetimes of individual gases vary by many orders of magnitude in the urban atmosphere (approx. from $10^{-6}$ to $10^6\,\mathrm{s}^{-1}$; McRae et al., 1982). The non-linear system of the stiff chemical ODEs is solved by the TWOSTEP solver (Verwer and Simpson, 1995; Verwer et al., 1996) using fast Gauss-Seidel iterative techniques, with numerical error control and restart in case of detected numerical inaccuracies (Walker et al., 2003). The TWOSTEP solver is applied to chemical reaction mechanisms available in EPISODE-CityChem for the photochemical transformation
on the Eulerian grid (EMEP45, EmChem03-mod, and EmChem09-mod) and in the sub-grid component (EP10-Plume). For solving the EMEP45 scheme, the Gauss-Seidel iterative technique is used for all compounds except for the oxygen atoms and OH for which reactions are very fast and where we use the steady state approximation instead (Walker et al., 2003). The relative error tolerances for the TWOSTEP Solver are set to 0.1 (10 % relative error) for all chemical compounds, while the absolute error tolerances are set in a range from $2.5 \times 10^8$ molecule $\mathrm{cm}^{-3}$ to $1.0 \times 10^{15}$ molecule $\mathrm{cm}^{-3}$ depending on compound.
Photodissociation rates are specified as a function of the solar zenith angle and cloud cover, as given in Appendix A. The sink terms for dry deposition and wet removal of gases and particles are presented in Appendix B.

### 2.1.1 Development and description of the EMEP45 chemistry scheme

The EMEP45 chemistry scheme developed at NILU (Walker et al., 2003) contains 45 chemical compounds and about 70 chemical reactions, as compared to 70 compounds and about 140 reactions of the original EMEP mechanism (Simpson,
1992, 1993, 1995; Andersson-Sköld et al., 1999).

The intention of the development of EMEP45 was to obtain a condensed chemical scheme for urban areas that still captures the key aspects of the photochemistry in the urban atmosphere. The reduction of the EMEP mechanism was guided by the following considerations: first, the new chemistry scheme is applied in rather polluted urban regions and second, the residence time of the atmospheric compounds in the urban domain is normally limited to less than a day.
The main simplification in EMEP45 compared to the original EMEP mechanism is the neglect of the peroxy radical self-reactions. The self-reactions of peroxy radicals, either between organic peroxy radical ($RO_2$) and hydrogen peroxy radical ($HO_2$), or between two organic peroxy radicals,

$$HO_2 + HO_2 \quad \longrightarrow \quad H_2O_2 + O_2 \tag{R1}$$

$$RO_2 + HO_2 \quad \longrightarrow \quad RO_2H + O_2 \tag{R2}$$

$$RO_2 + RO_2 \quad \longrightarrow \quad products\,, \tag{R3}$$

are in competition with the reaction of $RO_2$ (or $HO_2$) with NO leading to photochemical ozone formation:

$$RO_2 + NO \quad \longrightarrow \quad RO + NO_2 \tag{R4}$$





At ambient levels of $NO_X$ typical of moderately or more polluted areas, reactions (R.1) - (R.3) will be negligible compared with reaction (R.4). Thus, all reactions of organic peroxy radicals of type (R.2) and (R.3) were omitted in the EMEP45 scheme. However, due to their relevance, the reaction of $HO_2$ with the methyl peroxy radical ($CH_3O_2$) and the $HO_2$ self-reaction (R.1) were included. EMEP45 includes a simple four-reaction scheme for the oxidation of isoprene ($C_5H_8$) with the OH radical. All

reaction rates and coefficients in EMEP45 are according to the International Union of Pure and Applied Chemistry (IUPAC) 2001 recommendation (Atkinson et al., 2000).

### 2.1.2  Development of the EmChem03-mod scheme and the EmChem09-mod scheme

The EMEP45 scheme was updated in recent years at the Helmholtz-Zentrum Geesthacht (HZG). All reaction rate constants were updated in accordance with the default chemistry scheme EmChem09 of the EMEP/MSC-W model (Simpson et al., 2012).

The resulting scheme is called EmChem03-mod and consists of 45 gas-phase species, 51 thermal reactions and 16 photolysis reactions, as listed in Table S2. The most important technical change compared to EMEP45 is that the new scheme can be dynamically updated and further extended with new chemical reactions and compounds. The chemical pre-processor of the EMEP/MSC-W model, GenChem, developed at the EMEP group (Simpson et al., 2012) is used to convert lists of input chemical species and reactions to differential equations of the TWOSTEP solver in Fortran 90 code. This makes the update and

extension of the new scheme entirely flexible.

In the next step, the EmChem09-mod scheme (Table S3) was developed based on the current EMEP chemistry mechanism EmChem09 (Simpson et al., 2012), by (1) replacing the detailed isoprene chemistry with the simplified isoprene reaction scheme from EMEP45, (2) adding monoterpene oxidation reactions and (3) including semi-volatile organic compounds (SVOCs) as reaction products which can potentially act as precursors for secondary organic aerosol (SOA) constituents.

EmChem09-mod includes the reactions between organic peroxy radicals and $HO_2$ and other organic peroxy radicals; it is, therefore, appropriate for low $NO_X$ conditions in rural and sub-urban areas of the city domain where levels of $NO_X$ are often below 1 ppbv. With EmChem09-mod the chemistry of BVOCs, emitted from urban vegetation, can be simulated. Two monoterpenes, $\alpha$−pinene and limonene, are model surrogates to represent slower and faster-reacting monoterpenes ($\alpha$−pinene: $5.32\times10^{-11}$ cm$^3$ s molecule$^{-1}$; limonene: $1.7\times10^{-10}$ cm$^3$ s molecule$^{-1}$; for the OH-reaction, both at 298 K).

The scheme considers the OH-initiated oxidation of isoprene, as well as the oxidation of $\alpha$−pinene and of limonene by OH, $NO_3$ and $O_3$. Limonene has two reactive sites (double bonds) allowing for a rapid reaction chain to oxidation products with low vapour pressure. The lumped reaction scheme of $\alpha$−pinene is adopted from Bergström et al. (2012) and that of limonene is based on Calvert et al. (2000). In total, EmChem09-mod includes 70 compounds, 67 thermal reactions and 25 photolysis reactions.

### 30  2.1.3  Development and description of the EP10-Plume chemistry scheme

In the sub-grid components, i.e. the Gaussian models for line and point source dispersion, the PSS assumption involving $O_3/NO/NO_2$ was replaced by the EP10-Plume scheme, for computation of the chemistry at the local receptor grid points. EP10-

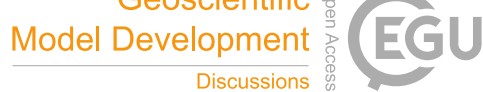



Plume includes only the reactions of $O_3$, NO, $NO_2$, $HNO_3$, CO and the photochemical oxidation of formaldehyde (HCHO). It contains 10 compounds and 17 reactions; Table S4 provides a list.

Only a small portion of $NO_X$ from motor vehicles and combustion sources is in the form of $NO_2$, the main part being NO. The largest fraction of ambient $NO_2$ originates from the subsequent chemical oxidation of NO. The only reactions considered to be relevant in the vicinity of $NO_X$ emission sources are:

$$NO + O_3 \quad \longrightarrow \quad NO_2 + O_2 \qquad\qquad (R5)$$

$$NO_2 + h\nu \quad \longrightarrow \quad NO + O(^3P) \qquad\qquad (R6)$$

$$O(^3P) + O_2 \quad \longrightarrow \quad O_3 \;. \qquad\qquad (R7)$$

For conditions in northern Europe, an instantaneous equilibrium between the three reactions relating NO, $NO_2$ and $O_3$ is assumed, the so-called PSS, and implemented in the EPISODE model. In EP10-Plume the three reactions are however treated explicitly. Reactions occurring with negligible rates at $NO_X$ levels typically of moderately or highly polluted areas were excluded from the scheme. HCHO and acetaldehyde are important constituents of the vehicle exhaust gas (e.g. Rodrigues et al., 2012). The photolysis of HCHO is a source of $HO_2$ radicals:

$$HCHO + h\nu \quad \longrightarrow \quad CO + 2\,HO_2 \qquad\qquad (R8a)$$

$$HCHO + h\nu \quad \longrightarrow \quad CO + H_2 \qquad\qquad (R8b)$$

$$OH + HCHO \quad \longrightarrow \quad CO + 2\,HO_2 \qquad\qquad (R9)$$

$$OH + CO \quad \longrightarrow \quad HO_2 + CO_2 \;. \qquad\qquad (R10)$$

HCHO also reacts with the OH radical to give two $HO_2$ radicals. $HO_2$ competes with ozone for the available NO (reaction (R.4)) and the reaction between $HO_2$ and NO results in additional NO-to-$NO_2$ conversion. Since the generation of $HO_2$ radicals through HCHO photolysis does not depend on the entrainment of photo-oxidants from the background air, it can trigger the photochemical reaction cycle even in traffic plumes very close to the source. Carbon monoxide (CO) has a lifetime of about two months towards OH (at $[OH] = 1.2\times10^6$ molecules $cm^{-3}$). Reaction (R.10) is therefore not relevant near sources and of very low relevance on urban scale. For completeness of the OH-to-$HO_2$ cycling, (R.10) was however included in EP10-Plume.

## 2.2 Extensions to the source dispersion

Sub-grid models to resolve the dispersion close to point sources and line sources are embedded in the EPISODE model to account for sub-grid variations as a result of emissions along open roads and streets as well as along plume trajectories from elevated point source releases. The sub-grid model for line sources, i.e. open road and urban street traffic, is the Gaussian model HIWAY-2 (Petersen, 1980) from US EPA with modifications. The sub-grid model for point sources, e.g. stacks of industrial plants and power plants, is the Gaussian segmented plume trajectory model SEGPLU (Walker and Grønskei, 1992). SEGPLU computes and keeps a record of subsequent positions of plume segments released from a point source and the corresponding pollutant concentration within each plume segment. The vertical position of the plume segment is calculated from the plume rise





of the respective point source. A detailed description of the implementation of HIWAY-2 and SEGPLU in the EPISODE model is given in part one (Hamer et al., 2019). In this section, the extensions of the sub-grid models for simulation of the dispersion near sources within CityChem are described.

In CityChem, a simplified street canyon model (SSCM) to compute concentrations for receptor points that are located in
street canyons is introduced. The street canyon model follows in most aspects the Operational Street Pollution Model (OSPM; Berkowicz et al., 1997). A fundamental assumption of this model is that when the wind blows over a rooftop in a street canyon, an hourly averaged recirculation vortex is always formed inside the canyon (Hertel and Berkowicz, 1989). The part of the street canyon covered by the vortex of recirculating air is called the recirculation zone.

### 2.2.1 Implementation of a simplified street canyon model (SSCM) for line source dispersion

The contribution of a line source s to the receptor concentration located within an urban street canyon is the sum of the direct contribution ($C_{\text{scdir,s}}$) from the traffic plume plus a contribution from the recirculation of the traffic plume ($C_{\text{screc,s}}$) due to the vortex inside the canyon (Berkowicz et al., 1997):

$$C_{\text{line,s}} = C_{\text{scdir,s}} + C_{\text{screc,s}}. \tag{1}$$

The leeward receptor inside a street canyon is exposed to the direct contributions from the emissions inside the recirculation
zone (unless the wind direction is close to parallel) and a recirculation contribution. For the receptor on the windward side, only the emissions outside the recirculation zone are considered for the direct contribution. If the recirculation zone extends through the whole canyon, no direct contribution is given to the windward receptor. The length of the recirculation zone, $L_{\text{rec}}$, is estimated as being twice the average building height of the canyon and limited by $W_{\text{sc}}$. The recirculation zone is modelled as a trapezium with the upper length $L_{\text{top}}$ being half of the baseline length $L_{\text{base}}$, where $L_{\text{base}}$, is defined as $\min(L_{\text{rec}}, L_{\text{max}})$.
The direct contribution is calculated using a Gaussian plume model; $C_{\text{scdir,s}}$ at the receptor point located at distance $x$ from the line source, i.e. from the mid-line of the street, by integrating along the wind path at street level. The integration path depends on wind direction, the extension of the recirculation zone and the street canyon length (Hertel and Berkowicz, 1989):

$$\int_{x_{\text{start}}}^{x_{\text{end}}} \frac{dC_{\text{scdir,s}}}{dx}dx = \sqrt{\frac{2}{\pi}}\frac{Q_{\text{s}}}{W_{\text{sc}}\sigma_{\text{w}}} \cdot \int_{x_{\text{start}}}^{x_{\text{end}}} \frac{1}{x + \frac{u_{\text{street}}h_0}{\sigma_{\text{w}}}}dx. \tag{2}$$

Where $h_0$ is a constant that accounts for the height of the initial pollutant dispersion ($h_0 = 2$ m is used in SSCM), $\sigma_{\text{w}}$ is the
vertical velocity fluctuation due to mechanical turbulence generated by wind and vehicle traffic in the street, and $u_{\text{street}}$ is the wind speed at street level, calculated assuming a logarithmic reduction of the wind speed at rooftop towards the bottom of the street. Note that the wind direction at street level in the recirculation zone is mirrored compared to the roof level wind direction. Outside the recirculation zone, the wind direction is the same as at roof level. The vertical velocity fluctuation is calculated as a function of the street level wind speed, and the traffic produced turbulence by the following relationship (Berkowicz et al.,
30 1997):

$$\sigma_{\text{w}} = \sqrt{(\alpha_{\text{s}}u_{\text{street}})^2 + (\sigma_{\text{w0}})}, \tag{3}$$

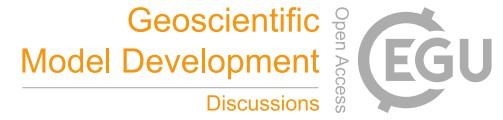



where $\alpha_s$ is a proportionality constant, empirically assigned a value of 0.1, and $\sigma_{w0}$ is the traffic-induced turbulence, in SSCM assigned a value of $0.25 \, \text{m s}^{-1}$, typical for traffic on working days between 8 a.m. and 7 p.m. in situations where traffic-induced turbulence dominates (Kastner-Klein et al., 2000; fig. 6 therein).

The integration path for Eq. (2) begins from $x_{\text{start}}$ which is defined as the distance from the receptor point where the plume has the same height as the receptor, which is zero in the case that $h_0$ is smaller or equal to the height of the receptor. The upper integration limit is $x_{\text{end}}$ defined by tabular values in Ottosen et al. (2015; table 3 therein). The integration is performed along a straight line path against the wind direction. The calculation of the maximum integration path, $L_{\text{max}}$, depends on the wind direction with respect to the street axis, $\theta_{\text{street}}$, i.e. the angle between the street and the street level wind direction (Ottosen et al., 2015).

The contribution from recirculation is computed using a simple box model and concentrations are computed assuming equality of the inflow and outflow of the pollutant. This is expressed by the relationship (Berkowicz et al., 1997):

$$C_{\text{screc,s}} = \frac{Q_s}{W_{\text{sc}}} \cdot \frac{L_{\text{base}}}{\sigma_{\text{wt}} L_{\text{top}} + \sigma_{\text{hyp}} L_{\text{hyp}}}, \tag{4}$$

where $\sigma_{\text{wt}}$ is the ventilation velocity of the canyon as given by Hertel and Berkowicz (1989) and $\sigma_{\text{hyp}}$ is the average turbulence of the hypotenuse of the trapezium (slant edge towards the opposite street side). The length of the hypotenuse of the trapezium is calculated as $L_{\text{hyp}} = \sqrt{(L_{\text{base}}/2) + H_{\text{sc}}^2}$, assuming the leeward side edge of the recirculation zone to be the vertical building wall, with the length of the building height.

For a given receptor point, the concentration contribution from a line source is calculated either by HIWAY-2 or SSCM. HIWAY-2 does not calculate line source concentration contributions to receptors that are upwind of a line source and for receptor points that are very close to the line source. For all windward/leeward receptor points, which are (1) located within a model grid cell defined as street canyon cell (see below), and (2) located close enough to a line source (i.e. within the actual street canyon), and (3) located at a road link with length $> 8 \, \text{m}$, the concentration contribution from the street is calculated by SSCM. For all windward receptors which do not fulfil these conditions, the concentration contribution is calculated by HIWAY-2.

The complex and divers geometry of street canyons is approximated by three generic types for which average street canyon geometry properties are applied (Table 1). Each line source for which the geometric mid-point is located in a grid cell with urban land use (land use classes 32–35 defined in TAPM) is identified as a potential street canyon. A disadvantage of this method is that some streets and roads, especially in the sparse-built urban areas outside the inner city, will be classified as street canyon despite being open roads with open spaces between buildings.

Furthermore, it is assumed that all buildings at the street canyon line source have the same average building height, $H_{\text{sc}}$, and that there are no gaps between the buildings. The average building heights for the TAPM land use classes were obtained by the intersection of the 3-D city model LoD1-DE Hamburg (LGV, 2014) - which contains individual building heights - with the urban land use information of CORINE (CLC, 2012). The width of the street canyon, $W_{\text{sc}}$, is defined as twice the width of the (line source) street width $W$, to account for sidewalks and to avoid too narrow canyons. The length of the street canyon, $L_{\text{sc}}$, corresponds to the length of the line source within the grid cell.


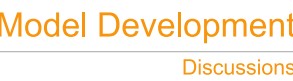
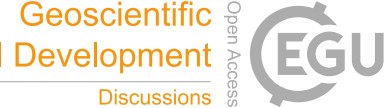

### 2.2.2 Implementation of the WMPP based plume rise model

Plume rise for elevated point sources due to momentum or buoyancy is calculated within the point source sub-grid dispersion model based on the plume rise equations originally presented by Briggs (1969, 1971 and 1975) and takes into account different boundary layer (BL) stability conditions as characterized by the inverse Monin-Obukhov length. Input parameters such as

pollutant emission rates, stack height, diameter, exhaust gas temperature and exit velocity as well as height and width of the building adjacent to the stack can be provided for each point source by the model user.

The wind speed profile function of the WMPP (WORM Meteorological Pre-Processor) is utilized in the CityChem extension to calculate the wind speed at the actual plume height. The meteorological pre-processor WMPP has been developed as part of NILU's WORM open road line source model (Walker, 2011, 2010), to calculate various meteorological parameters needed

by the WORM model. In the current version of the WORM model, the profile method is applied, using hourly observations of wind speed at one height, e.g. 10 m, and temperature difference between two heights, e.g. 10 m and 2 m, to calculate the other derived meteorological parameters.

Given the above input data, and an estimate of the momentum surface roughness, WMPP calculates friction velocity ($u_*$), temperature scale ($\theta_*$) and inverse Obukhov length ($L^{-1}$) according to Monin-Obukhov similarity theory. These quantities are

calculated by solving the following three nonlinear equations:

$$u_* = \frac{\kappa \cdot \Delta u}{\int\limits_{z_{u1}}^{z_{u2}} \varphi_m\left(z, L^{-1}\right) z^{-1} dz}; \qquad \theta_* = \frac{\kappa \cdot \Delta \theta}{\int\limits_{z_{t1}}^{z_{t2}} \varphi_h\left(z, L^{-1}\right) z^{-1} dz}; \qquad L^{-1} = \frac{\kappa \cdot g}{T_{\text{ref}}} \frac{\theta_*}{u_*^2}, \tag{5}$$

where $\kappa$ is Von Kármán's constant (0.41), $g$ is the acceleration of gravity (9.81 m s$^{-2}$); $\Delta u$ is the wind speed difference between heights $z_{u2}$ and $z_{u1}$, where $z_{u2}$ is e.g. 10 m, and $z_{u1} = z_{0m}$ where the wind speed is zero, so that $\Delta u = u_{10m} - 0 = u_{10m}$; $\Delta \theta$ is the difference in potential temperature between heights $z_{t2}$ and $z_{t1}$, which are e.g. 10 m and 2 m respectively, so that $\Delta \theta = T_{10m} -$

$T_{2m} + 0.01$ and where $T_{\text{ref}}$ is a reference temperature, here taken to be the average of $T_{2m}$ and $T_{10m}$.

In Eq. (5), the similarity functions $\varphi_m$ and $\varphi_h$ are defined as follows (Högström, 1996):

$$\varphi_m\left(z, L^{-1}\right) = \begin{cases} \left(1 + \alpha_m\left(zL^{-1}\right)\right)^{-\frac{1}{4}} & \text{if } L^{-1} < 0 \text{ (unstable atm.)} \\ 1 + \beta_m\left(zL^{-1}\right) & \text{if } L^{-1} > 0 \text{ (stable atm.)} \\ 1 & \text{if } L^{-1} = 0 \text{ (neutral atm.)} \end{cases} \tag{6}$$

and

$$\varphi_h\left(z, L^{-1}\right) = \begin{cases} Pr_0\left(1 + \alpha_h\left(zL^{-1}\right)\right)^{-\frac{1}{2}} & \text{if } L^{-1} < 0 \text{ (unstable atm.)} \\ Pr_0\left(1 + \beta_h\left(zL^{-1}\right)\right) & \text{if } L^{-1} > 0 \text{ (stable atm.)} \\ Pr_0 & \text{if } L^{-1} = 0 \text{ (neutral atm.)} \end{cases} \tag{7}$$

where $Pr_0 = 0.95$ is the Prandtl number for neutral conditions, and where the empirical coefficients are defined as $\alpha_m = -19.0$, $\alpha_h = -11.6$, $\beta_m = 5.3$ and $\beta_h = 8.2$.





This set of similarity functions is then used to calculate vertical profiles of temperature and wind speed. The temperature at a height (in m above ground) is thus calculated by

$$T_z = T_{z_{\mathrm{ref}}} - \frac{g}{c_p}\left(z - z_{\mathrm{ref}}\right) + \frac{\theta_*}{\kappa} \int\limits_{\nu=z_{\mathrm{ref}}}^{\nu=z} \varphi_h\left(\nu, L^{-1}\right)\nu^{-1}d\nu, \tag{8}$$

where $z_{\mathrm{ref}} = 10$ m. Similarly, the wind speed at height $z$ (m) above ground is calculated by

$$u_z = u_{z_{\mathrm{ref}}} + \frac{u_*}{\kappa} \int\limits_{\nu=z_{\mathrm{ref}}}^{\nu=z} \varphi_m\left(\nu, L^{-1}\right)\nu^{-1}d\nu. \tag{9}$$

In CityChem, WMPP is used in the sub-grid point source model to calculate the wind speed at plume height according to Eq. (9). WMPP can also be used to calculate the convective velocity scale $w_*$ and the mixing height $h_{\mathrm{mix}}$, but this is not implemented in CityChem.

## 2.3 Additional modifications

Here we describe the modifications in the CityChem extension to read hourly 3-D boundary concentrations from the output of the CMAQ model and to determine sub-grid concentrations from a regular receptor grid in the surface model layer.

### 2.3.1 Adapting 3-D boundary conditions from the CMAQ model

CityChem has the option to use the time-varying 3-D concentration field at the lateral and vertical boundaries from the CMAQ model as initial and boundary concentrations for selected chemical species. The adaption of boundary conditions from CMAQ output in the EPISODE model is based on the implementation for boundary conditions from the Copernicus Atmosphere Monitoring Service (CAMS; http://www.regional.atmosphere.copernicus.eu/) as described in part one (Hamer et al., 2019). The regional background concentrations are adopted for the grid cells (outside the computational domain) directly adjacent to the boundary grid cells of the model domain and for the vertical model layer that is on top of the highest model layer. The outside grid cell directly adjacent to boundary grid cell is filled with the CMAQ concentration value for inflow conditions and with the concentration value of the boundary grid cell for outflow conditions (zero-concentration gradient at the outflow boundary).

The utility BCONCC v2.0 (included in the CityChem distribution) is used to produce EPISODE-format binary files containing hourly varying 3-D boundary concentrations for the most relevant chemical compounds. BCONCC makes use of the IOAPI version 3.1 library (https://www.cmascenter.org/ioapi) to access CMAQ output files. The IOAPI (Models-3/EDSS Input/Output Application Programming Interface) provides a variety of data structure types for organizing the data and a set of data access routines.

The preparation of chemical boundary conditions from CMAQ model output is done in two steps. First, the city's 3-D domain extent plus one grid cell to each side is cut out from the CMAQ model grid, interpolating the hourly concentrations to the horizontal main grid resolution of EPISODE using bilinear interpolation. Second, EPISODE-format binary files for




boundary conditions (BCON files) containing background concentrations of all individual CityChem compounds are created for the defined model domain in the required input format. Linear interpolation is used to convert concentrations from the vertical layers of the CMAQ model to the vertical layers of the EPISODE-CityChem model. Temperature and pressure of the METCRO3D file (meteorological input file of the CMAQ simulation) are used to convert the concentration of gaseous

compounds from mixing ratio (ppm) to mass-based concentrations ($\mu g \, m^{-3}$).

### 2.3.2   Description of the regular receptor grid

In the CityChem extension, a regular receptor grid is defined, for which time-dependent surface concentrations of the pollutants at receptor points are calculated by summation of the Eulerian grid concentration of the corresponding grid cell (i.e. the background concentration), and the concentration contributions from the sub-grid models due to dispersion of line source and

point source emissions. Regular receptor grids with typical resolution $100 \times 100 \, m^2$ have also been used in earlier versions of EPISODE, but primarily for capturing sub-grid scale concentration contributions from larger industrial point sources. The establishment of a regular receptor grid is an integral part of CityChem to enable higher resolution output required for comparison with monitor data acquired near line sources. Line sources are a major source of pollutant emissions affecting the inner-city air quality; thus the use of the regular receptor grid provides information at much higher spatial resolution than the Eulerian

grid output alone. The regular receptor grid in the EPISODE-CityChem differs from the downscaling approach by Denby et al. (2014) which allocates sampling points at high density along roads and other line sources but much fewer further away from the line sources. While Denby et al. (2014) interpolate the model-computed high-density set of receptor concentrations to the desired output resolution using ordinary kriging, EPISODE-CityChem gives as output the receptor point concentrations on a regular 2-D grid covering the entire model domain.

The instantaneous concentration $C_{\mathrm{rec}}$ in an individual receptor point $r^*$ of the receptor grid with coordinates $(x_r, y_r, z_r)$ is defined as:

$$C_{\mathrm{rec}}(r^*) = C_m + \sum_{s=1}^{S} C_{\mathrm{line},s} + \sum_{p=1}^{P} C_{\mathrm{point},s} \,, \tag{10}$$

where $C_m$ is the main grid concentration of the grid cell $(x, y, 1)$ in which the receptor point is located. The grid (background) concentration $C_m$ used in Eq. (10) corresponds to a modified Eulerian 3-D grid concentration, i.e. $C(x, y, z)$, to avoid that emis-

sions of point and lines sources are counted twice. $C_{\mathrm{point},p}$ is the instantaneous concentration contribution of point source $p$ calculated by the point source sub-grid model and $C_{\mathrm{line},s}$ is the instantaneous concentration contribution of line source $s$ calculated by the line source sub-grid model. Since $C_{\mathrm{rec}}$ is not added to the main grid concentration but kept as separate (diagnostic) variable, double-counting of emitted pollutant mass is prevented. In the CityChem extension, receptor point concentrations represent the high-resolution ground concentration of a cell with the grid cell area of the receptor grid.

On the 3-D Eulerian grid, time-dependent concentration fields of the pollutants are calculated by solving the advection/-diffusion equation with terms for chemical reactions, dry deposition and wet deposition, and area emissions. The hourly 2-D and 3-D fields of meteorological variables and the hourly 2-D fields of area emissions are given as input to the model with the spatial resolution of the Eulerian grid. As the model steps forward in time, an accurate account of the total pollutant mass from



the area, point and line sources is kept within the Eulerian grid model component. Emissions from line sources are added to the Eulerian grid concentrations each model time step.

## 3 Test of different model extensions

For the test of the various model extensions, EPISODE was run as a 1-D column model, with vertical exchange as only trans-
port process. Emissions were injected into the ground cell (grid centre at UTM coordinates: $(X)\,568500, (Y)\,5935550, 32\,N$) with an area of $1 \times 1\,\text{km}^2$ and flat terrain ($15\,\text{m a.s.l.}$). Table 2 shows the general setup for the 1-D column and the specific configuration for the tests. Mixing height, surface roughness and friction velocity were kept constant ($h_{\text{mix}} = 250\text{m}$, $z_0 = 0.8\,\text{m s}^{-1}$, $u_* = 0.12\,\text{m s}^{-1}$). Hourly varying meteorological variables included: air temperature, temperature gradient, relative humidity, sensible and latent heat fluxes, total solar radiation and cloud fraction. The test simulations are performed for a period of five
days, results were taken as an average of the period.

### 3.1 Test of the photochemistry on the Eulerian grid

#### 3.1.1 Tests of the original EMEP45 photochemistry

When the condensed EMEP45 photochemistry was developed various tests were carried out to compare the condensed mechanism with the standard EMEP chemical mechanism. Results from box model studies with the two chemical mechanisms
revealed that there were generally small differences between the full and the condensed chemical mechanisms. Even for conditions more representative for a rural environment, the difference between the standard EMEP and the condensed mechanism was small. For these more rural conditions, the condensed mechanism gave slightly lower levels of NO and $NO_2$, while the ozone concentration was almost identical in the two mechanisms. For urban conditions, these differences were expected to be significantly smaller.

The EPISODE model with the condensed EMEP45 mechanism furthermore participated in the CityDelta project (Cuvelier et al., 2007) where it was applied to the city of Berlin. CityDelta was the first in a series of projects (later named EuroDelta) dedicated to photochemical model inter-comparisons. When evaluated against observations of $NO_2$ and $O_3$, the EPISODE model with the EMEP45 chemistry performed favourably when compared to the suite of atmospheric models participating in the CityDelta project (Walker et al., 2003).

#### 25  3.1.2 Test of ozone formation with EmChem03-mod

The ozone-$NO_X$-VOC sensitivity of the EmChem03-mod scheme in the Eulerian model component was analysed by repeated runs with varying emissions of $NO_X$ and NMVOC using the daily cycle of mean summer meteorology with clear sky but low wind speed ($0.1\,\text{m s}^{-1}$). The ozone net production in the runs was taken at the maximum daily $O_3$ during the simulation.

    An area source of traffic emissions of $NO_X$ and NMVOC in the ground cell of the 1-D column was activated in the test. The
variation of ozone precursor emissions from the traffic area source was done in a systematic way in order to derive the ozone





isopleth diagram (Fig. 2a), which shows the rate of $O_3$ production (ppb h$^{-1}$) as a function of $NO_X$ and NMVOC concentrations. Compound abundances are given in mixing ratios (ppb) for this test to enable comparison with the literature on ozone formation potentials.

The ozone-precursor relationship in urban environments is a consequence of the fundamental division into a $NO_X$-limited
and a VOC-limited chemical regimes. VOC/$NO_X$ ratios are an important controlling factor for this division of chemical regimes (Sillman, 1999). VOC-limited chemistry generally occurs in urban centres where $NO_2$ concentrations are high due to traffic emissions. The rural areas downwind locations of the city are typically $NO_X$-limited (Ehlers et al., 2016).

The "ridgeline" of the ozone isopleth diagram marks the local maxima of the $O_3$ production and divides two different photochemical regimes. Below the line is the $NO_X$-limited regime, where $O_3$ increases with increasing $NO_X$ while it is hardly
affected by increasing VOC. Above the line is the VOC-limited regime, where $O_3$ increases with increasing VOC and decreases with increasing $NO_X$. The "ridgeline" in Fig. 2a follows a line of constant VOC/$NO_X$ ratio, in the case of EmChem03-mod it is close to the ratio 10:1; whereas a slope of 8:1 is more typically found (e.g. Dodge, 1977). The traffic NMVOC mixture includes a high share of aromatics (35 %) represented by o-xylene in the model. Due to the high reactivity of the NMVOC mixture, the "ridgeline" is tilted towards higher VOC/$NO_X$ ratios compared to the ozone isopleths for a NMVOC mixture with lower
reactivity.

The split into $NO_X$-limited and VOC-limited regimes are closely associated with sources and sinks of odd hydrogen radicals (defined as the sum of OH, $HO_2$ and $RO_2$). Odd hydrogen radicals are produced in the photolysis of ozone and intermediate organics such as, for example, formaldehyde. Odd hydrogen radicals are removed by the reactions that produce hydrogen peroxide (R.1) and organic peroxides (R.2). They are also removed by reaction with $NO_2$, producing nitric acid ($HNO_3$),
according to:

$$OH + NO_2 + M \quad \longrightarrow \quad HNO_3 \ . \tag{R11}$$

When peroxides represent the dominant sink for odd hydrogen, then the sum of peroxy radicals is insensitive to changes in $NO_X$ or VOC. This is the case for the concentrations represented as solid and dash-dotted lines in Fig. 2c–d. Doubling $NO_X$ emissions from solid lines to dash- dotted lines only marginally changes the peroxy radical sum concentration (Fig. 2d).
When $HNO_3$ is the dominant sink of odd hydrogen, then the OH concentration is determined by the equilibrium between the producing reactions (e.g. photolysis of $O_3$) and the loss reaction (R.11) and thus decreases with increasing $NO_X$ (Fig. 2c–d; from dashed to dotted lines), while it is either unaffected, or increases due to photolysis of intermediate organics, with increasing VOC.

Plotting the isopleths for the ratio of the production rate of peroxides to the production rate of $HNO_3$ (Fig. 2b) shows that
this ratio is closely related to the split between $NO_X$-limited and VOC-limited regimes. The ratio is typically 0.9 or higher for $NO_X$-limited conditions, and 0.1 or less for VOC-limited conditions (Sillman, 1999). The "ridgeline" that separates the two regimes should be at a ratio of 0.5 (Sillman, 1999); which is the case in Fig. 2b. However, the curves representing the ratio are shifted towards higher $NO_X$ mixing ratios compared to the isopleth diagram for the ratio displayed in Sillman (1999; figure 8 therein). For instance, for 100 ppbC NMVOC and 5 ppb $NO_X$, the ratio is below 0.1 (VOC-limited) in the isopleth



diagram of Sillman (1999), while it is 1.3 ($NO_X$-limited) in Fig. 2b. The reason for this discrepancy is the lack of the reactions producing organic peroxides ($RO_2H$) in EmChem03-mod and thus the reduced removal of odd hydrogen in conditions with high VOC/$NO_X$ ratio. In conditions with $NO_X$ below 20 ppbv, EmChem03-mod has a too high efficiency of the NO-to-$NO_2$ conversion via reaction (R.4).

### 3.1.3    Test of EmChem09-mod photochemistry

The EmChem09-mod scheme was compared to the EmChem03-mod scheme for conditions with relatively low levels of $NO_X$ ($< 20\,\mu g\,m^{-3}$). The configuration of the test was the same as in Sect. 3.1.2 with an area source of traffic emissions of $NO_X$ ($0.043\,g\,s^{-1}$ in the $1 \times 1\,km^2$ ground cell) and varying emissions of NMVOC corresponding to VOC/$NO_X$ ratios of 4:1, 8:1 and 15:1. The daily cycle of ozone with EmChem09-mod shows $O_3$ concentrations which are lower for VOC/$NO_X$ ratio of 4:1 (VOC-limited), similar for VOC/$NO_X$ ratio of 8:1 (transition) and higher for VOC/$NO_X$ ratio of 15:1 ($NO_X$-limited) than with EmChem03-mod (Fig. 3a). Compared to EmChem03-mod, the EmChem09-mod scheme includes reactions between organic peroxy radicals and $HO_2$ and other organic peroxy radicals. In conditions with low levels of $NO_X$, the rates from these reactions will be in competition to the reaction rates of organic peroxy radicals with NO.

The lower $O_3$ with EmChem09-mod in VOC-limited conditions is related to the competition between the organic peroxy radical self-reactions and the reaction with NO, preventing additional NO-to-$NO_2$ conversion. Compared to EmChem03-mod, the removal of odd hydrogen through (R.11) to form $HNO_3$ weakens (Fig. 3b), the formation of $H_2O_2$ and organic peroxides enhances (Fig. 3c), and the formation of peroxyacetyl nitrate (PAN) is suppressed (Fig. 3d); the latter due to the competing reaction between the acetyl peroxy radical ($CH_3COO_2$) and $HO_2$, not included in EmChem03-mod. As a result, less $NO_2$ is lost and the $NO_X$ concentrations in EmChem09-mod increase compared to EmChem03-mod (Fig. S1), which reduces ozone production in the VOC-limited regime.

The higher $O_3$ with EmChem09-mod in $NO_X$-limited conditions is related to the much higher production of peroxides and the reduced production of PAN and $HNO_3$ compared to EmChem03-mod. Despite a slightly lower formation of. The $NO_X$ concentrations in EmChem09-mod are higher, which increases ozone production in the $NO_X$-limited regime.

## 3.2    Test of EP10-Plume sub-grid photochemistry

The photochemistry in the sub-grid component of EPISODE-CityChem was tested for dispersion from a single line source aligned in the SE-NW diagonal of the $1 \times 1\,km^2$ grid cell. The line source was oriented perpendicular to wind direction; emitting $NO_X$ and NMVOC with a ratio of 1:2. The HIWAY-2 line source model was used in the test (SSCM was not activated). Photochemistry tests were made as follows: (1) no chemistry; (2) photochemical steady state assumption (PSS) for $O_3$/NO/$NO_2$ (default); and (3) with EP10-Plume, using the numerical solver. Inside the centre cell, ground air concentrations downwind of the line source were recorded using additional receptor points every 10 meters up to a distance of 300 m from the line source.

Comparing $O_3$ (black lines), $NO_2$ (red lines) and NO (blue lines) concentrations from the three tests with increasing down-wind distance $x$ shows that dilution alone (test with no chemistry; Fig. 4a) leads to a decay of NO which follows a power function of the form $y = ax - b$ while $O_3$ remains constant at level of the background concentration ($30\,\mu g\,m^{-3}$).



Applying the PSS reduces $O_3$ immediately at the line source by reaction (R.5) to one-fourth of the concentration without chemistry. At the line source (0 m distance), PSS converts roughly 15 µg m$^{-3}$ NO to 21 µg m$^{-3}$ NO$_2$, as deduced from the differences between the no chemistry and the PSS test run. The third option, EP10-Plume, gives very similar results to PSS, with $O_3$, NO and NO$_2$ concentrations deviating by at most 4 % from the solution of the PSS. In EP10-Plume, the line-emitted

HCHO during daytime reacts with OH or undergoes photolysis to give HO$_2$ radicals. However, the odd hydrogen radicals are rapidly removed by reaction (R.11) and the effect of emitted HCHO on $O_3$ is negligible. It is noted that HCHO accounts for only 2.7 % of the traffic NMVOC emissions. Further testing showed that the share of HCHO has to be increased by a factor of 10 or more (for the same VOC emission rate) in order to exceed the PSS concentration of $O_3$ close to the line source.

### 3.3   Test of the source dispersion extensions

### 3.3.1   Test of SSCM for line source dispersion

Tests with the simplified street canyon model (SSCM; see Sect. 2.2.1) were performed for different roof level wind speeds (0.5, 1.0, 1.5, 2.0, 4.0, and 6.0 m s$^{-1}$) and compared to results from the HIWAY-2 line source dispersion model. The street canyon was oriented along the SE-NW diagonal of the grid cell, canyon width was 18 m and average building height was 18 m, with no gaps between buildings. Receptor points were placed symmetrically on the northeast side and the southwest side of the canyon,

in 5 m distance from the street. Time-averaged modelled concentrations of PM$_{10}$, emitted from the line source as chemically inert tracer, are shown as function of wind direction and wind speed in Fig. 5 for the northeast side (left) and southwest side (right) receptor. The wind direction dependency at the two receptors is simply shifted by 180° with respect to each other, due to the symmetric arrangement. With SSCM, the leeward concentrations are generally higher than the windward concentrations (grey-shaded areas in the figure). For both models, maximum concentrations are calculated for wind direction close to parallel

with the street (135° and 315°).

For this specific street canyon, with an aspect ratio ($W_{sc}/H_{sc}$) equal to one, the recirculation zone extends through the whole canyon at large wind speed and the windward receptor only receives a contribution from the recirculation. At low wind speed, here at 2 m s$^{-1}$ or below, the windward side starts to receive a direct contribution, because the extension of the vortex decreases at low wind speed. At wind speed below 0.5 m s$^{-1}$, the vortex disappears and the traffic generated turbulence determines the

concentration levels. Gaussian models are not designed to simulate dispersion in low-wind conditions. Therefore, a lower limit of the rooftop wind speed was placed at 0.5 m s$^{-1}$ in this test, preventing the test of lower wind speeds. It is, however, obvious from Fig. 5a that the influence of wind direction on concentrations at 0.5 m s$^{-1}$ is much reduced compared to larger wind speeds.

Similar to SSCM, the simulation with HIWAY-2 shows a local maximum at the windward side when the wind is perpen-

dicular to the street and a local minimum at the leeward side when the wind is perpendicular. In HIWAY-2, the pollution from traffic is dispersed freely away from the street because it applies to open road without buildings. In SSCM, the leeward side is influenced directly by the traffic emissions in the street and in addition by the recirculated polluted air. HIWAY-2 neglects

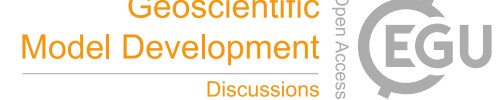


the contribution of recirculated polluted air. This is also the reason why the baseline contribution (as addition to the urban background) is higher in SSCM.

### 3.3.2 Test of WMPP based point source dispersion

The WMPP (WORM meteorological pre-processor) model code was extensively tested using meteorological observations from
a four months measurement campaign at Nordbysletta in Lørenskog, Norway in 2002 (Walker, 2011; Walker, 2010).

WMPP (see Sect. 2.2.2) is used in the plume rise module of SEGPLU for the calculation of the wind speed at 1) stack height; 2) actual plume height; and 3) at final plume height. The modification of the plume rise module is similar as in the "NILU plume" parameterization, implemented in the WRF-EMEP model; as presented in Karl et al. (2014). In comparison with two simple schemes for plume rise calculation, "NILU plume" gave lower final plume rise from an elevated point source for all
tested atmospheric stability conditions. In neutral conditions, the maximum concentration at ground ($C_{max}$) was found to be roughly twice as high as for the two simple plume rise schemes. In unstable conditions, all plume rise schemes gave similar effective emission heights.

The WMPP integration in the sub-grid point source model for the near-field dispersion around a point source was tested in different atmospheric stability conditions and compared to the standard point source parameterisation in EPISODE (termed
"default" in the following). A single point source was located at the midpoint of the $1 \times 1\,km^2$ grid cell. The dispersion of $SO_2$, treated as non-reactive tracer, released from the point source stack was studied by sampling ground air concentrations from a regular receptor grid with 100-m resolution within a radius of 2 km around the point source. Transport on the Eulerian grid was deactivated so that the test corresponds to a stand-alone test of the Gaussian point source model.

Details about the point source and resulting hourly ground concentrations (averaged for 5 days) at the location of maximum
impact, $C_{max}$, for different stability conditions (slightly stable, neutral, slightly unstable, very unstable) are summarized in Table 3.

For WMPP, the maximum impact lies within 250 m and 400 m downwind of the point source, in neutral and unstable conditions. The effective emission height, $H_{eff}$ in neutral conditions is about half of that computed by the "default" parameterisation. For WMPP, $H_{eff}$ decreases with enhanced instability (from neutral to very unstable) while $C_{max}$ increases correspondingly. The
increase of $C_{max}$ computed by WMPP is 41 %. $C_{max}$ should be roughly proportional to the square of the effective emission height (Hanna et al., 1982); thus the decrease from 40.5 m (neutral) to 32.4 m (very unstable) implies a potential increase of $C_{max}$ by 25 %. The higher increase of $C_{max}$ than expected might be due to continuous wind from one direction (225°) and the relatively low wind speed (1 m s$^{-1}$) in the test. For "default", $H_{eff}$ and $C_{max}$ are not affected by changing stability in neutral or unstable conditions; computed $C_{max}$ is a factor of 4–6 smaller than for WMPP. In stable conditions, $C_{max}$ is several orders
of magnitude smaller than in neutral and unstable conditions, for both plume rise parameterisations. The maximum impact is found in 1700 m distance from the point source, comparable to previous tests with the point source model (Karl et al., 2014).



# 4 Application of EPISODE-CityChem to air quality modelling for Hamburg

## 4.1 Setup of model experiments for the application for AQ modelling in Hamburg

EPISODE-CityChem was run as part of a one-way nested model chain, which coupled the model off-line to the CMAQv5.0.1 CTM (Byun et al., 1999; Byun and Schere, 2006; Appel et al., 2013) driven by COSMO-CLM mesoscale meteorological model

version 5.0 (Rockel et al., 2008) for the year 2012 using the ERA-Interim re-analysis as forcing data (Geyer, 2014). CMAQ was run with a temporal resolution of one hour over a European domain and an intermediate nested domain over Northern Europe with 64-km and 16-km horizontal resolution, respectively. The chemical boundary conditions for the European domain were taken from FMI APTA global reanalysis (Sofiev et al., 2018). Within the Northern Europe domain, an inner domain over the Baltic Sea region with 4-km horizontal resolution was nested (Fig. 6a), which provided the initial and hourly boundary

conditions for the chemical concentrations of the Hamburg model domain. The hourly meteorological fields for the study domain Hamburg ($30 \times 30\,km^2$) were obtained from the inner domain in a nested simulation with TAPM (Hurley et al., 2005) with a 1-km horizontal resolution (D4 in Fig. 6b).

The meteorological component of TAPM is an incompressible, non-hydrostatic, primitive equation model with a terrain-following vertical sigma coordinate for 3-D simulations. The outer domain (D1 in Fig. 6b) is driven by synoptic-scale me-

teorology of the European Centre for Medium-Range Weather Forecasts (ECMWF) from three-hourly synoptic scale ERA5 reanalysis ensemble means on a longitude/latitude grid at 0.3-degree grid spacing. In addition, wind speed and direction observations at seven measurement stations of the German Weather Service (DWD) are used to nudge the predicted solution towards the observations.

A vegetative canopy, soil scheme, and an urban scheme with seven urban land use classes (Hurley, 2008) are used at the

surface, while radiative fluxes, both at the surface and at upper levels, are also included. In regions belonging to one of the urban land use classes, the specific urban land use characteristics (fraction of urban cover, urban albedo and conductivity, urban anthropogenic heat flux and urban roughness length) are used to calculate the surface temperature and specific humidity as well as surface fluxes and turbulence. A complete list of the meteorological variables and fields used from TAPM as input to EPISODE-CityChem for the AQ simulations is given in the User's Guide for EPISODE-CityChem, included in the CityChem

distribution.

For a better representation of local features, the coarsely resolved standard land cover classes and elevation heights, that are provided together with the TAPM model, were updated with 100 m Corine Land Cover 2012 data (CLC, 2012) and terrain elevation data was adopted from the German Digital Elevation Model (BKG, 2013) on 200-m horizontal resolution.

The procedure to adapt hourly 3-D concentrations of the CMAQ model computed for the 4-km resolution domain as lateral

and vertical boundary conditions is described in Sect. 2.3.1. CMAQ concentrations from the 4-km resolution domain were interpolated to the 1-km resolution (in UTM projection) of the Hamburg study domain, preserving a nesting factor of four (64/16/4/1 km) for the nested model chain. The study domain is located in the southwest part of the 4-km CMAQ domain (CD04); the west border of the study domain is 30 km distant from the CD04 west border and the south border is 21 km distant from the CD04 south border (inset in Fig. 6a). This is considered to be sufficient to avoid that concentrations in the study





domain are affected by domain border effects. The contribution of the re-circulation of $NO_X$ from the coarser outer domain to the budget of $NO_X$ in the inner domain is very small due to the predominant westerly winds.

The background concentrations are adopted for the grid cells directly adjacent to the grid cells of the model domain (with $nx \times ny$ cells per model layer) and also for the vertical model layer that is on top of the highest model layer. Boundary conditions

from CMAQ concentrations are created for the gas-phase chemical compounds: $O_3$, $NO$, $NO_2$, $H_2O_2$, $SO_2$, $HCHO$, $CO$, $N_2O_5$, $HNO_3$, $PAN$ and the individual VOC. Boundary conditions for $PM_{2.5}$ includes primary aerosol components: elemental carbon (EC), primary organic aerosol (POA), sea salt (NaCl), and mineral dust; secondary inorganic aerosol (SIA) components: sulphate ($SO_4^{2-}$), ammonium ($NH_4^+$), nitrate ($NO_3^-$), and SOA ($PM_{2.5}$ was defined including modes I and J of the CMAQ aerosol components). Since the focus of the AQ study is mainly on photochemistry and fine particulate mass, boundary conditions for

$PM_{10}$ were approximated as $[PM_{10}] = [PM_{2.5}] \times 1.5$.

### 4.1.1 Description of the model setup and configuration for Hamburg

The vertical and horizontal structure of the 3-D Eulerian grid of the EPISODE-CityChem is determined by the model domain structure of the TAPM simulation. The model input of boundary conditions and gridded area emissions have to be with the same horizontal resolution as the meteorological fields. A horizontal resolution of 1000 m was chosen for the $30 \times 30 \, km^2$ domain of

Hamburg. The horizontal resolution is in practice limited by the available gridded area source emission data. Finer resolution increases the required computational time; for instance, using a horizontal resolution of 500 m for the study domain results in a four times higher number of grid cells and a halved model time step ($dt = 300 \, s$ instead of $dt = 600 \, s$), increasing the total computational time for one simulation month by a factor of 2.8 compared to the applied resolution. The EPISODE-CityChem model was set up with the vertical dimension and resolution matching that of TAPM, with a layer top at 3750 m height above

ground, avoiding the need for vertical interpolation. The layer top heights of the lowest 10 layers were: 17.5 m, 37.5 m, 62.5 m, 87.5 m, 125 m, 175 m, 225 m, 275 m, 325 m, and 375 m. Table 4 provides details of the vertical and horizontal structure of EPISODE-CityChem and TAPM (pollution grid) D4 as used for the Hamburg study domain and CMAQ (CD04). The computational time for a one-month simulation with EPISODE-CityChem is 10.7 h on an Intel® Xeon (R) CPU E5-2637 v3@3.50 GHz, 64 GB RAM.

The EPISODE-CityChem simulation was performed using the recommended numerical schemes for physics and chemistry, including the new urban parametrisation for vertical eddy diffusivity (urban $K(z)$, see part one; Hamer et al., 2019). The segmented plume model SEGPLU with WMPP based plume rise was used for the point source emissions. The line source model HIWAY-2 with the street canyon option was used for the line source emissions. Table 5 summarizes the chosen model processes and options.

Area, point and line source emissions for the study domain of Hamburg were used from various data sources for the different emission sectors classified by the Selected Nomenclature for sources of Air Pollution (SNAP) of the European Environmental Agency (EEA), applying top-down and bottom-up approaches (Matthias et al., 2018). Table 6 gives an overview. Spatially gridded annual emission totals for area sources with a grid resolution of $1 \times 1 \, km^2$ were provided by the German Federal Environmental Agency (Umweltbundesamt, UBA). The spatial distribution of the reported annual emission totals has been



done at UBA using the ArcGIS based software GRETA ("Gridding Emission Tool for ArcGIS") (Schneider et al., 2016). Hourly area emissions with a 1-km horizontal resolution for SNAP cat. 03 (commercial combustion), 06 (solvent and other product use), 08 (other mobile sources, not including shipping), and 10 (agriculture and farming) were derived from the UBA area emissions by temporal disaggregation using monthly, weekly and hourly profiles.

For SNAP cat. 02 (domestic heating) the daily average ground air temperature obtained from the TAPM simulation is used to create the annual temporal profiles. The day-to-day variation of domestic heating emissions is based on the heating degree day concept Schneider et al. (2016), implemented in the Urban Emission Conversion Tool (UECT) utility (Hamer et al., 2019). Domestic heating emissions (SNAP cat. 02) for Hamburg are distributed between 32 % district heating, 40 % natural gas, 14 % fuel oil and 14 % electricity (Schneider et al., 2016).

NMVOC emissions in the UBA dataset were distributed over individual VOC of the chemical mechanism using the VOC-split of the EMEP model (Simpson et al., 2012) for all SNAP sectors.

A total of 120 points sources were extracted from the PRTR database (PRTR, 2017) and from the registry of emission data for point sources in Hamburg, representing the largest individual stack emissions.

The line source emission dataset (emissions of $NO_X$, $NO_2$ and $PM_{10}$) provided by the city of Hamburg contained 15816 road
links within the study domain. The $NO_X$ emission factor from road traffic for the year 2012 was increased by 20 % for all street types because the average $NO_X$ emission factor in the new HBEFA v3.3 for passenger cars is higher by 19.4 % (diesel cars: 21 %) than in HBEFA v3.1 used in the road emission inventory (UBA, 2010). To estimate NMVOC traffic emissions, an average NMVOC/$NO_X$ ratio of 0.588, derived from UBA data for SNAP cat. 07, was used.

A $NO_2$/$NO_X$ ratio of 0.3 was applied to re-calculate $NO_2$ emissions for this study because of the expected higher real-world
$NO_2$ emissions from diesel vehicles. The applied value is higher than suggested by the reported range (3.2–23.5 vol- %) of the primary $NO_2$ emission fraction from vehicular traffic in London (Carslaw and Beevers, 2005) and the $NO_2$/$NO_X$ ratio of 0.22 for passenger cars in urban areas assumed by Keuken et al. (2012) for the Netherlands. But considerations based on the higher $NO_2$/$NO_X$ ratio from diesel passenger cars (from 0.12 to > 0.5; Carslaw and Rhys-Tyler, 2012) and the review by Grice et al. (2009) who assumed that Euro 4–6 passenger cars emit 55 % of the total $NO_X$ as $NO_2$, justify the use of the high $NO_2$/$NO_X$
ratio for the Hamburg vehicle fleet.

### 4.1.2 Experiment to test the CMAQ nesting versus TAPM

The entrainment of $O_3$ and $PM_{2.5}$ from the regional background into the model domain and their effect on the concentrations inside the domain was studied with a numerical experiment using the model setup as described in Sect. 4.1.1. A constant concentration offset (BCON offset) was added to the hourly CMAQ concentrations at the lateral and vertical boundaries. In
a series of test runs, the BCON offset of $O_3$ was increased from 0 to 60 µg m$^{-3}$ in steps of 10 µg m$^{-3}$ and the BCON offset of $PM_{2.5}$ was increased from 0 to 30 µg m$^{-3}$ in steps of 5 µg m$^{-3}$ between the runs. A linear relationship was found between the monthly mean concentration (July 2012) of $O_3$ and $PM_{2.5}$ in the grid cell, where the inner-city urban background station 13ST is located, and the BCON offset (Fig. S2). Fitting a linear regression model of the form $y = a + bx$ to the data gave a slope of 0.66 and 0.93 for $O_3$ and $PM_{2.5}$, respectively. Since $PM_{2.5}$ is treated as a chemical non-reactive tracer in the model, the



reason for a slope smaller than one is the removal by dry and wet deposition within the study domain. For ozone, the addition of an offset to the concentrations at the boundaries does not fully propagate into the grid cell concentration at station 13ST due to removal by dry deposition, photolysis by sunlight and the chemical reaction with $NO_X$ emitted in the urban area. This initial test demonstrates the high importance of accurate concentrations at the boundaries to represent the regional background for the

study domain.

A second experiment based on the model setup as described in Sect. 4.1.1 was carried out to compare the use of 3-D concentrations (with EPISODE-CityChem) against the use of 2-D concentrations (with TAPM) from the regional model output as boundary conditions for the study domain. In both cases, hourly concentrations computed in CMAQ simulations with 4-km grid resolution for the southern Baltic Sea region were utilized. For the simulation with EPISODE-CityChem the boundary

conditions from hourly 3-D concentrations were taken as described in Sect. 2.3.1. For the simulation with TAPM the boundary conditions from hourly 2-D concentrations were prepared by using the horizontal wind components on each of the four lateral boundaries for weighting the boundary concentrations around the Hamburg study domain. CMAQ concentrations for the TAPM boundary conditions were taken from the seventh vertical model layer, with a mid-layer height of approximately 385 m above ground, as being representative of the vertical profile average concentrations. The same urban emissions were used for the

two models. Results of $O_3$ and $NO_2$ from the two model simulations and from the 4-km resolution CMAQ simulation for July 2012 were extracted for the surface grid cells where urban background stations of Hamburg are located and compared to the observations from the monitoring stations (Fig. S3).

The statistical performance of the three models in terms of correlation coefficient (Corr) and overall bias (Bias) during this period was evaluated by comparison against the measurements at the four stations for $O_3$ (Table 7) and $NO_2$ (Table S5).

Ozone concentrations extracted from the CMAQ 4-km resolution grid are in good agreement with observations in the urban background of Hamburg, with Corr = 0.76 and Bias = -1.5 µg m$^{-3}$ (-3 %), as average of the four stations. For $NO_2$ the agreement is fairly good, with Corr = 0.58 and Bias = -3.56 µg m$^{-3}$ (-26 %), as average of the four stations. At the inner-city station 13ST, observed $NO_2$ concentrations are underestimated by CMAQ (Bias = -7.85 µg m$^{-3}$ (-36 %)) which is probably caused by the coarse resolution of the model and missing road traffic emissions in the area of Hamburg. The regional background $O_3$ from

CMAQ slightly underestimates observed $O_3$ at the urban background in the southwest of the Hamburg study domain in July in particular during episodic high photochemical ozone formation. This can be seen for 52NG, a station influenced by the inflow from the southwest, where $O_3$ modelled by CMAQ has a negative bias (Bias = -7.4 µg m$^{-3}$ (-36 %)).

The performance of EPISODE-CityChem and TAPM for the urban background in the Hamburg study domain is similar for $O_3$ (Corr$_{EPISODE}$ = 0.52; Corr$_{TAPM}$ = 0.50) and $NO_2$ (Corr$_{EPISODE}$ = 0.53; Corr$_{TAPM}$ = 0.50). Modelled $NO_2$ concentrations

from EPISODE-CityChem are generally too low in the urban background, except for the inner-city. For TAPM the negative bias of modelled $NO_2$ is much smaller. However, TAPM overestimates $NO_2$ concentrations at station 27TA by 60 % (Bias = 20.33 µg m$^{-3}$). Station 27TA is frequently in the outflow of the urban area. Inspection of the TAPM simulation revealed that the high $NO_2$ at 27TA is caused by the frequent passage of a pollution plume from port activity and harbour industry in the southwest part of Hamburg. Modelled $O_3$ at station 27TA is too low in the TAPM simulation (Bias = -12.39 µg m$^{-3}$) due to the

titration of ozone in the pollution plume.





PM$_{2.5}$ and PM$_{10}$ were compared at station 13ST (Fig. S4). As shown by the initial experiment, particulate matter from the inflow into the study domain strongly affects the modelled concentrations at the inner-city station. EPISODE-CityChem gives similar daily mean concentrations of PM$_{2.5}$ and PM$_{10}$ as the concentrations extracted from the 4-km resolution CMAQ simulation. Modelled PM$_{2.5}$ and PM$_{10}$ from TAPM is constantly higher by 5–10 µg m$^{-3}$ than PM$_{2.5}$ and PM$_{10}$ from EPISODE-CityChem and closer to the observed concentrations. The procedure for the TAPM boundary conditions as hourly 2-D concentrations from the seventh model layer of CMAQ appears to give realistic PM concentrations at the domain inflow. The procedure for the EPISODE-CityChem boundary conditions is more accurate but inherits the too low PM concentrations from the CMAQ surface layer. In all further model simulations for the evaluation of EPISODE-CityChem for Hamburg, a BCON offset between 5 and 10 µg m$^{-3}$ PM is used to improve the agreement with the observations of the urban background.

### 4.1.3 Experiment to test the new versus old vertical diffusion scheme

Maximum pollutant concentration levels near emission sources at the ground are often associated with strongly stable conditions and low wind speeds; such as during night-time and persistent daytime inversions in winter. The effect of the vertical eddy diffusivity parametrisation on the vertical ozone profile was studied with a numerical experiment using the model setup as described in Sect. 4.1.1. A model run with the standard $K(z)$-method (Byun et al., 1999) and a model run with new urban $K(z)$-method (Hamer et al., 2019) were performed for a seven-day period in January (12–19 Jan. 2012). Modelled ground air concentration and vertical profile of O$_3$ from the two runs were compared for the 1-D column above the model grid cell where station 13ST is located. Fig. 7 shows the time sequence of the vertical O$_3$ concentration profile and the ground concentration changes of O$_3$, NO, NO$_2$ and total hydrocarbons (HC−tot) in this grid cell. On the first and last day of the experiment neutral conditions prevailed, whereas between day 2 and 6 stable conditions with daytime inversions occurred. In the stably stratified nocturnal BL, the vertical exchange of O$_3$ is limited, leading to a depletion in the surface layer due to titration with emitted NO from the traffic sources and partly through dry deposition.

There are clear differences in the vertical O$_3$ profile and the ground air concentration levels of pollutants between the two parametrisations of vertical eddy diffusivity. Vertical ozone profiles for the standard $K(z)$-method show a gradual change of the O$_3$ concentration with height, typically increasing from ∼25 to ∼70 µg m$^{-3}$ between 0 m and 200 m above ground. With the new urban $K(z)$-method the vertical distribution of O$_3$ is rather homogeneous as seen for the last four days of the experiment. The more intense vertical exchange during the night for the new urban $K(z)$-method is also reflected in ground level pollutant concentrations.

The new urban $K(z)$-method reduces the accumulation of NO$_X$ and HC−tot at the ground in the stable nocturnal BL, which was especially pronounced during the night of the third day, when $h_{mix}$ was constant at 100 m for several hours. Daily maximum O$_3$ concentrations in ground-level air are more than doubled with the new method compared to the standard method due to the increased down-mixing of ozone from the nocturnal reservoir layer. Higher daytime O$_3$ concentrations, in addition, lead to higher OH concentrations which in turn cause a faster decay of hydrocarbons.

The new urban $K(z)$-method improves the agreement between modelled O$_3$ and observed O$_3$ for all but the first day. Observed O$_3$ on the first day is overestimated with the new method. This might indicate a problem for neutral stability, however the

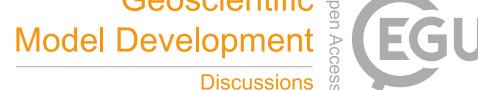

model matches the observed $O_3$ on the neutral conditions of the last day of the experiment. Hence the overestimation of $O_3$ concentrations on the first day is interpreted a consequence of too high ozone in the reservoir layer, coming from the regional background, which was one third higher on the first day when on the following days. The titration of ozone by reaction with NO (giving $NO_2$) at night leads to a decay of modelled $O_3$ during the night which is of the same magnitude as for the observed

$O_3$ concentrations. In sum, the experiment shows that new urban $K(z)$-method prevents too low modelled concentrations of ozone and excessive accumulation of NO at the ground in the stable nocturnal BL.

## 4.2   Presentation and evaluation of model results

A one-year simulation with EPISODE-CityChem was performed for the study domain using the model setup as described in Sect. 4.1.1. Evaluation of the model results for Hamburg was done in a three-stage procedure:

1. Statistical performance analysis of the prognostic meteorological model component of the TAPM model;

2. Statistical performance analysis of the EPISODE-CityChem model by comparison of modelled concentrations against observed concentrations;

3. Model performance analysis with respect to the objectives set forth in the AQD for the use of the model in policy applications.

Statistical indicators of the evaluation included the mean (modelled/observed), standard deviation (STD; modelled/observed), correlation coefficient (Corr), root mean square error (RMSE), overall bias (Bias), normalized mean bias (NMB) and index of agreement (IOA). See Appendix C for the definition of the statistical indicators.

The FAIRMODE (Forum for Air Quality Modelling in Europe) DELTA Tool version 5.6 (Thunis et al., 2012a,b; Thunis et al., 2013; Pernigotti et al., 2013; FAIRMODE, 2014; Monteiro et al., 2018) was used in stages two and three for the evaluation of

model results from air quality simulations for Hamburg. DELTA Tool focuses primarily on the air pollutants regulated in the current AQD (EC, 2008). Delta Tool works with modelled-observed data pairs at surface level, i.e. temporal series of modelled and monitoring data at selected ground level locations, in particular at AQ monitoring stations.

In the first stage of the model evaluation, meteorological data on temperature, relative humidity, total solar radiation, wind speed and wind direction were examined. Hourly based data from the meteorological station at Hamburg Airport (Fuhlsbüttel)

(operated by DWD) and from the 280 m high Hamburg weather mast at Billwerder (operated by Universität Hamburg) was analysed. Observation data from the DWD station at 10 m height and from the Hamburg weather mast at 10 m and 50 m height was used in the analysis. TAPM modelled meteorological data from the $1 \times 1 \, km^2$ grid cell of the D4 domain, where the stations are located, at the corresponding height were extracted for the comparison with observations. Table S6 provides an overview of the statistical analysis of TAPM model data.

Hourly temperature predicted by the TAPM model was in excellent agreement with the observed temperature at both stations and both heights, showing high correlation (Corr $\geq 0.98$) and small overall bias ($\leq 1.00\,°C$). Relative humidity also showed good agreement but with lower correlation (Corr $= 0.74$). Total solar radiation was predicted by TAPM with high correlation





(Corr = 0.86) but the high positive overall bias of $26.56 \, \mathrm{W \, m^{-2}}$. Situations with reduced solar radiation due to high cloud coverage are often not well captured by TAPM. The IOA for temperature, relative humidity and total solar radiation was 0.99 (average of all observations), 0.86 and 0.92, respectively.

TAPM shows good predictive capabilities for wind speed and direction. Due to the assimilation of wind observations at the DWD Hamburg airport station for nudging wind speed and direction in TAPM meteorological runs, the meteorological performance for wind speed and direction will only be compared at the Hamburg weather mast. Modelled hourly data of wind speed at Hamburg weather mast agreed well with the observations throughout the year at 10 m (Corr = 0.87, Bias = -0.08 $\mathrm{m \, s^{-1}}$) and 50 m height (Corr = 0.85, Bias = -0.02 $\mathrm{m \, s^{-1}}$); and was within the observed variability. Southwest and west are the most frequent wind directions in Hamburg due to prevailing Atlantic winds, followed by winds from southeast (Bruemmer et al., 2012). Mean wind direction was computed as circular average (unit vector mean wind direction) for model and observation data. At Hamburg weather mast, modelled and show good agreement with the observations (IOA ≥ 0.89) and observed mean wind direction differed by 16.9° and 6.2° at 10 m and 50 m height, respectively. The difference is due to a slightly higher frequency of winds from west predicted by TAPM.

In the second stage, the statistical performance of EPISODE-CityChem was analysed by comparing modelled concentrations against observed concentrations from the AQ monitoring network of Hamburg. The stations of the monitoring network and the available measurements of pollutant concentrations are listed in Table S7. The monitoring network covers all parts of the city (Fig. 6c). A minimum data availability is required for statistics to be produced at a given AQ monitoring station. In Delta Tool, the requested percentage of available data over a selected time period (here: one year) is 75 % as defined in the AQD. This has been fulfilled by all stations in Hamburg, except for $O_3$ and $PM_{2.5}$ measurements at two stations. The statistical analysis included all stations where the data availability criterion was fulfilled. For the comparison, model output at the exact geographic location of the monitoring stations from the model was used. Concentrations of $NO_2$ and $NO$ were measured at all stations included in this study. The model performance statistics are listed in Table 8 for $O_3$ (based on daily max. of 8-h running mean), in Table 9 for $PM_{10}$ (based on daily mean) and in Table 10 for $NO_2$ (based on hourly values).

The model performs fairly well for the $O_3$ daily maximum of the 8-h running mean, with IOA of 0.63 (average of all stations), correlation of 0.41 (average of all stations) and small NMB (within ±15 %).

The performance of the model for $PM_{10}$ based on daily means is moderate, with IOA of 0.54 (average of all stations). Correlation values are generally low, with station average Corr = 0.34, but RMSE values are fairly small, within the range 9.3–12.2 $\mathrm{\mu g \, m^{-3}}$ for all stations. EPISODE-CityChem was not capable to simulate observed particle peak concentrations from long/short-range transportation events throughout the year. The low correlation on daily basis is related to an apparent overestimation of the weekday-weekend difference by the model.

EPISODE-CityChem performs well for $NO_2$ based on hourly values, with IOA of 0.67 (average of all stations) and correlation of 0.51 (average of all stations). The average performance at the traffic stations (Corr = 0.56, IOA = 0.71 on average) is better than for the other stations. For most urban background stations, the model tends to underestimate the observation mean. The NMB is negative for stations of the urban background in suburban areas (NMB = -25.1 % on average), indicative for too low $NO_X$ emissions from the suburban areas or too efficient dispersion of the local emissions. The weakest temporal correlation



between modelled hourly NO$_2$ and observed values was obtained for urban background station 80KT (Altona-Elbhang; Corr = 0.39).

Annual mean concentrations of regulatory air pollutants from the EPISODE-CityChem model output were compared to the available observation data (Fig. 8). The model reproduces the spatial variation of NO and NO$_2$ concentrations (Fig. 8a, b) and the concentration gradients of NO$_2$ and NO between the urban background (80KT, 51BF, 52NG, 13ST; 61WB, 54BL, 27TA, 74BT), traffic stations (68HB, 64KS, 70MB, 17SM) and industrial stations (21BI, 20VE). For most stations the annual mean observed NO$_2$ is underestimated (bias is within ±25 µg m$^{-3}$). Annual mean SO$_2$ was compared at five stations (Fig. 8d). With the exception of the industrial station 20VE, modelled annual mean SO$_2$ agreed with the observed concentrations within a factor of two. At station 20VE, modelled SO$_2$ is about six times higher than observed SO$_2$. Obviously, the model overestimated the influence of SO$_2$ emissions from nearby industrial sources.

Observed levels of annual mean O$_3$ at the five urban background stations (13ST, 27TA, 51BF, 54BL, 52NG) are captured by the model within ±15 % (Fig. 8c). The summer (JJA) mean of modelled O$_3$ is lower by 26 % than the observed at station 13ST, but in excellent agreement with the observed for the other stations.

Station Sternschanze (13ST) is an inner-city urban background monitoring site. This is the only station, where observations for all regulatory air pollutants considered in this study (NO, NO$_2$, O$_3$, SO$_2$, PM$_{2.5}$ and PM$_{10}$) were available with required data coverage. The underestimation of ozone at 13ST during summer is also obvious from the time series of O$_3$ concentrations (daily maximum of 8-h running mean; see Fig. S5c), accompanied by modelled daily NO concentrations which are frequently higher than observed.

The diurnal cycles of observed median O$_3$ and NO$_2$ concentrations at 13ST for the whole year, summer (JJA) and autumn (SON) show a minimum of O$_3$ in the early morning hours, between 4 and 7 a.m., a daily maximum of O$_3$ between noon and 4 p.m., whereas NO$_2$ peaks two times during the day (6–8 a.m. and at 6–8 p.m.), coinciding with the traffic rush hours (Fig. S6). In summer afternoons, between 2 p.m. and 7 p.m., modelled median O$_3$ is below the bandwidth of observed ozone (25th to 75th percentile) and modelled O$_3$ is up to 30 µg m$^{-3}$ lower than observed concentrations. In autumn, the daily cycle of modelled median NO$_2$ closely follows the observed daily cycle but is ca. 10 µg m$^{-3}$ smaller than the observed median NO$_2$. For the other averaging intervals, modelled median O$_3$ and NO$_2$ concentrations agree with the observations within the bandwidth of 25th to 75th percentile around the observed median.

The EPISODE-CityChem model tends to underestimate observed PM$_{2.5}$ and PM$_{10}$ in the high concentration range. Levels of PM$_{2.5}$ and PM$_{10}$ in the model are controlled by primary emission of particulate matter and their atmospheric dispersion, while secondary aerosol formation is not considered in the model. The comparison of the time series of daily means at station 13ST in Fig. S5 indicates, that the model underestimates observed peak events with high PM$_{2.5}$ and PM$_{10}$ concentrations during the winter season (DJF). These events are likely related to short or long-range transport of anthropogenic emitted PM or secondary produced inorganic PM. Observed levels of annual means of PM$_{2.5}$ and PM$_{10}$ are matched by the model within ±32 % (Fig. 8e, f). The agreement between modelled and observed annual mean PM$_{10}$ at the two traffic stations, 17SM and 68HB, is within ±10 %.





In the third stage of the model evaluation, it is assessed whether the model results have reached a sufficient level of quality for a given policy support application. The model quality objective in the FAIRMODE DELTA Tool has been constructed on the basis of the observation uncertainty and describes the minimum level of quality to be achieved by a model to be fit for policy use (Thunis et al., 2012a,b; Thunis et al., 2013; Pernigotti et al., 2013). The model quality indicator (MQI) is based on the RMSE and provides a general overview of the model performance. The associated model performance criteria (MPC) for correlation, standard deviation and bias can be used to highlight which of the model performance aspects need to be improved. Details on the MQI and MPC are given in Appendix C.

Fig. 9 shows the model performance evaluation of EPISODE-CityChem in terms of fitness for purpose in form of scatter diagram and Target diagram (Thunis et al., 2012a; Monteiro et al., 2018) for $NO_2$ (hourly), $O_3$ (daily max. of the 8-h running mean) and $PM_{10}$ (daily mean). For the yearly averaged values shown in the scatter diagrams, the model quality objective is the bias MPC (as defined by Eq. (C12)). The scatter diagrams related to the bias MPC (Fig. 9a–c) for yearly averaged $NO_2$, $O_3$ and $PM_{10}$ indicate fulfilment (stations lie within green-shaded zone) for all stations.

The EPISODE-CityChem model fulfils the model performance objectives for $NO_2$, $O_3$ and $PM_{10}$ both in terms of bias MPC and MQI (max. MQI, $NO_2$: 0.864, $O_3$: 0.706, $PM_{10}$: 0.834) (Fig. 9d–f). All stations are located within the green-shaded zone, indicating the fulfilment of the RMSE criteria as defined by Eq. (C10), but outside of the dashed circle, i.e. the difference between model and observations is not within the measurement uncertainty range. For all three pollutants, deficits in the model performance are related to the centred RMSE (abscissa of the Target diagram), while bias is small. The error related to correlation dominates the model performance (all stations are in the left quadrant of the Target diagram). The estimated model uncertainty of the predicted hourly $NO_2$ and the predicted daily mean $PM_{10}$ is 34 % and 37 %, respectively.

Fig. 10 depicts spatial maps of the annual mean concentrations of $NO_2$, $O_3$, total NMVOC, $SO_2$, gaseous sulphuric acid ($H_2SO_4$) and $PM_{2.5}$ from the model output of the receptor grid (resolution $100 \times 100\,m^2$). With the exception of $PM_{2.5}$, concentrations of aforementioned compounds are modulated by photochemical reactions in the model simulation. Due to the large temporal, spatial and compositional variations of the input from anthropogenic emissions of $NO_X$, NMVOC and CO within the urban area, the atmospheric chemistry in urban environments is complex. Prevailing winds from the west, on an annual basis, allow for a simplified view of the inflow-outflow pattern for ozone within the study domain of Hamburg. Following the inflow-outflow direction in space from west to east (30 km), modelled $O_3$ concentration starts with ca. $50\,\mu g\,m^{-3}$ at the western border, largely reduces in the inner-city to $15$–$30\,\mu g\,m^{-3}$ and gradually increases again over the eastern part to ca. $40\,\mu g\,m^{-3}$.

Modelled ozone at the outflow border does not reach the level at the inflow border. Within the urban area, the traffic-related emissions of NO destroy much of the $O_3$ (mainly at night when $O_3$ is not recycled through photolysis of $NO_2$), clearly seen as minimum concentrations along the traffic network (Fig. 10b). Thus, the inner urban area provides an efficient sink for ozone, which qualitatively is in accord with findings of the REPARTEE (Regents Park and Tower Environmental Experiment) measurement campaign carried out in London in the autumn of 2006 and 2007 (Harrison et al., 2012).

Photochemical production of $O_3$ from $NO_X$, NMVOC and CO, emitted in the urban area, is very limited in the inner-city. Main sources of NMVOC in Hamburg are solvent use (SNAP cat. 06) and traffic emissions. NMVOC annual mean concentrations of more than $40\,\mu g\,m^{-3}$ were modelled close to roads in the inner-city (Fig. 10c). Ehlers et al. (2016) report the similarity

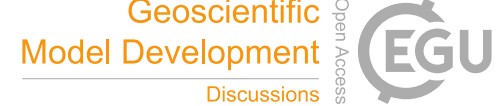



of NMVOC fingerprints in air samples taken in the inner-city of Munich and in a road tunnel in Berlin with the fingerprint of petrol cars under cold-start conditions. Our model simulation finds that the loss of NMVOC by OH-initiated oxidation in the model is inhibited due to the low $O_3$ concentrations in the inner-city. In summer, modelled OH midday maximum concentrations are in the range $(0.5\text{–}2.0) \times 10^6$ molecule cm$^{-3}$ in the inner-city. Modelled o-xylene, which is the model surrogate

compound for the sum of aromatic VOC is 5–10 µg m$^{-3}$ (1.2–2.2 ppbv), in some distance from the roads. This is in the same range as concentrations measured in central London (ca. 2 ppbv for the sum of aromatics; Valach et al., 2015).

SO$_2$ is an important precursor for secondary aerosol formation. SO$_2$ emissions in Hamburg are mainly from industrial point sources and ship traffic in the harbour area. Highest yearly averaged modelled SO$_2$ concentrations are in the range of 20–40 µg m$^{-3}$; in the proximity of the main sources (Fig. 10d). In the atmosphere, SO$_2$ reacts with the OH radical and with CH$_3$O$_2$

to give gaseous sulphuric acid (H$_2$SO$_4$). The presence of sulphuric acid in gaseous concentrations of $10^6 - 10^7$ molecule cm$^{-3}$ is necessary in order to observe new particle formation events in the atmosphere (Zhang et al., 2012). In the model, a constant very low BCON value ($10^{-5}$ µg m$^{-3}$) was chosen for H$_2$SO$_4$, leading to a reduced sulphuric acid concentration in the boundary cells (Fig. 10e). Towards the inner domain, H$_2$SO$_4$ quickly increases due to the oxidation of SO$_2$ advected to Hamburg from the regional background. Modelled annual mean H$_2$SO$_4$ peaks in the harbour area with up to 0.018 µg m$^{-3}$. On spatial average,

H$_2$SO$_4$ annual mean concentration is 0.013 µg m$^{-3}$, corresponding to $8.2 \times 10^7$ molecule cm$^{-3}$, higher than typical ambient concentrations in the urban atmosphere. For comparison, reported maximum midday H$_2$SO$_4$ concentrations in Beijing are in the range $(0.3\text{–}1.1) \times 10^7$ molecule cm$^{-3}$ (Wang et al., 2013). Too high modelled H$_2$SO$_4$ is explained by the fact that condensation of sulphuric acid on pre-existing particles is not accounted for in the model. Condensation is the most important atmospheric sink of sulphuric acid and also leads to the formation of particulate sulphate.

Previous studies have shown that levels of PM$_{2.5}$ in the urban background are to a large extent controlled by the atmospheric transport from up-wind regions. Modelled PM$_{2.5}$ levels in the urban background of Hamburg are on annual average of 2.5 µg m$^{-3}$ higher than the regional background. The urban increment due to road traffic exceeds the urban background by 10–20 µg m$^{-3}$ in the model simulation (Fig. 10f). Hotspots of PM$_{2.5}$ pollution are found within the harbour area to the southwest of the inner city, where several refineries and industry are located. These industrial emissions are represented as point source

plumes in the model.

Modelled and observed annual mean NO$_2$ at the four traffic stations exceed the annual limit value of 40 µg m$^{-3}$ in 2012 (cf. Fig. 8b). The model simulation suggests that there is wide-spread exceedance of this limit value in the city, mainly at the main streets and the road junctions of the inner city and along the motorways outside the inner city (Fig. 10a).

A 6-day episode with high ozone in July 2012 (5–10 July) was analysed. Fig. 11 shows the O$_3$, NO, NO$_2$ and OH concen-

trations, together with wind direction and total solar radiation during the episode as a function of time at inner-city station 13ST. Six short time periods P1–P6 with O$_3$ underestimation by the model at station 13ST were identified (grey-shaded area). The modelled OH is controlled by the diurnal variation of total solar radiation with the highest OH concentration close to local solar noon in the range $(0.5\text{–}5) \times 10^6$ molecule cm$^{-3}$ at 13ST and station 27TA in the eastern part of Hamburg (Fig. S7), and $(5\text{–}16) \times 10^6$ molecule cm$^{-3}$ at station 54BL close to western border (Fig. S8). Total solar radiation predicted by the model





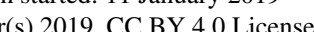

agreed fairly well with the observations during the ozone episode, except during P5 where the observed daily maximum was underestimated by ~40 %.

During P1, wind direction observed at Hamburg weather mast changed from southeast to northwest. After the end of the first day, the wind was again from east to southeast, which was not reproduced by the model. After P3, predicted wind flow
was from westerly directions (190°–250°) for the remainder of the episode, in agreement with observations. A sudden drop of observed $O_3$ during the night of the second day can be noted at 54BL (Fig. S8), obviously related with shallow mixing height of the nocturnal BL (modelled $h_{mix} < 100$ m) and high observed and modelled $NO_2$ due to accumulation of $NO_X$ emissions in the surface layer. The model was able to reproduce the low observed $O_3$ ($< 10 \, \mu g \, m^{-3}$) during this night.

P1 and P6 are the periods with the most severe underprediction of $O_3$, with modelled $O_3$ below $10 \, \mu g \, m^{-3}$. Modelled $O_3$
drops by ~$40 \, \mu g \, m^{-3}$ compared to the 6-h average before the two periods. During both periods, EPISODE-CityChem predicted high NO and $NO_2$ concentrations resulting in the efficient titration of $O_3$ by reaction with NO in the late afternoon and at night. Modelled $NO_X$ concentrations during P1 and P6 are several times higher than observed $NO_X$ concentrations. Corresponding modelled $NO_X$ peaks are not present at 54BL and 27TA, implying that the high $NO_X$ steams from emissions within the urban area rather than from the advection of a $NO_X$ plume through the domain boundaries. The same seems to be the case for the
night time period P4 although modelled NO agrees with observed NO and the titration of $O_3$ in the model was weaker, leading to a smaller reduction of modelled $O_3$ (by ~$25 \, \mu g \, m^{-3}$) compared to 6-h average before P4, than in the periods P1 and P6.

During the three daylight periods P2, P3 and P5 the observed daily $O_3$ maximum is underestimated by 40–50 $\mu g \, m^{-3}$ by the model. Too low modelled total solar radiation could be the reason for the too low $O_3$ production during P5, but for the two other periods observed total solar radiation was reproduced by the model. During P2, predicted wind direction was from
the northwest (270°–300°), in contrast to observed wind direction at the Hamburg weather mast (90°–110°). While at BL54, close to the western border, predicted $O_3$ agreed with the observed $O_3$ (Fig. S8), the air mass with ozone was not correctly advected to the inner city leading to the underestimation of $O_3$ at 13ST. The clear-sky period P3 starts at about 6 a.m. with a rapid increase of observed $O_3$ during the morning. Modelled $h_{mix}$ is still below 200 m during the first hours of P3, causing an accumulation of $NO_X$ emissions of the rush hour in the model, leading to higher $NO_X$ concentrations than observed. It cannot
be excluded, that missing emissions of biogenic VOC (BVOC) from trees in urban green parks and at roadside contributed to the apparent underestimation of observed ozone during the daylight periods.

## 5  Planned improvements to the EPISODE model

The future development of the EPISODE model with respect to photochemistry and the dispersion near sources is outlined in the following. Specifically, the implementation of the photochemistry in relation to sub-grid modelling and the treatment
of aerosol formation on the urban scale will be in the focus of the planned development for the next versions of EPISODE-CityChem.

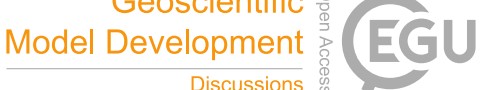

## 5.1 Photolysis parameterisation development

The procedure for calculating photo-dissociation rates (*j*-values) has not been changed since the original development of the EMEP45 mechanism and is thus as documented by Walker et al. (2003). This procedure was based on the procedure used in the EMEP oxidant model at that time and needs to be revised and updated. The plan is to update the methodology in accordance with the present version of the EMEP unified model (Simpson et al., 2012). In the present EMEP model, the *j*-values are based on pre-calculated rates from a detailed radiative-transfer model (PHODIS, Kylling et al., 1998) and interpolated between certain fixed cloud fractions.

## 5.2 EP10-Plume development

The PSS might apply in street canyons because there, the distance between source and receptor is short, hence only the fastest chemical reactions can have a significant influence on the photochemical transformation in the street canyon air. On the time scales applying to a street canyon, CO and hydrocarbons can be treated as inert tracers. However, the photochemical scheme applied in the sub-grid component also needs to consider situations with a larger distance between source and receptor (within a cell of the Eulerian grid).

The photochemical steady-state relationship is usually not valid in the urban air because organic peroxy radicals compete with $O_3$ to convert NO to $NO_2$ as a result of the oxidation of VOCs in sunlight. EP10-Plume considers only the photochemical degradation of HCHO, but not the oxidation of other reactive VOCs emitted from traffic. It is planned to develop a more detailed chemistry scheme, including VOC, at the receptor points.

Background ozone concentrations are taken into account in the photochemical transformation in the sub-grid component (in both PSS and EP10-Plume). Due to the low ventilation in street canyons, the residence time of pollutants becomes comparable to the time scales of the reactions involved in the PSS assumption, i.e. on the order of tens of seconds. This implies that the exchange rate between the plume from the line source and the background air can become a limiting factor for the photochemical transformation in street canyons. The exchange rate is governed by the residence time of the pollutants at the street. However, in the sub-grid line source model, it is assumed that background $O_3$ is instantaneously mixed with NO from the plume of the line source. The rate of reaction (R5) might, therefore, be overestimated in the sub-grid model, depending on the ambient conditions for the plume mixing. Hertel and Berkowicz (1989) take the exchange rate into account for the PSS and suggest that the residence time of pollutants in a street canyon can be approximated by $H_{sc}/\sigma_{wt}$, where $\sigma_{wt}$ is the ventilation velocity. The exchange rate of ozone will be considered in the refinement of the photochemistry of the sub-grid component.

## 5.3 EmChem09-mod developments

### 5.3.1 Emissions of biogenic VOC

Biogenic emissions of VOCs from trees might be relevant in the VOC-limited regime. Many urban trees such as European aspen (Populus) and deciduous oaks emit large amounts of isoprene (Karl et al., 2009). The ozone formation potential of





BVOC emitted from urban trees is sufficiently high to outperform the ozone uptake capacity of the trees (Grote et al., 2016). Missing emissions of BVOCs from trees in urban green parks and at roadside might partly explain the underestimation of observed $O_3$ in summer in the inner-city by the model. Isoprene, in the presence of a sufficiently high level of $NO_X$, can contribute substantially to the $O_3$ formation in the urban atmosphere. Further, monoterpenes are considered to be a relevant

source of organic peroxy radicals at night (Platt et al., 2002). EmChem09-mod includes the chemical reactions for the OH-initiated oxidation of isoprene and for two types of monoterpenes, represented by $\alpha$-pinene and limonene. It is planned to implement a module for the computation of inline emissions of isoprene and monoterpenes as a function of temperature and solar radiation using data on tree-specific BVOC emission factors by Karl et al. (2009) and high-resolution urban land use information, for example from the Copernicus Urban Atlas.

### 5.3.2 Secondary formation of particulate matter

Currently, both $PM_{2.5}$ and $PM_{10}$ are treated as inert tracers with no secondary aerosol formation. The first step towards a better representation of the particulate phase will be the separation of particulate matter into individual particulate chemical compounds. Traffic emissions of fine ($PM_{2.5}$) and coarse ($PM_{10}$ - $PM_{2.5}$) particulate matter will be separated into (non-volatile) POA and EC, thereby assuming that any emission of SVOCs and intermediate-volatility volatile organic compounds (IVOC)

immediately undergo irreversible condensation to the exhaust particulate matter. A clear advantage of the approach is that it avoids the need for discounting SVOC and IVOC from the city's particulate matter emission inventory. Mineral dust and sea salt that is imported to the city from the regional background needs to be treated as separate compounds. The main focus is on formation of secondary inorganic aerosol (SIA). Thus, the thermodynamic equilibrium solver MARS (Binkowski and Shankar, 1995) could be used for the calculation of the partitioning between gas and fine-mode particles (Simpson et al., 2012). However,

MARS does not account for mineral dust components and sea salt; the latter could be relevant for the formation of SIA in coastal cities because it increases the water associated with the particulate matter.

### 5.3.3 Ultrafine particles

Among the emerging, yet unregulated pollutants, in cities are ultrafine particles (UFP, diameter less than 100 nm). Major sources of UFPs in urban environments are motor vehicle exhaust emissions (e.g. Harrison et al., 2011). Emission from ships

contribute to UFP pollution in harbour cities (Pirjola et al., 2014). High concentrations of ultrafine particles are formed during new particle formation events (Salma et al., 2016). Studies evidenced the relevance of episodes of new particle formation in urban environments for cities situated in high insolation regions such as southern Europe (e.g. Pey et al., 2008; Dall'Osto et al., 2013; Brines et al., 2015). UFPs are usually evaluated in terms of particle number (PN) concentrations. In urban environments, UFPs dominate the total PN concentrations, but only make a small contribution to particulate matter. A simplified parame-

terization for the treatment of dry deposition and coagulation of particles (Karl et al., 2016) has already been implemented in EPISODE for modelling PN concentrations in Oslo (Kukkonen et al., 2016). The MAFOR aerosol dynamics solver (Karl et al., 2011; Karl et al., 2016) will be implemented in EPISODE-CityChem to compute the information on the size distribution of

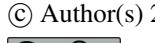


UFPs and the total PN. The solver includes nucleation, coagulation, as well as growth due to condensation of sulphuric acid and low volatile/semi-volatile organic vapours from biogenic and anthropogenic sources.

## 5.4  SSCM development

In addition to the dependence from the wind direction (considered through $\theta_{\mathrm{street}}$, the angle between wind direction and road axis), the direct contribution in SSCM is mainly sensitive to the emission intensity of the line source, the street level wind speed, and the integration path. The integration path corresponds to the length of the recirculation zone but extends to $L_{\mathrm{max}}$ for close to parallel wind directions. The length of the recirculation zone ($L_{\mathrm{rec}}$) depends on the building height along the canyon; while $L_{\mathrm{max}}$ is a function of the canyon width for large $\theta_{\mathrm{street}}$ and a function of the canyon length if $\theta_{\mathrm{street}}$ is below 45°. Both dimensions ($L_{\mathrm{rec}}$ and $L_{\mathrm{max}}$) are only roughly estimated in SSCM because it considers only generic street canyon types and not the site-specific street canyon geometry. It is planned to refine SSCM with respect to a better representation of the urban street canyon geometry. For example, by using spatially resolved information of building height and street canyon width by extracting the data from 3-D city building models for each road segment.

## 5.5  WMPP development

Envisaged future improvements to the WMPP will be to extend it to use energy budget methods (Thom et al., 1975; van Ulden and Holtslag, 1985; Fritschen and Simpson, 1989; Tunick, 2006) in combination with net surface radiation and heat flux to determine wind and turbulence profiles and mixing height. The method will be based on data such as solar radiation, cloud cover, air temperature, wind speed, relative humidity, precipitation and surface conditions.

## 6  Conclusions

The CityChem extension of the urban AQ model EPISODE (Slørdal et al., 2003, 2008; Hamer et al., 2019) offers a detailed treatment of the atmospheric chemistry in urban areas and a more advanced treatment for the dispersion close to point emission sources, such as industrial stacks, and line emissions sources, such as open roads and street canyons. The EPISODE-CityChem model, which is based on the core of the EPISODE model, integrates the CityChem extension into an urban CTM system, with the capability of simulating photochemistry and dispersion of multiple pollutants on urban scales.

EPISODE consists of a 3-D Eulerian grid CTM model with embedded Gaussian dispersion models that track the sub-scale dispersion of pollutants from line and point emission sources until the fine-scale variability becomes unimportant. The integration of WMPP in the sub-grid point source model for the dispersion around an elevated point source increases the maximum ground concentration of an inert tracer by a factor of four to six in neutral and unstable conditions compared to the standard parameterization in EPISODE. The integration of SSCM in the sub-grid line source model results in higher concentrations in street canyons because it considers the reduced ventilation inside the canyon and the recirculation of the traffic plume.





The EPISODE-CityChem model takes into account that long-range transport contributes to urban pollutant levels by using hourly varying pollutant concentrations at the lateral and vertical boundaries from the CMAQ model (Byun and Schere, 2006) as initial and boundary concentrations. The model reads meteorological fields generated by the prognostic meteorology component of TAPM (Hurley, 2008; Hurley et al., 2005) but can also use meteorological fields constrained by observations.

The performance of EPISODE-CityChem was evaluated with a series of tests to study the basic functionalities of the CityChem extension and with a first application to the air quality situation in the city of Hamburg, Germany. The performance analysis with the FAIRMODE DELTA Tool for the air quality in Hamburg showed that the model fulfils the model performance objectives for $NO_2$ (hourly), $O_3$ (daily max. of the 8-h running mean) and $PM_{10}$ (daily mean) set forth in the AQD, qualifying the model for use in policy applications.

Observed levels of annual mean $O_3$ at the five urban background stations in Hamburg are captured by the model within $\pm 15\%$. At the inner-city urban background station the summer mean of modelled $O_3$ is 25 % lower than observed. Analysis of hourly $O_3$ concentrations during a summer ozone episode revealed that too low modelled $O_3$ in the late afternoon and at night is related to the efficient titration of $O_3$ by reaction with NO, which accumulated in the shallow nocturnal BL. Further investigation of the occasionally high abundance of $NO_X$ at night would require sensitivity analysis of the various source
categories contributing to the $NO_X$ levels in the inner city, which remains as a task for future studies.

BVOC emission from urban parks and forests might partly explain the underestimation of observed ozone in summer. There is evidence, that the contribution of BVOC emissions to ozone formation can be up to 60 % to ozone levels during heat waves in densely populated areas (Churkina et al., 2017), depending on type and amount of urban vegetation. In the future, BVOC emissions in urban areas are expected to become even more important in ozone formation if anthropogenic NMVOC emissions
continue to decline as a result of technological progress (Wagner and Kuttler, 2014). The implementation of BVOC emissions as function of temperature and solar radiation will be in the focus of coming developments of the CityChem extension.

Envisaged applications of the EPISODE-CityChem model are urban air quality studies, emission control scenarios in relation to the traffic bans introduced in German cities and the source attribution of sector-specific emissions to observed levels of air pollutants. The model can also be utilized in the evaluation of air pollution exposure and in the assessment of adverse health
impacts. Features of the model that facilitate its application to urban AQ in cities worldwide include the integrated utilities for input preparation and output processing, moderate computational demand, photochemistry options (ozone formation studies), high spatial and temporal resolution and the demonstrated fitness for policy use.

*Code and data availability.* The source codes of the EPISODE-CityChem model version 1.2 and the pre-processing utilities are accessible in release under the RPL license at http://doi.org/10.5281/zenodo.2158225 (Karl and Ramacher, 2018).
A tar package with example data for a one month simulation and the User's Guide are included in the release. All pre-processing tools are written in Fortran 90. Software requirements for the utilities and the EPISODE-CityChem model are installation of the gcc/gfortran fortran90 compiler (version 4.4. or later) and the netCDF library (version 3.6.0 or later).

The following data sets are available for download from the HZG ftp server upon request:





- input data for the one-year AQ simulation of Hamburg (full set ca. 50 GB);

- DELTA Tool data for comparison of model output and measurements;

- model output data of the AQ simulation of Hamburg (full set ca. 50 GB).

## Appendix A: Photodissociation rates

The photo-dissociation coefficients of photolysis reactions are calculated according to the expression:

$$
j_n = \begin{cases}
\text{CLF}_n \; \varepsilon_{1,n} \exp\left(\varepsilon_{2,n}/\cos\left(\theta_z\right)\right) & \theta_z < 60° \\
\text{CLF}_n \; \varepsilon_{1,n} \exp\left(\varepsilon_{2,n} \, \alpha_0\left(\theta_z\right)\right) & 60° \le \theta_z < 89° \\
\text{CLF}_n \; \varepsilon_{1,n} \exp\left(\varepsilon_{2,n} \, \alpha_0\left(89°\right)\right) & \theta_z \ge 89° \,,
\end{cases}
\tag{A1}
$$

where $\theta_z$ is the zenith angle, $\alpha_0$ denotes the optical air mass for large zenith angles, and $\text{CLF}_n$ is the cloud correction factor for reaction number $n$:

$$
\text{CLF}_n = \begin{cases}
\left(1.0 - \text{CL}/0.2\right) + \varepsilon_{3,n} \; \text{CL}/0.2 & \text{CL} \le 0.2 \\
\varepsilon_{3,n} + \left(\text{CL} - 0.2\right) \left(\varepsilon_{4,n} - \varepsilon_{3,n}\right)/0.6 & \text{CL} > 0.2 \,.
\end{cases}
\tag{A2}
$$

The actual fractional cloud cover of low clouds (0.0 to 1.0), CL, is either based on observational data of cloud coverage or from the total solar radiation field (calculated by TAPM) using the approximation for the transmission coefficient of short wave radiation suggested by Burridge and Gadd, as given in Stull (1988). Empirical values for $\varepsilon_1$, $\varepsilon_2$, $\varepsilon_3$ and $\varepsilon_4$ for the photolysis reactions are tabulated in Table S2.

## Appendix B: Treatment of deposition on the Eulerian grid

**Dry deposition**

The dry deposition of gases and aerosols is treated based on the resistance analogy, where the inverse deposition velocity of gases is the sum of three resistances in series, the aerodynamic resistance $R_a$ (m s$^{-1}$), the quasi-laminar layer resistance, $R_b$ (m s$^{-1}$), and the surface (canopy) resistance $R_c$ (m s$^{-1}$). Gravitational settling of coarse particles is considered for the dry deposition of aerosols. The loss rate of a gaseous species $i$ to the land or water surface, within a volume of unit area and height

$\Delta z$ (here the thickness of the lowermost layer), is given by the product of the deposition velocity $V_{\text{dry}}$ (m s$^{-1}$) at the reference height $z_{\text{ref}}$ (here the mid-point height of the lowermost model layer) and the concentration ($C_i$) at that height:

$$
\frac{\Delta C_i(z_{\text{ref}})}{dt} = -V_{\text{dry}} \, C_i(z_{\text{ref}})/\Delta z \,,
\tag{B1}
$$

The dry deposition velocity of gases, $V_{\text{dry,g}}$, is calculated as (Simpson et al., 2003):

$$
V_{\text{dry,g}} = \frac{1}{R_a + R_b + Rc} \,,
\tag{B2}
$$




The aerodynamic resistance at $z_{\mathrm{ref}}$ is calculated based on surface layer similarity theory as function of the Monin-Obukhov length and the friction velocity:

$$R_a = \frac{1}{\kappa u_*} \left( \ln \frac{z_{\mathrm{ref}}}{z_0} - \Psi_{\mathrm{H}} \left( \frac{z_{\mathrm{ref}}}{L} \right) + \Psi_{\mathrm{H}} \left( \frac{z_0}{L} \right) \right), \tag{B3}$$

where $\Psi_H$ is the influence function for heat transfer, $z_0$ is the surface roughness (for momentum) and $L$ is the Monin-Obukhov

length. The quasi-laminar layer resistance is calculated according to the parameterisation given by Simpson et al. (2003). The canopy resistance, i.e. deposition due to capture of pollutants by the surface, is currently only considered by a minimum value, i.e. $R_c = R_{c,\mathrm{min}} = 1\,\mathrm{m\,s^{-1}}$. The parameterisation of the canopy resistance is complex, since it depends both on surface characteristics and the chemical characteristics of the depositing gas.

The dry deposition velocity of particles, $V_{\mathrm{dry,p}}$, is calculated as (Simpson et al., 2003):

$$V_{\mathrm{dry,p}} = \frac{1}{R_a + R_b + R_a R_b v_s} + v_s, \tag{B4}$$

where $v_s$ is the gravitational settling velocity and the other terms are as for gases.

Equation (B4) involves the assumption that all deposited particles stick to the surface, so that the surface resistance becomes zero. The dry deposition velocity of atmospheric aerosols depends on their sizes. The current formulation distinguishes between $PM_{2.5}$ and $PM_{10}$, which are presently assigned the particle diameters of $0.3\,\mathrm{\mu m}$ and $4\,\mathrm{\mu m}$. All the resistances are integrated over

the aerosol sizes, assuming a log-normal particle size distribution with the geometric standard deviations of $2.0\,\mathrm{\mu m}$ and $2.2\,\mathrm{\mu m}$ for $PM_{2.5}$ and $PM_{10}$ respectively.

**Wet deposition**

Wet deposition is described as a sink term within the advection/-diffusion equation and can be parameterized by $dC_i/dt = -\Lambda \cdot C_i$; where $C_i$ is the grid concentration of a gaseous or particulate species and $\Lambda$ is the scavenging coefficient ($\mathrm{s^{-1}}$). Wet scav-

enging is different from zero in grid cells where precipitation (rainfall or snowfall) occurs. The chosen crude approach for representing wet deposition treats in-cloud scavenging in the same way as below-cloud scavenging. Further, the cloud base is assumed to be at the model top, which means that scavenging occurs throughout the entire 1-D model column for which the precipitation rate in the surface grid cell is greater than zero. For the short-term estimation of near-ground concentrations in urban areas, below-cloud scavenging is expected to be the dominant wet removal process. A more accurate treatment of the

below-cloud scavenging requires knowledge of the cloud base height (which is not standard output of the TAPM model) in order to limit wet deposition to the model layers that are actually affected by raining clouds and to separate between in-cloud and below-cloud scavenging. The scavenging of gases is calculated as (Simpson et al., 2003):

$$\Delta C_{i,\mathrm{wet}} = -C_i \frac{W_{\mathrm{sub}} P_r}{H_{\mathrm{sub}} + \rho_w}, \tag{B5}$$

where $W_{\mathrm{sub}}$ is the sub-cloud scavenging coefficient for gases, supplied as a constant value by the model user, $P_r$ ($\mathrm{kg\,m^2\,s^{-1}}$) is

the precipitation rate, $H_{\mathrm{sub}}$ is the scavenging depth (corresponding to the total vertical depth of the model) and $\rho_w$ is the water density ($1000\,\mathrm{kg\,m^{-3}}$).

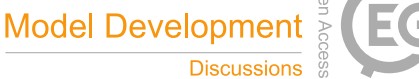



Precipitation is a 2-D surface field, either from observations of precipitation rate or computed by TAPM. The wet deposition rate of particulate compounds is calculated as (Simpson et al., 2003):

$$\Delta C_{i,\text{wet}} = -C_i \frac{A P_r}{V_{\text{dr}}} \, \overline{E} \,, \tag{B6}$$

where $V_{\text{dr}}$ is the raindrop fall speed ($V_{\text{dr}} = 5 \, \text{m s}^{-1}$), $A = 5 \, \text{m}^3 \, \text{kg}^{-1} \, \text{s}^{-1}$ is an empirical coefficient when the Marshall-Palmer size
distribution is assumed for rain drops, and $\overline{E}$ is the tabulated size-dependent collection efficiency of aerosols by the rain drops.

## Appendix C: Statistical indicators and model performance indicators

The statistical analysis of the model performance the following statistical indicators are used: overall bias (Bias), normalized mean bias (NMB), standard deviation (STD), root mean square error (RMSE), correlation coefficient (Corr) and index of agreement (IOA).

The overall bias captures the average deviations between the model and observed data and is defined as follows:

$$\text{Bias} = \overline{M} - \overline{O} \,, \tag{C1}$$

where $M$ and $O$ stand for the model and observation results, respectively. The overbars indicate the time average over $N$ time intervals (number of observations).

The normalized mean bias is given by:

$$\text{NMB} = \frac{\text{Bias}}{\overline{O}} = \frac{\overline{M} - \overline{O}}{\overline{O}} \,. \tag{C2}$$

The root mean square error combines the magnitudes of the errors in predictions for various times into a single measure and is defined as:

$$\text{RMSE} = \sqrt{\frac{1}{N} \sum_{i=1}^{N} (M_i - O_i)^2} \,, \tag{C3}$$

where subscript $i$ indicates the time step (time of observation values). RMSE is a measure of accuracy, to compare prediction
errors of different models for a particular data and not between datasets, as it is scale-dependent.

The correlation coefficient (Pearson $r$) for the temporal correlation is defined as:

$$\text{Corr} = r = \frac{\frac{1}{N} \sum_{i=1}^{N} \left( M_i - \overline{M} \right) \left( O_i - \overline{O} \right)}{\text{STD}_M \, \text{STD}_O} \,. \tag{C4}$$

$\text{STD}_M$ and $\text{STD}_O$ are the standard deviation of model and observation data, respectively. The standard deviations are:

$$\text{STD}_M = \sqrt{\frac{1}{N-1} \sum_{i=1}^{N} \left( M_i - \overline{M} \right)^2} \quad \text{and}$$

$$\text{STD}_O = \sqrt{\frac{1}{N-1} \sum_{i=1}^{N} \left( O_i - \overline{O} \right)^2} \,. \tag{C5}$$





The index of agreement is defined as:

$$\text{IOA} = 1 - \frac{\sum_{i=1}^{N} (M_i - O_i)^2}{\sum_{i=1}^{N} \left( |M_i - \overline{M}| + |O_i - \overline{O}| \right)^2} . \tag{C6}$$

An IOA value close to 1 indicates agreement between modelled and observed data. The denominator in Eq. (C6) is referred to as the potential error.

The model performance criteria (MPC) for dispersion models is the minimum level of quality that has to be achieved for use in policy support related to AQ regulations. The MPC implemented in the FAIRMODE Delta Tool have been constructed on the basis of the observation uncertainty (Thunis et al., 2012a).

The uncertainty of a single observation value $U_{95}(O_i)$ is expressed as:

$$U_{95}(O_i) = k\, u_r^{\text{RV}} \sqrt{(1 - \alpha^2)\, O_i^2 + \alpha^2\, (\text{RV})^2} , \tag{C7}$$

where $u_r^{\text{RV}}$ represents the relative measurement uncertainty estimated around a reference value, RV, for a given time averaging, e.g. the hourly or daily limit values of the Air Quality Directive (AQD). The fraction of uncertainty around the RV is given by $\alpha^2$. Most commonly, the expanded uncertainty is scaled by using a value of 2 for the coverage factor, k, to achieve a level of confidence of approximately 95 percent.

The root mean square of the observation uncertainty ($\text{RMS}_U$) is then:

$$\text{RMS}_U = \sqrt{\frac{1}{N} \sum_{i=1}^{N} (U_{95}(O_i))^2} . \tag{C8}$$

A model quality indicator (MQI) is defined as the ratio between the model-observation bias and a quantity proportional to the observation uncertainty as:

$$\text{MQI} = \frac{|O_i - M_i|}{\beta\, U_{95}(O_i)} , \tag{C9}$$

with $\beta = 2$ in the DELTA Tool.

Using Eq. (C8), the MQI can be generalized to a time series by:

$$\text{MQI} = \frac{\text{RMSE}}{\beta\, \text{RMS}_U} \leq 1 . \tag{C10}$$

The model quality objective (MQO) is fulfilled when the MQI is less or equal 1.

A characteristic of the MQI is that errors in Bias, STDM and Corr are condensed into a single indicator value, as follows:

$$\text{MQI}^2 = \frac{\text{Bias}^2}{(\beta\, \text{RMS}_U)^2} + \frac{(\text{STD}_M - \text{STD}_O)^2}{(\beta\, \text{RMS}_U)^2} + \frac{2\, \text{STD}_O - \text{STD}_M\, (1 - \text{Corr})}{(\beta\, \text{RMS}_U)^2} . \tag{C11}$$

From Eq. (C11), the model performance criterion (MPC) for the error of bias, standard deviation and correlation can be derived. The bias MPC is derived from Eq. (C11) assuming Corr = 1 and $\text{STD}_M = \text{STD}_O$, as follows:

$$\text{MPC(bias)} = \frac{\text{Bias}^2}{(\beta\, \text{RMS}_U)^2} \leq 1 . \tag{C12}$$





The MQI as described by Eq. (C10) is used as main indicator in the Target diagram (Thunis et al., 2012a). In the normalised Target diagram, it represents the distance between the origin and a given station point. The normalised bias (first term on the right hand side of Eq. (C11) is used for the y-axis while the centred root mean square error (CRMSE) (sum of the two last terms on the right hand side of Eq. (C11)) is used to define the x-axis. More details on the normalised Target diagram can be found in Thunis et al. (2012a).

*Author contributions.*

**Matthias Karl:** development of research questions; main development of the CityChem extension; EPISODE-CityChem simulations; evaluation of the air concentrations data; framework for data processing; visualisation and plotting; major writing tasks

**Sam-Erik Walker:** overall structure; development of EMEP45 and WMPP, implementation of the TWOSTEP solver, text contributions

**Sverre Solberg:** co-development of EMEP45; text contributions

**Martin O. P. Ramacher:** preparation of the input data sets for the study of Hamburg; TAPM simulations to produce air concentration data and meteorological input data; evaluation of TAPM meteorological data; test of early EPISODE-CityChem builds; setup of DELTA Tool

*Competing interests.* The authors declare that they have no conflict of interest

*Acknowledgements.* This work was partially funded by the ERA-PLANET trans-national project SMURBS (SMart URBan Solutions for air quality, disasters and city growth), Grant Agreement n. 689443, under the EU Horizon 2020 Framework Programme. We thank the city of Hamburg, Ministry of Environment and Energy (Behörde für Umwelt Energie Hamburg, BUE) for providing AQ monitoring data and the Agency for Immission Control and Plant Installations (Amt für Immissionsschutz und Betriebe, IB) for providing traffic emission data and industrial emission totals. We kindly acknowledge Ingo Lange and the Meteorological Institute of Universität Hamburg for compiling and providing meteorological measurement data. Moreover, we would like to thank Stefan Feigenspan and Stephan Nordmann of the German Federal Environmental Agency (Umweltbundesamt, UBA) for providing gridded emission totals for Hamburg. Peter Hurley (CSIRO) for support with TAPM. Britt Ann Høiskar (NILU) is thanked for permission to distribute the CityChem extension to the EPISODE model under the RPL license. We thank Paul Hamer (NILU) for the coordination of the two-part series publication on EPISODE and for the graphics of Figure 1. ECMWF ERA 5 reanalysis ensemble means for 2012 is distributed by the Copernicus Climate Change Service. The air quality model CMAQ is developed and maintained by the U.S. Environmental Protection Agency (US EPA). COSMO-CLM is the community model of the German climate research (www.clm-community.eu). The simulations with COSMO-CLM and CMAQ were performed at the German Climate Computing Centre (DKRZ) within the project "Regional Atmospheric Modelling" (Project Id 0302).





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





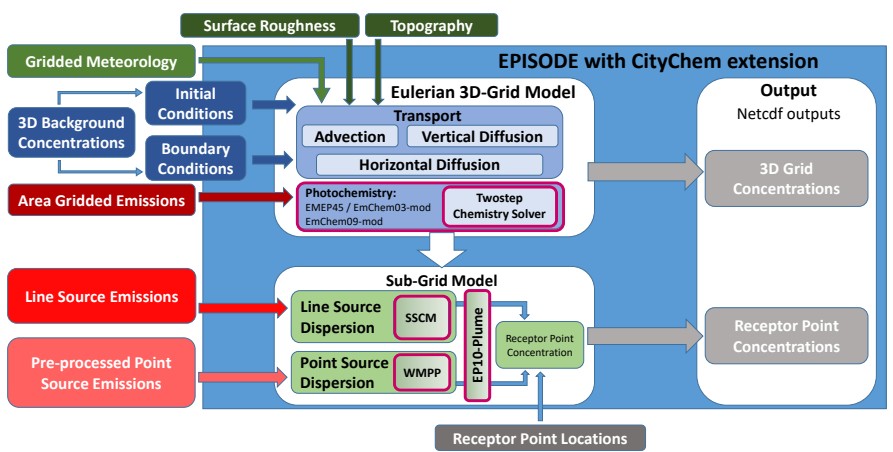

**Figure 1.** Schematic diagram of the EPISODE model with the CityChem extension (EPISODE-CityChem model). The large blue box represents operations carried out during the execution of the EPISODE model. The components of the EPISODE model are the Eulerian grid model and the sub-grid models. The inputs for EPISODE are specified on the periphery. Modules belonging to the CityChem extension are shown with a magenta frame.





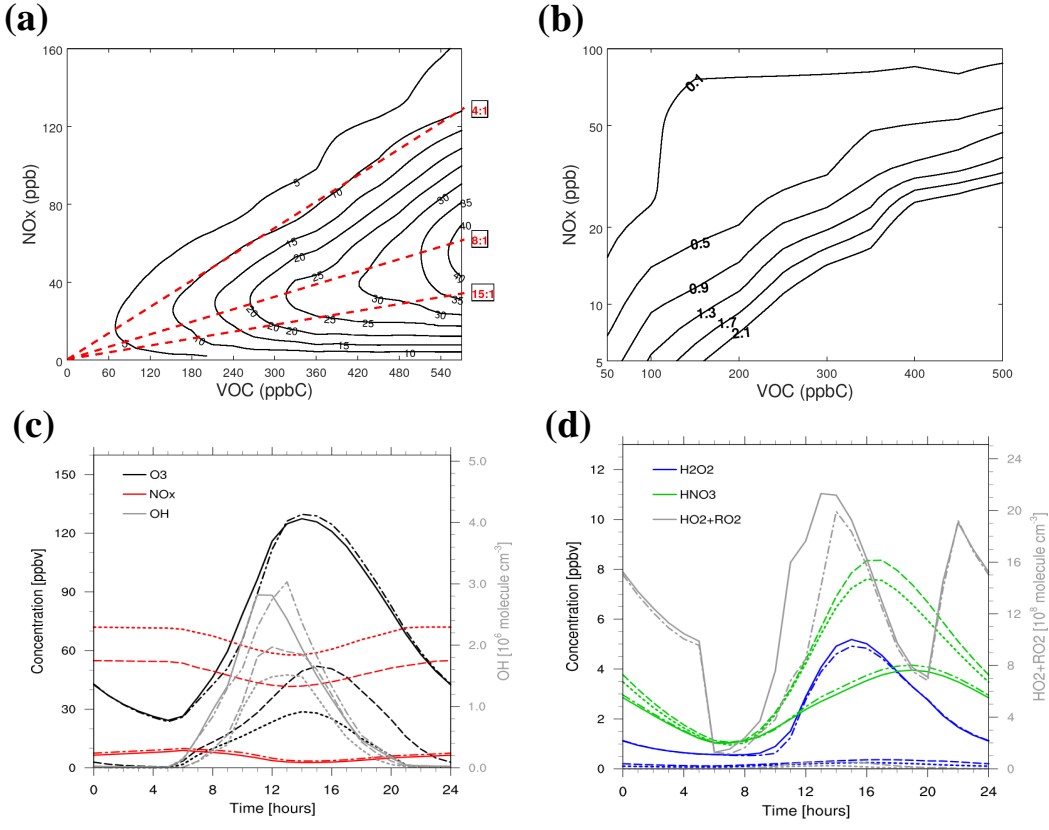

**Figure 2.** Test of relationships between ozone, $NO_X$ and NMVOC in EmChem03-mod: (a) ozone isopleth diagram, (b) isopleth diagram showing the ratio of the production rate of peroxides to the production rate of nitric acid, (c) concentration time series of $O_3$ (black), $NO_X$ (red), and OH (grey; second y-axis) and (d) concentration time series of $H_2O_2$ (blue), $HNO_3$ (green), and $HO_2 + RO_2$ concentration (grey, second y-axis). Daily concentration cycle as average from a test run with NMVOC emission of $695 \times 10^{-8}$ g s$^{-1}$ m$^{-2}$ and varying $NO_X$ emissions: $1 \times 10^{-8}$ g s$^{-1}$ m$^{-2}$ (solid lines), $2 \times 10^{-8}$ g s$^{-1}$ m$^{-2}$ (dash-dotted lines), $38 \times 10^{-8}$ g s$^{-1}$ m$^{-2}$ (dashed lines), and $55 \times 10^{-8}$ g s$^{-1}$ m$^{-2}$ (dotted lines). Lines of constant VOC/$NO_X$ ratio (4:1, 8:1 and 15:1) are annotated with red dashed lines in part (a). Note the logarithmic scale of the y-axis in part (b).



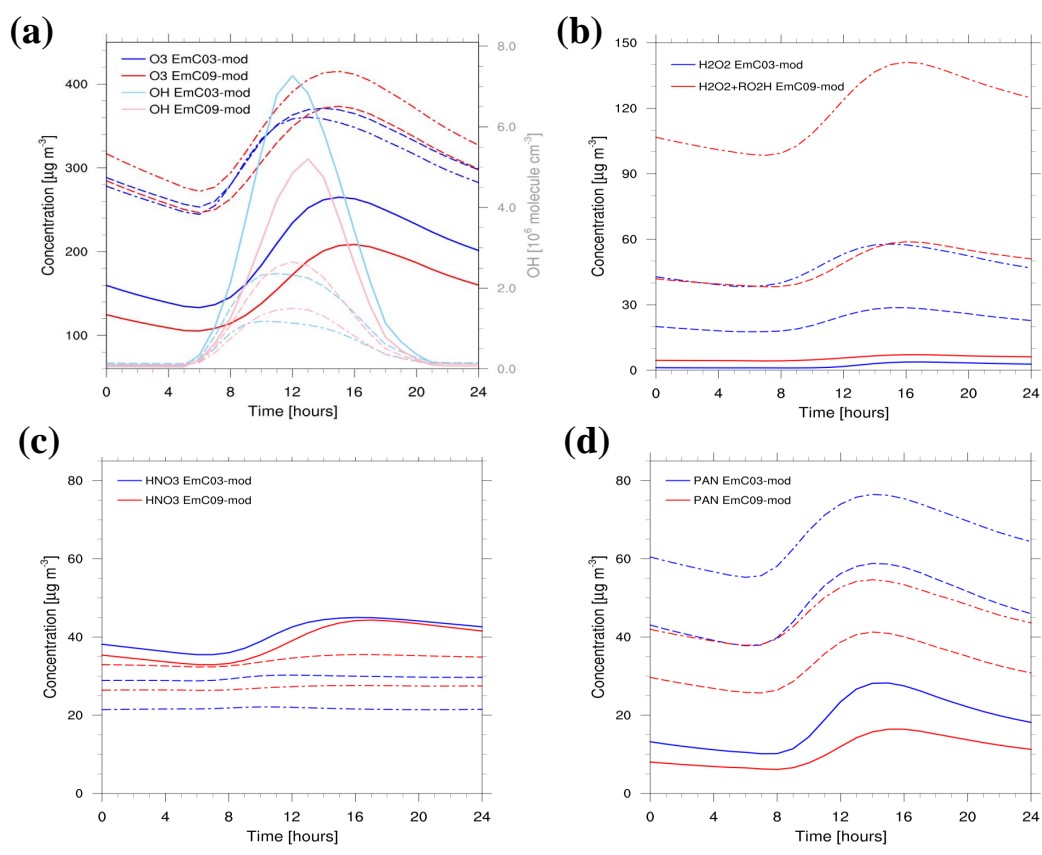

**Figure 3.** Comparison of EmChem09-mod (red lines) with EmChem03-mod (blue lines) for three different VOC/NO$_X$ ratios: (a) O$_3$ and OH (light colours, second y-axis); (b) H$_2$O$_2$ and organic peroxides (abbreviated as RO$_2$H); (c) HNO$_3$; and (d) PAN. Daily concentration cycle as average from a test run with NO$_X$ emission of $4.3 \times 10^{-8}$ g s$^{-1}$ m$^{-2}$ and NMVOC emissions corresponding to a VOC/NO$_X$ ratio of 4:1 (solid lines), 8:1 (dashed lines) and 15:1 (dash-dotted lines), respectively.





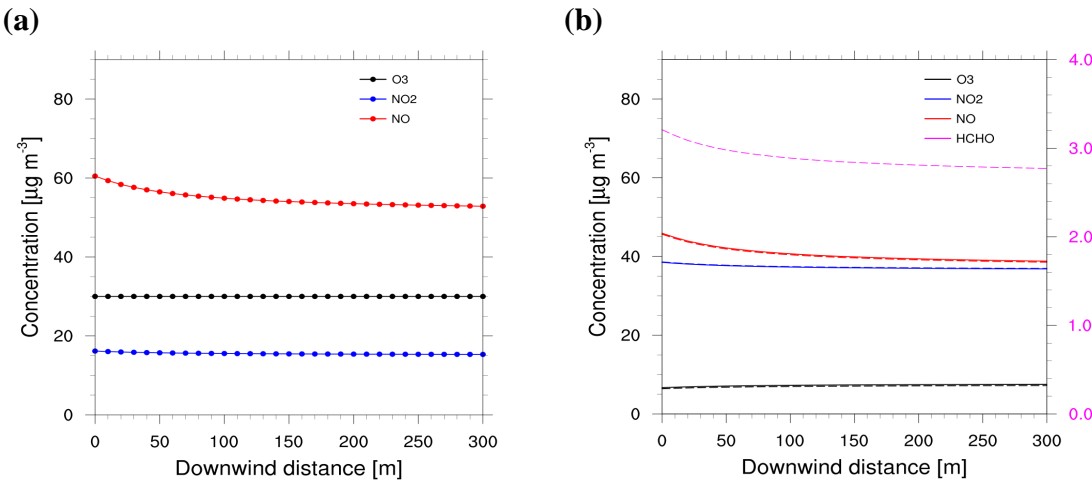

**Figure 4.** Photochemistry downwind of a line source in the SE-NW diagonal of the $1 \times 1\ \mathrm{km}^2$ grid cell: (a) concentration of $O_3$ (black), NO (red), $NO_2$ (blue) with no chemistry (lines with filled circles); and (b) concentration of $O_3$, NO, $NO_2$ and HCHO (magenta, second y-axis) with PSS (solid lines) and EP10-Plume (dashed lines and magenta line).

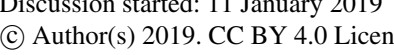
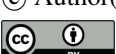


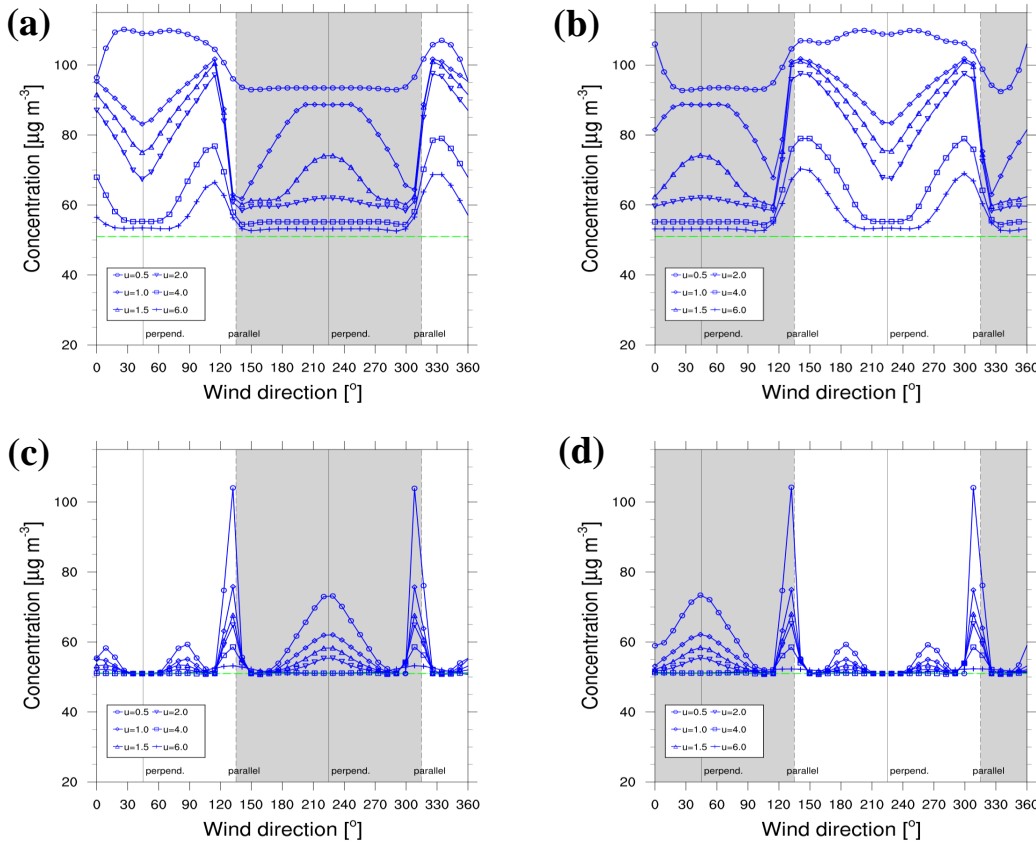

**Figure 5.** Test of the street canyon model: concentration of an inert tracer ($PM_{10}$) at the northeast side receptor (left column) and at the southwest side receptor (right column) as function of wind direction and wind speed: (a) from a simulation with SSCM; and (b) from a simulation with HIWAY-2 (no street canyon). The background concentration taken from the grid cell at a maximum distance from the line, is shown as a green line. Wind speed (in $m\,s^{-1}$) is given in the legend. Grey-shaded area indicates when the station is on the windward side of the street canyon. The solid vertical line indicates wind perpendicular to the street and dashed vertical line indicates wind parallel to the street.





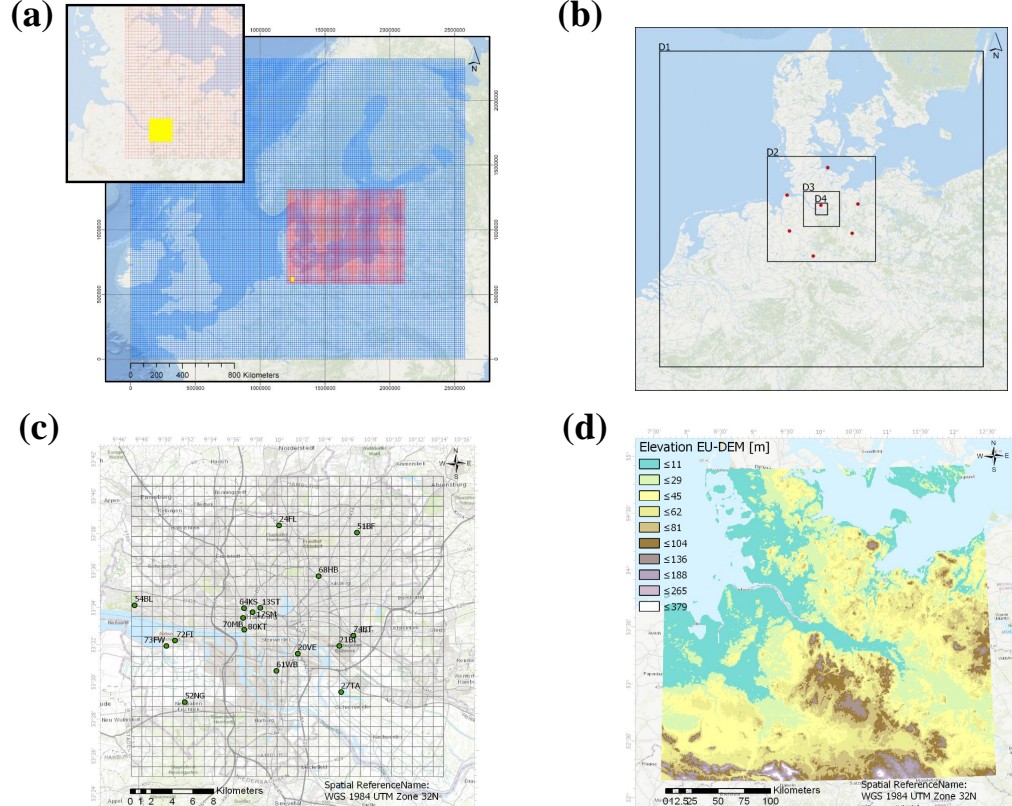

**Figure 6.** Model domains: (a) nested CTM simulations with CMAQ showing the 16-km (CD16, blue grid) and 4-km (CD04, red grid) resolution nests and the study domain (yellow box), inset in the upper left corner shows a zoom to the study region ; (b) domains used for the nested meteorological simulations within TAPM (D1–D4; black frames); the red dots indicate DWD station locations, from which observation data was assimilated to nudge the wind field predictions in TAPM; (c) study domain of Hamburg ($30 \times 30\,\mathrm{km}^2$) for simulation with CityChem, corresponds to the innermost nest (D4) of the nested model chain of both CMAQ and TAPM simulations. Green dots indicate positions of stations of the air quality monitoring network; and (d) terrain elevation (m a.s.l.) from the Digital Elevation Model over Europe (EU-DEM; https://www.eea.europa.eu/data-and-maps/data/eu-dem) for the extent of D2.





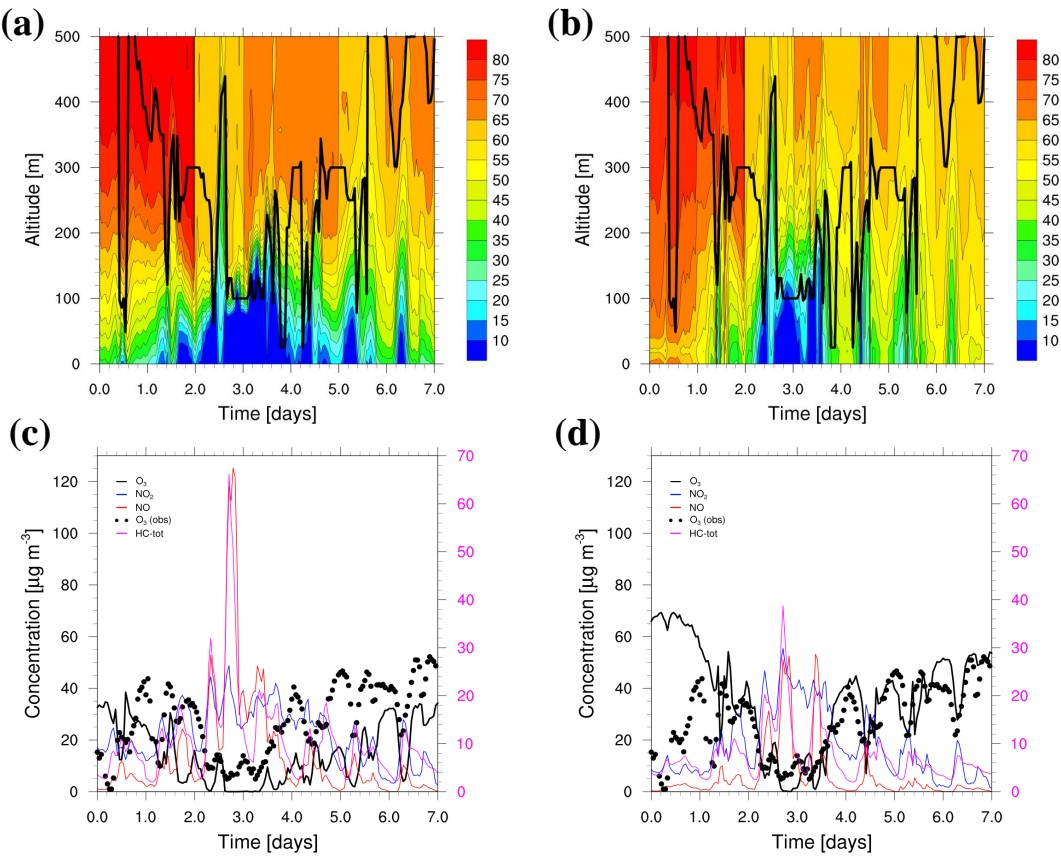

**Figure 7.** Test of the vertical eddy diffusivity using the standard $K(z)$-method versus using the new urban $K(z)$-method for a 7-day time sequence in January 2012, inspecting the 1-D column above the grid cell where station 13ST is located: (a) vertical profile of $O_3$ ($\mu g\,m^{-3}$) with standard $K(z)$-method, (b) vertical profile of $O_3$ ($\mu g\,m^{-3}$) with new urban $K(z)$-method (black line in the vertical profile plots denotes mixing height, $h_{mix}$), (c) time series of ground air (lowermost model layer) concentrations with standard $K(z)$-method, and (d) time series of ground air concentrations with new urban $K(z)$-method. In figure parts (c) and (d), black line indicates $O_3$, red line NO, blue line $NO_2$, magenta line HC−tot (refers to the second y-axis) and black circles are observed $O_3$ concentrations at station 13ST.





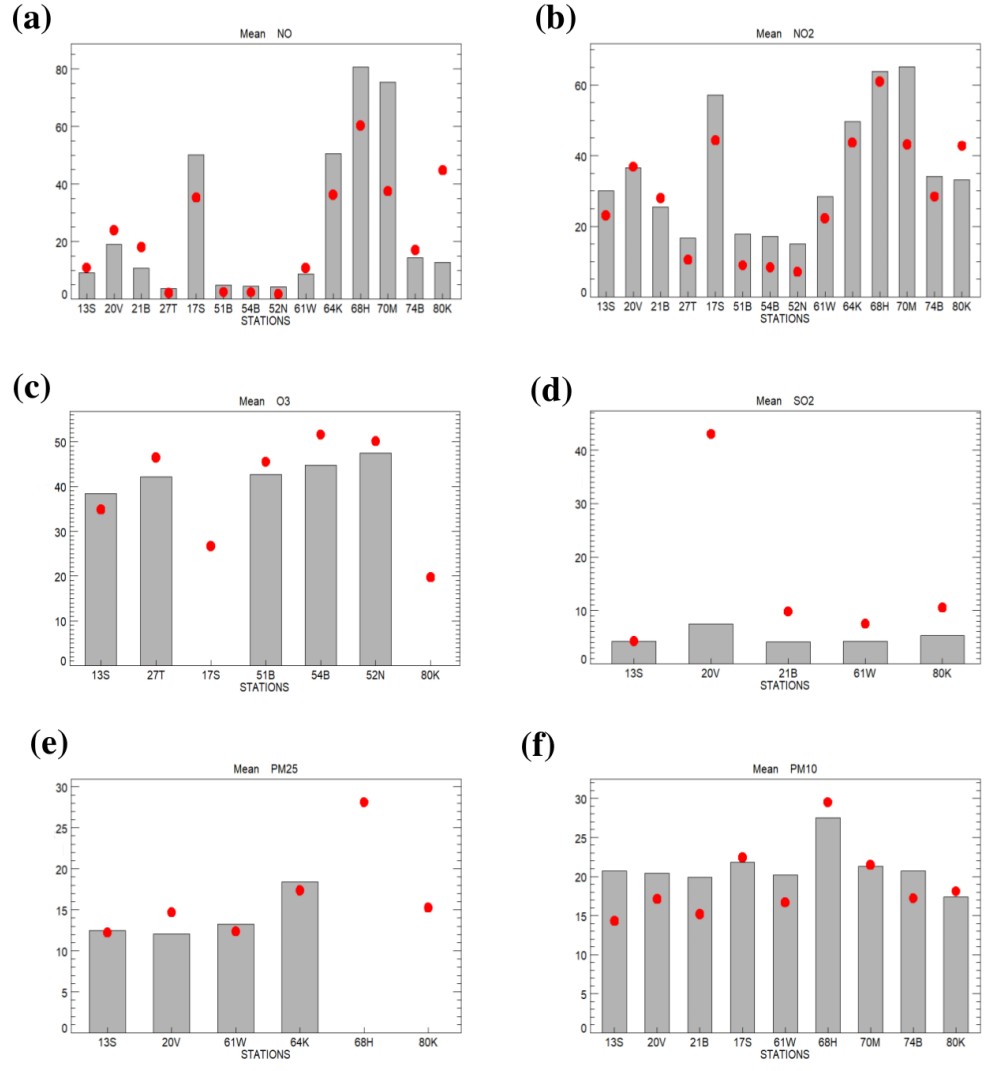

**Figure 8.** Bar plots comparing modelled and observed mean annual concentrations (in $\mu g\,m^{-3}$) for monitoring stations of the Hamburg AQ network: (a) NO, (b) $NO_2$, (c) $O_3$, (d) $SO_2$, (e) $PM_{2.5}$, and (f) $PM_{10}$. Observed values as grey filled bars, modelled values indicated as red circles. Observation data not shown for stations were data completeness was less than 75 % (for $O_3$: 17SM and 80KT, for $PM_{2.5}$: 68HB and 80KT)



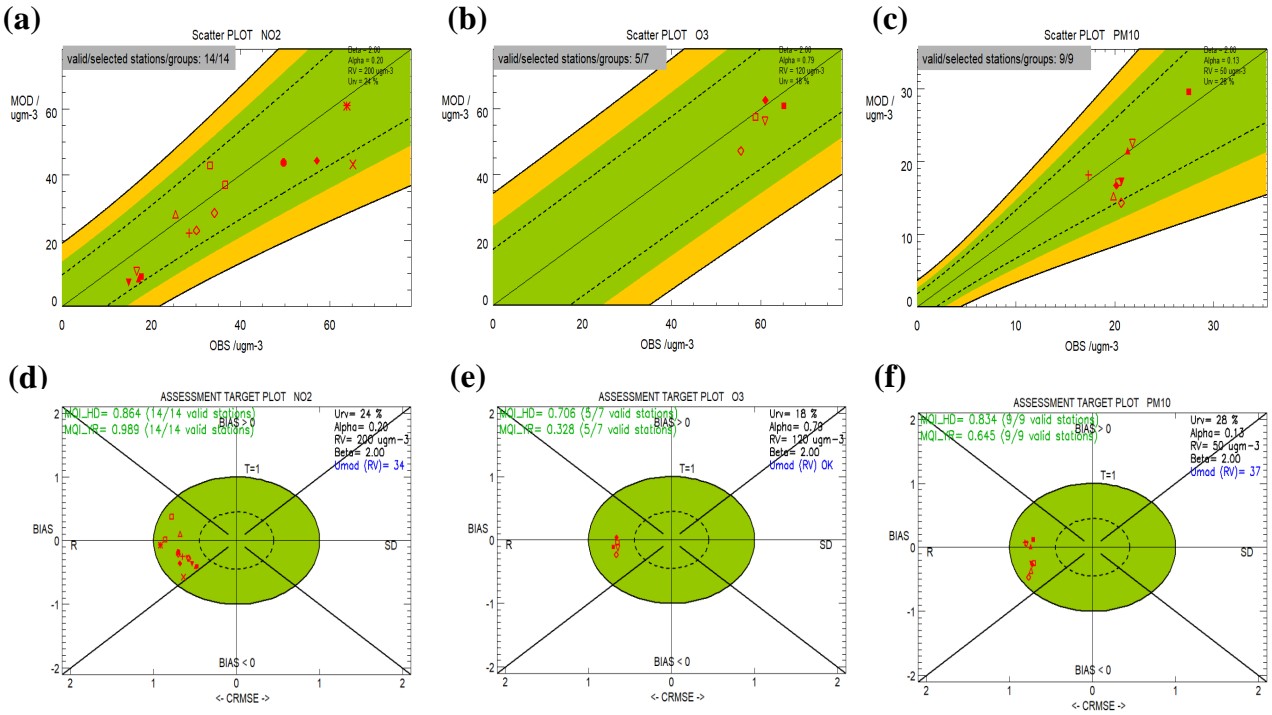

**Figure 9.** Model performance evaluation for monitoring stations of the Hamburg AQ network: (a) scatter diagram for $NO_2$ hourly values, (b) scatter diagram for $O_3$ daily max. of 8-h running mean, (c) scatter diagram for $PM_{10}$ daily values (d) target plot $NO_2$ hourly values, (e) target plot $O_3$ daily max. of 8-h running mean, and (f) target plot $PM_{10}$ daily values. In the scatter diagrams, the uncertainty parameters ($\beta$, $\alpha$, RV, $u_r^{RV}$) used to produce the diagram calculated are listed on the top right-hand side; dashed and solid lines indicate $NMB/2RMS_U$ ratios of 0.5 and 1. The Target diagrams further indicate MQI ($MQI_{HD}$ for hourly/daily values) for the station most distant from the origin and the model uncertainty, Umod(RV).



**Figure 10.** Spatial maps of the annual concentration average ($\mu g\ m^{-3}$) for Hamburg from CityChem model simulation output for the receptor grid ($100 \times 100\ m^2$): (a) $NO_2$, (b) $O_3$, (c) total NMVOC, (d) $SO_2$, (e) $H_2SO_4$, and (f) $PM_{2.5}$. Maps are created using ESRI® ArcMap TM, with an overlay on a topographic base map showing the network of main roads as grey lines.





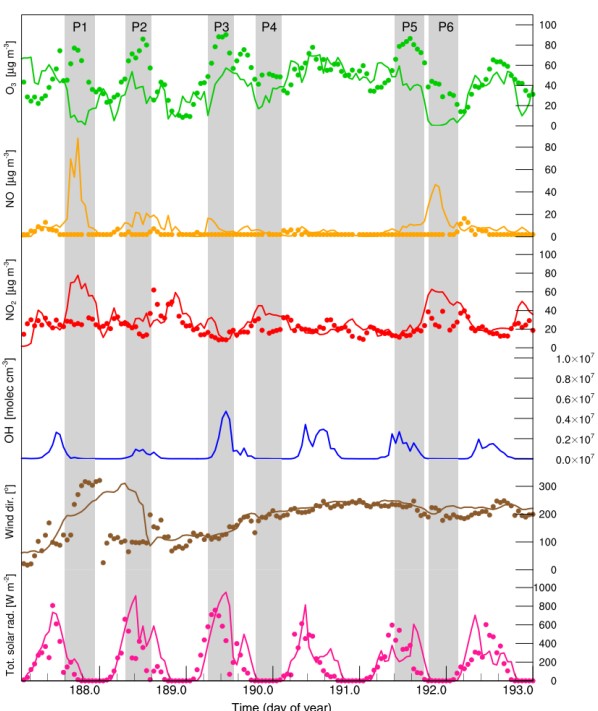

**Figure 11.** Ozone variation in a summer 6-day episode (5–10 July 2012): hourly concentrations of $O_3$ ($\mu g\,m^{-3}$), NO ($\mu g\,m^{-3}$), $NO_2$ ($\mu g\,m^{-3}$), OH radical (molecule $cm^{-3}$), wind direction (degrees) and hourly total solar radiation (W $m^{-2}$) at station Sternschanze (13ST) representative of inner-city urban background. Observations indicated by filled circles and model results by solid lines. Six short periods with ozone underpredicted by the model at 13ST shown as grey shaded area, labelled with P1–P6.



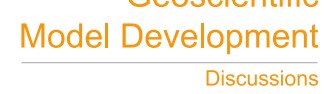

**Table 1.** The geometry of three generic street canyon types in CityChem. For street canyons of type "urban medium", $H_{sc}$ is taken as the mean value of "urban low" and "urban high".

| TAPM land use class | Street canyon type | Average building height, $H_{sc}$ [m] | Building density |
|---|---|---:|---|
| 32 | urban low | 6.6 | sparse-built area |
| 33 | urban medium | 12.3 | medium density area |
| 34 and 35 | urban high | 18.0 | dense-built area |

**Table 2.** Setup of the 1-D column model for the tests of model extensions.

| Model parameter | EmChem03-mod | EmChem09-mod | EP10-Plume | SSCM | WMPP |
|---|---|---|---|---|---|
| | | Photochemistry | | line source | point source |
| 1-D column grid cell area and height | $1 \times 1\,km^2$ 3750 m | $1 \times 1\,km^2$ 3750 m | $1 \times 1\,km^2$ 3750 m | $1 \times 1\,km^2$ 3750 m | $1 \times 1\,km^2$ 3750 m |
| Eulerian grid transport | vertical upstream advection and semi-implicit Crank-Nicholson diffusion scheme with the new urban $K(z)$ parameterization | | | | no transport |
| Eulerian grid photochemistry | EmChem03-mod | EmChem09-mod | — | EmChem09-mod | — |
| Local photochemistry | — | — | EP10-Plume | PSS | — |
| Wind direction (WD) and wind speed (WS) | WD: 225° WS: $0.1\,m\,s^{-1}$ | WD: 225° WS: $0.1\,m\,s^{-1}$ | WD: 225° WS: $1\,m\,s^{-1}$ | various WD and WS values | WD: 225° WS: $1\,m\,s^{-1}$ |
| Other meteorol. data | the daily cycle of meteorological conditions typical for July in Hamburg, Germany | | | | |
| Background concentration in [$\mu g\,m^{-3}$] | $O_3$: 60 NO: 5 $NO_2$: 10 | $O_3$: 60 NO: 5 $NO_2$: 10 | $O_3$: 30 NO: 5 $NO_2$: 10 | $O_3$: 60 $PM_{10}$: 10 | $SO_2$: 0 |
| Emission sources | area source | | one line source in the SE-NW diagonal | | one point source |
| Emissions | various $NO_X$ and VOC emission rates | $NO_X$: $4.3 \times 10^{-8}$ VOC: $(17–65) \times 10^{-8}$ [$g\,s^{-1}\,m^{-2}$] | $NO_X$: $2.0 \times 10^{-4}$ VOC: $3.9 \times 10^{-4}$ [$g\,(s\,m)^{-1}$] | $PM_{10}$: $1.6 \times 10^{-4}$ [$g\,(s\,m)^{-1}$] (inert) | $SO_2$: $1.0\,g\,s^{-1}$ (inert) |



**Table 3.** Test of point source dispersion of $SO_2$ (handled as an inert tracer) for different atmospheric stability conditions in flat terrain at wind speed $1\,\mathrm{m\,s^{-1}}$. Hourly concentration is given at the location where the maximum is found for the 5-day average within a radius of $2.0\,\mathrm{km}$ around the point source. Parameterisation of point source: exhaust gas temperature: 20°C; stack height: 10 m; exit velocity: $5\,\mathrm{m\,s^{-1}}$ ; stack radius: 0.5 m (circular opening). Emission rate: $1\,\mathrm{g\,s^{-1}}$. The default is the standard parameterization in EPISODE.

| Parameter | slightly stable | | neutral | | slightly unstable | | very unstable | |
|---|---|---|---|---|---|---|---|---|
| | default | WMPP | default | WMPP | default | WMPP | default | WMPP |
| $\Delta T/\Delta z$ [K m$^{-1}$] | 0.01 | | -0.01 | | -0.016 | | -0.10 | |
| Effective emission height, $H_{\mathrm{eff}}$ [m] | 54.4 | 47.4 | 85.8 | 40.5 | 85.9 | 38.8 | 86.0 | 32.4 |
| Distance of max. ground conc. [m] | 1700 | 1700 | 830 | 390 | 830 | 390 | 830 | 250 |
| Hourly ground air concentration at maximum [µg m$^{-3}$] | 0.03 | 0.03 | 18.7 | 79.8 | 18.7 | 85.3 | 18.6 | 112.7 |



**Table 4.** Vertical and horizontal structure of the 3-D Eulerian grid of the EPISODE-CityChem model and comparison with TAPM and CMAQ models for the simulation of AQ in Hamburg.

| Model dimension | EPISODE-CityChem | TAPM | CMAQ |
|---|---|---|---|
| Horizontal size of the domain (X × Y) | $30 \times 30\,\text{km}^2$ | $30 \times 30\,\text{km}^2$ | $916 \times 724\,\text{km}^2$ |
| Horizontal resolution | 1000 m | 1000 m | 4000 m |
| Model grid and coordinate system | Universal Transverse Mercator (UTM) coordinate system with WGS 1984 as reference geoid | Universal Transverse Mercator (UTM) coordinate system with WGS 1984 as reference geoid | Equidistant grid with Lambert conformal projection |
| Vertical dimension and coordinate | 24 layers, terrain-following sigma-coordinate system | 30 layers, terrain-following sigma-coordinate system | 30 layers, sigma hybrid pressure coordinate system |
| Lowest model layer depth [m] | 17.5 | 10 | 36 |
| Number of vertical layers below 1000 m | 16 | 16 | 12 |
| Vertical top height | 3750 m | 8000 m | 100 hPa |




**Table 5.** The configuration of EPISODE-CityChem model processes in the AQ simulations for Hamburg.

| Process | Option, numerical scheme | Description, reference |
|---|---|---|
| Vertical advection and diffusion | Vertical upstream advection and semi-implicit Crank-Nicholson diffusion scheme with the new urban $K(z)$ parameterisation | Byun et al. (1999); Hamer et al. (2019) |
| Horizontal 2-D advection | Positive definite 4th degree Bott scheme | Bott (1989); Hamer et al. (2019) |
| Horizontal 2-D diffusion | Fully explicit forward Euler scheme | Smith (1985); Hamer et al. (2019) |
| Photochemistry on the Eulerian main grid | EmChem09 reaction scheme solved with TWOSTEP algorithm | Sect. 2.1.2; Appendix A |
| Sub-grid photochemistry | EP10-Plume reaction scheme solved with TWOSTEP algorithm | Sect. 2.1.3; Appendix A |
| Sub-grid line source dispersion | HIWAY-2 model coupled with SSCM for street canyons | Sect. 2.2.1 |
| Sub-grid point source dispersion | SEGPLU model with WMPP based plume rise | Sect. 2.2.2 |





**Table 6.** Emission sectors data for the simulation of air quality in Hamburg. Classification according to Selected Nomenclature for sources of Air Pollution (SNAP). The top heights of layer 1, 2, 3, and 4 are 17.5 m, 37.5 m, 62.5 m, and 87.5 m above ground, respectively. Point source emission data for SNAP categories 01, 04, 05, and 09 were collected from the PRTR database (PRTR, 2017) and from the registry of emission data for point sources in Hamburg as reported under the German Federal Immission Control Act (BImSchV 11).

| SNAP | SNAP name | Source type | Vertical distribution | Emission data source and approach |
|---|---|---|---|---|
| 01 | Combustion in energy and transformation industries | Point | Plume rise | Bottom-up approach. Data set on European stacks by Pregger and Friedrich (2009). |
| 02 | Non-industrial combustion plants (domestic heating) | Area (1 × 1 km²) | 80 % in layer 1; 20 % in layer 2 | GRETA software (Schneider et al., 2016), top-down with spatial and temporal disaggregation. |
| 03 | Combustion in manufacturing industry | Area (1 × 1 km²) | 80 % in layer 1; 20 % in layer 2 | GRETA software (Schneider et al., 2016), top-down with spatial and temporal disaggregation. |
| 04 | Production processes | Point | Plume rise | Bottom-up approach. Data set on European stacks by Pregger and Friedrich (2009). |
| 05 | Extraction and distribution of fossil fuels and geothermal energy | Point | Plume rise | Bottom-up approach. Data set on European stacks by Pregger and Friedrich (2009). |
| 06 | Solvent and other product use | Area (1 × 1 km²) | 100 % in layer 1 | GRETA software (Schneider et al., 2016), top-down with spatial and temporal disaggregation. |
| 07 | Road transport | Line | At 0 m above ground | Bottom-up method using emission factors from HBEFA version 3.1 (UBA, 2010). |
| 08 | Other mobile sources and machinery | Area (1 × 1 km²) | Shipping: 25 % in each layer 1-4. Other: same as for SNAP cat. 10. | Shipping: Aulinger et al. (2016). Other: GRETA software (Schneider et al., 2016), top-down with spatial and temporal disaggregation. |
| 09 | Waste collection, treatment and disposal activities | Point | Plume rise | Bottom-up approach. Data set on European stacks by Pregger and Friedrich (2009). |
| 10 | Agriculture and farming | Area (1 × 1 km²) | 80 % in layer 1; 20 % in layer 2 | GRETA software (Schneider et al., 2016), top-down with spatial and temporal disaggregation. |





**Table 7.** Comparison of performance statistics of CMAQ (4-km res.), EPISODE-CityChem and TAPM for O$_3$ based on hourly concentration values for a 14 days period in July 2012 at four urban background monitoring stations. Observed mean concentrations at 52NG, 51BF, 13ST and 27TA are 57.42, 49.06, 47.57, and 49.34 μgm$^{-3}$, respectively. Number of observations: $N = 336$. Definition of statistical indicators as in Appendix C. EPISODE-CC is short for EPISODE-CityChem.

| Station code | Model | $\overline{M}$ [μg m$^{-3}$] | Bias [μg m$^{-3}$] | Corr [–] | RMSE [μg m$^{-3}$] |
|---|---|---|---|---|---|
| 52NG | CMAQ | 50.02 | -7.40 | 0.74 | 12.18 |
| | EPISODE-CC | 57.66 | 0.23 | 0.46 | 15.03 |
| | TAPM | 53.60 | -3.82 | 0.48 | 13.96 |
| 51BF | CMAQ | 51.47 | 2.41 | 0.76 | 11.11 |
| | EPISODE-CC | 52.41 | 3.35 | 0.57 | 22.15 |
| | TAPM | 51.08 | 2.02 | 0.50 | 17.85 |
| 13ST | CMAQ | 46.95 | -0.62 | 0.79 | 12.53 |
| | EPISODE-CC | 40.23 | -7.33 | 0.65 | 15.83 |
| | TAPM | 42.13 | -5.43 | 0.61 | 17.20 |
| 27TA | CMAQ | 48.95 | -0.39 | 0.75 | 12.82 |
| | EPISODE-CC | 52.98 | 3.64 | 0.40 | 19.12 |
| | TAPM | 36.95 | -12.39 | 0.42 | 19.90 |

**Table 8.** Model performance statistics for O$_3$ based on daily maximum of the 8-h running mean concentration at all stations with sufficient data availability in 2012.

| Station code | $\overline{O}$ [μg m$^{-3}$] | $\overline{M}$ [μg m$^{-3}$] | STD$_O$ [μg m$^{-3}$] | STD$_M$ [μg m$^{-3}$] | NMB [%] | Corr [–] | RMSE [μg m$^{-3}$] | IOA [–] |
|---|---|---|---|---|---|---|---|---|
| 13ST | 55.52 | 47.11 | 25.26 | 18.86 | -15.15 | 0.42 | 25.80 | 0.63 |
| 27TA | 58.88 | 57.41 | 23.88 | 19.07 | -2.49 | 0.39 | 24.01 | 0.61 |
| 51BF | 61.00 | 56.24 | 25.41 | 19.93 | -7.80 | 0.46 | 24.46 | 0.67 |
| 54BL | 61.08 | 62.57 | 24.94 | 19.50 | 2.44 | 0.41 | 24.52 | 0.63 |
| 52NG | 65.16 | 60.88 | 25.17 | 20.15 | -6.57 | 0.35 | 26.58 | 0.59 |





**Table 9.** Model performance statistics for PM$_{10}$ based on daily mean concentration at all stations with sufficient data availability in 2012.

| Station code | $\overline{O}$ [μg m$^{-3}$] | $\overline{M}$ [μg m$^{-3}$] | STD$_O$ [μg m$^{-3}$] | STD$_M$ [μg m$^{-3}$] | NMB [%] | Corr [–] | RMSE [μg m$^{-3}$] | IOA [–] |
|---|---|---|---|---|---|---|---|---|
| 13ST | 20.67 | 14.32 | 11.12 | 7.03 | -30.71 | 0.35 | 12.22 | 0.49 |
| 20VE | 20.38 | 17.13 | 9.74 | 10.11 | -15.95 | 0.41 | 9.81 | 0.61 |
| 21BI | 19.88 | 15.18 | 10.11 | 7.66 | -23.67 | 0.34 | 10.67 | 0.51 |
| 17SM | 21.79 | 22.45 | 11.61 | 11.56 | 3.03 | 0.32 | 11.35 | 0.50 |
| 61WB | 20.17 | 16.70 | 9.70 | 10.68 | -17.22 | 0.40 | 9.99 | 0.61 |
| 68HB | 27.52 | 29.51 | 11.03 | 17.34 | 7.23 | 0.27 | 12.20 | 0.53 |
| 70MB | 21.32 | 21.49 | 10.38 | 11.39 | 0.80 | 0.35 | 10.18 | 0.56 |
| 74BT | 20.68 | 17.23 | 9.82 | 9.13 | -16.67 | 0.28 | 10.34 | 0.51 |
| 80KT | 17.35 | 18.12 | 9.06 | 9.96 | 4.43 | 0.34 | 9.32 | 0.56 |

**Table 10.** Model performance statistics for NO$_2$ based on hourly concentration at all stations with sufficient data availability in 2012.

| Station code | $\overline{O}$ [μg m$^{-3}$] | $\overline{M}$ [μg m$^{-3}$] | STD$_O$ [μg m$^{-3}$] | STD$_M$ [μg m$^{-3}$] | NMB [%] | Corr [–] | RMSE [μg m$^{-3}$] | IOA [–] |
|---|---|---|---|---|---|---|---|---|
| 13ST | 30.09 | 23.08 | 15.73 | 15.90 | -23.31 | 0.58 | 16.04 | 0.73 |
| 20VE | 36.54 | 36.90 | 17.36 | 25.40 | 0.97 | 0.46 | 23.21 | 0.65 |
| 21BI | 25.45 | 27.96 | 15.12 | 15.66 | 9.89 | 0.45 | 16.35 | 0.67 |
| 27TA | 16.73 | 10.56 | 11.34 | 12.11 | -36.88 | 0.43 | 13.97 | 0.63 |
| 17SM | 57.12 | 44.35 | 27.60 | 25.49 | -22.34 | 0.59 | 27.35 | 0.72 |
| 51BF | 17.85 | 8.99 | 12.08 | 9.01 | -49.62 | 0.55 | 13.60 | 0.65 |
| 54BL | 17.20 | 8.37 | 12.65 | 9.19 | -51.35 | 0.55 | 13.91 | 0.64 |
| 52NG | 14.95 | 7.12 | 11.67 | 10.01 | -52.37 | 0.46 | 13.80 | 0.62 |
| 61WB | 28.49 | 22.28 | 15.19 | 15.93 | -21.80 | 0.47 | 17.13 | 0.67 |
| 64KS | 49.67 | 43.72 | 24.03 | 25.33 | -11.97 | 0.58 | 23.40 | 0.75 |
| 68HB | 63.85 | 60.96 | 36.00 | 38.48 | -4.53 | 0.53 | 36.33 | 0.73 |
| 70MB | 65.18 | 43.17 | 27.29 | 24.32 | -33.76 | 0.55 | 32.97 | 0.65 |
| 74BT | 34.13 | 28.44 | 16.97 | 20.19 | -16.67 | 0.52 | 19.33 | 0.70 |
| 80KT | 33.16 | 42.82 | 16.17 | 20.00 | 29.13 | 0.39 | 22.43 | 0.59 |