# Peer review of "The Eulerian urban dispersion model EPISODE. Part II: Extensions to the source dispersion and photochemistry for EPISODE-CityChem v1.2 and its application to the city of Hamburg"

_Geoscientific Model Development, 2018_

## Referee Comment (RC1) · Anonymous Referee #1 · 6 Mar 2019

Review of "*The Eulerian urban dispersion model EPISODE. Part II: Extensions to the source dispersion and photochemistry for EPISODE-CityChem v1.2 and its application to the city of Hamburg*" by Matthias Karl et al.

The submitted work describes the EPISODE model with the CityChem extension. When I read the manuscript was part 1 of the article series not available, which makes it somewhat difficult to assess the need and value of the the CityChem extension. The reader of this particular manuscript is also left with an incomplete description of the EPISODE-CityChem system.

EPISODE provides a promising set up with sub-grid scale Gausian models embedded into a Eulerian model with relevant photochemistry. As understood from the present manuscript does the CityChem extension offer three improvements compared to the original EPISODE model:

1. Modification of the sub-grid photochemistry scheme
2. Modification of the line source emission model
3. Modifications to the plume rise calculations

The three improvements are explained and visualised by presenting results from idealised sensitivity studies. In my view do the authors not quite get to the point when the respective improvement is presented and visualized.

-Section 2.1 discusses a number of chemical schemes. From my reading of section 2.1 can I not tell what photochemical module(s) is (are) actually used in EPISODE-CityChem.  Section 3.2 compares the performance of the updated photochemical scheme in the EPISODE-CityChem system with the standard PSS assumption. The conclusion (not shown in any graph!) is that the new scheme (EP10-Plume) behaves very similar to the original PSS-scheme. This is an interesting and valuable finding, but does not provide support for introducing an alternative photochemical scheme in the original EPISODE model.

-I found section 2.2.1 -describing the improvements of the line source emission model particularly difficult to follow.

-Section 2.2.2 details the calculation of wind speed at the actual plume height, and is a good reference for documenting how winds are calculated in the EPISODE-CityChem model although it is basically a standard similarity approach. As part 1 of the article series is not yet available it is impossible for the reader to know whether the "WMPP" is the sole improvement to the original EPISODE model or if the whole concept of plume rise calculations (Briggs, 1969; 1971; 1975) are also part of the development.

The set up of the Hamburg domain is illustrated with a figure of the domains (Fig. 6) along with the text in section 4.1. It is still very difficult to understand the whole modelling chain of EPISODE-CityChem (and the other models participating in the comparison). For boundary concentrations of air pollutants there appears to be 2-3 CMAQ nests inside the global FMI APTA chemical transport model. Similarly, is meteorology dynamically downscaled from ERA5 in, at least, five nested TAPM domains. An impressive undertaking, indeed, but rather bulky and inconvenient for most applications.

The location of the Hamburg study domain in the Southwest corner of the 4km x 4 km CD04-CMAQ domain does not make sense. The CD04 values at this point will hardly be any different than in the parent CD16 model, as the boundary is only 20-30 km away (less than an hour with a typical wind speed). I do not agree with the authors that: "This is considered to be sufficient to avoid that concentrations in the study domain are affected by domain border effects" (page 18-19, lines 34 and 1, respectively).

Many of the results presented are based on 1-2 week simulations (including, for example, section 4.1.2 and 4.1.3 and the final ~30 rows of section 4.2). I would be reluctant to draw any major conclusions from such limited data series.

The description of the "second experiment" involving the TAPM air quality model (P21, L6-17) is difficult to follow. Why only evaluate 14 days? I also lack a brief discussion on why the model results different. Is it due to different emissions or - for CMAQ, the seemingly best model- different meteorology? For the full 12-month evaluation (Figs 8,9,11 and Tables 8-10) is only EPISODE-CityChem compared against observations. Why is not the three-model comparison (Table 7) extended to the full 12-month period?

In this section 4.1.2 the high-resolution EPISODE-CityChem is compared with two coarser-resolution models for a two-week simulation over the city of Hamburg. As far as I can judge from the presentation is CMAQ superior to EPISODE-CityChem, which is interesting, since CMAQ is used as boundaries for EPISODE-CityChem. The test is not a good promotion for EPISODE-CityChem.

The model performance evaluation of EPISODE-CityChem using FAIRMODE DELTA Tool is brief, but difficult to follow (Figure 9 and lines 1-19 on page 26). It could perhaps be dropped (or moved to the Supplement as a stand-alone entity), in favour of a more focussed presentation.

The manuscript is rather long but at the same time superficial. I would like to recommend the authors to focus and streamline the presentation.

There are several annoying features that distracts the reader from truly appreciating the presentation. For example:

a.  As already pointed out is part 1 of the article series, not yet available. Still, the text frequently refers to Hamer et al. for explanations of background and details omitted in the presentation.
b.  The manuscript is long and the average reader lose focus after a while. Consider shorten and remove some parts:
    -TWOSTEP is unnecessary mentioned and explained several times.
    -There is no need, in my opinion to describe, in great detail, the interpolation and file format conversion of the boundary data to the EPISODE model (section 2.3.1).
    -What is the rationale for testing the EMEP45, EmChem03-mod and EmChem09-mod in this presentation. Doesn't that belong to the standard EPISODE model (part 1 of the article series?).
    -The section of comparing the new and standard K(z) profile (4.1.3) is not suited for this manuscript as the new K(z) method is described in part 1 of the article series. It also distracts the reader from the CityChem extensions -which should be the focus of this manuscript.
c.  I don't think it is good practice to discuss results in the main text that are only presented in the Supplement. This happens throughout the manuscript.
d.  The order of how items are presented and discussed is stochastic and confusing for the reader. Check, for example, section 4.2 where the different species are discussed in a seemingly random order, not even following the order in which they are presented in the figures.

e. An overwhelming number of acronyms are introduced and used throughout the manuscript. Acronyms admittedly decrease the length of the presentation but also decrease legibility when these are first introduced and when the reader needs to go back and search for the explanation of a particular term. Would it be possible to put all explanations in a common table, for easy reference?
Some acronyms are not described the first time they are introduced. For example $w\_sc$ and $L\_max$ on page 8; TAPM on page 9;CMAQ on page 11; …

Minor issues / typos:

1. Page 1, 2nd sentence: "… lower latitudes …". Why open with this in the Abstract? Hamburg is hardly lower latitudes. For most readers Hamburg clearly classifies as belonging to "…northern European cities."
2. P2, L20: "…150 km2 in size…". Unrealistically small domain if you use a 1 x 1 km2 resolution.
3. P3, Bullets: Change place/numbering of item 1 and 3. In the following sections you discuss modifications to photochemistry first (Section 2.1), then the street canyon model (Section 2.2.1) and finally the extension that provides realistic winds at actual plume height (section 2.2.2). Please also introduce the three modifications in the same order in the abstract (it is currently photochemistry, plume rise, line source).
4. P4, L16: "… fine particulate matter with (PM2.5;…". The sentence is not correct.
5. P4, L18-19: "… NO2, the major pollutant in many cities of northern Europe". This statement is opposite to what is said in the introduction (P1, L21-23), where PM and ozone is pointed out as the main air pollution issues in Europe!
6. P6, L21-22: "… NOx are often below 1 ppbv." This is a too strong sentence. NOx is rarely below 1 ppbv in "rural and sub-urban areas" of northern Europe.
7. P8, Eq 3: $sigma\_w0$ should likely be squared.
8. P9, L13-14: "…the average turbulence of the hypotenuse of the trapezium (slant edge towards the opposite street side)." Can this be explained so the un-initiated understands it?
9. P9, L15: ($L\_base/2$) should likely be squared.
10. P10, L1: Consider renaming the header of this section as it mainly goes through the method of deducing windspeed at height z. The "plume rise model" is only described by referring to Briggs.
11. P10, L20: I do not understand the +0.01 term. Can it be explained?
12. P12, Eq10: "$C\_point,s$" should likely read "$C\_point,p$"
13. P14, L12: "… the ratio 10:1; …". Why not indicate this line in Fig. 2a?
14. P15, L22: "Despite a slightly …". This sentence is not complete.
15. P17, L7: What is the difference between "actual plume height" and "final plume height" ?
16. P17, L25: "…roughly proportional to …" -> "…roughly inversely proportional to …" (?)
17. P17, L26: "…40.5 m (neutral) to 32.4 m …". Can you really defend three significant digits?
18. P19, L18: "…factor of 2.8 …". Why not "…factor of 8 …"?
19. P20, L22: "But considerations …". Strange start of sentence.
20. P23-28: Section 4.2 is much longer than any of the other sections. Can it be divided into sub-sections for increased legibility?
21. P23, L3: "…when on the following days." -> "…than on the following days." (?)
22. P24, L1: "26.56". Can you really defend four significant digits?
23. P24, L11-12. Sentence including "…, modelled and show …" is not correct.

24. P25, L7: "bias is within +/- 25 ug/m3". From Fig. 8b it is more like +/- 10 ug/m3
25. P25, L12: "…summer (JJA) mean…". Summer mean values are not shown in the manuscript.
26. P25, L19: "…diurnal cycles…". Not shown in the main manuscript. Avoid discussing details only shown in supplement.
27. P25, L27: "… underestimate observed PM2.5 and PM10…". From Fig. 8e,f, these are excellently reproduced!
28. P27, L5: "… in some distance…" -> "… at some distance…" (?)
29. P29, L5: "…EMEP unified model". I bvelieve "unified" (as of combining photochemistry and acid deposition in the same model) has been dropped since several years.
30. P29, L9,L14: Is there a reason why you spell out "photochemical steady-state" on line 14, but abbreviate to PSS on line 9?
31. P30, L23: "…, yet unregulated pollutants, in cities …" -> "…, yet unregulated, pollutants in cities …"
32. P35, L7: "The statistical …". The sentence is not correct.
33. P53, panels (d)-(f): Shouldn't the values to the left of 0 on the abscissa be negative?
34. P58: Spell out what version of TAPM (D4?) and CMAQ (CD4?) this refers to.
35. P61, Table 7: It takes a while to realise that TAPM also yields air quality results. In the main text TAPM is only mentioned in terms of dynamical downscaling of meteorology.
36. P61, Table 7: Why doesn't this table list the stations in the same order as in Tabs. 8-10? Why is station 54BL missing in Table 7? Why is Table 7 presenting absolute bias while Tabs 8-10 present relative bias?
37. P61-P62, Tables 8-10: Spell out in the table legend that the presented model performance statistics is for EPISODE-CityChem.

---

## Referee Comment (RC2) · Anonymous Referee #2 · 3 May 2019

EPISODE is a dispersion model for application in urban settings, including chemical reactions. The manuscript describes the architecture of the EPISODE model very carefully and in detail. Tests on modules (e.g., photochemistry) are performed, and described to very successful in terms of producing reasonable results in agreement with previous model and literature data, and referenced to sample data. The model is applied to the domain Hamburg. It is quite impressive how various sources of data sets are utilized in order to achieve the best possible model output. Eventually, the model fulfills performance objectives for the most important air pollutants and can thus

be used for regulatory applications. The authors also propose how the model can be further improved in the future. It is a fine contribution that should be published. There are only a few minor issues this reviewer want to raise:

page 1, line 27: replace "nitrogen oxide" by "nitric oxide"

page 7, R8a is not balanced. Do you you want to say: HCHO + 2O2 + hv => CO + 2HO2 ?

page 7, R10 is not balanced: Do you want to say: OH + CO + O2 => HO2 + CO2 ?

page 18, line 21: What is meant with "urban albedo and conductivity"? Do you mean heat conductivity of surface material?

---

## Author Response (AR1)

**Changes to manuscript ms-nr gmd-2018-325**

**The Eulerian urban dispersion model EPISODE. Part II: Extensions to the source dispersion and photochemistry for EPISODE-CityChem and its application to the city of Hamburg**

Matthias Karl (1), Sam-Erik Walter (2), Sverre Solberg (2), Martin O. P. Ramacher (1)

[1] Chemistry Transport Modelling, Helmholtz-Zentrum Geesthacht, 21502 Geesthacht, Germany.

[2] Norwegian Institute for Air Research (NILU), Kjeller, Norway.

**Dear Prof Dr Simone Marras,**

We highly appreciate the reviews of our manuscript ms-nr gmd-2018-325 that we received from two anonymous referees. We have replied to their comments in the Open Discussion. We have addressed all specific comments in the revised manuscript as will be described below. We carefully considered the concerns expressed by the referees in our revision of the manuscript. In accordance with the reviewer comments, we have modified the structure of the manuscript for a more consistent presentation of the new implementations of the CityChem extension in the EPISODE model.

Below follows: (1) the point-by-point replies to the two reviewers, (2) a list of relevant changes in the manuscript, and (3) the revised manuscript with changes highlighted.

Our responses to reviewers have been written in blue font.

Figure, table, section, and page numbers in the replies below refer to the original manuscript. The revised manuscript with changes highlighted has been sent along with this response.

**Referee #1**

1. The submitted work describes the EPISODE model with the CityChem extension. When I read the manuscript was part 1 of the article series not available, which makes it somewhat difficult to assess the need and value of the the CityChem extension. The reader of this particular manuscript is also left with an incomplete description of the EPISODE-CityChem system.

Part 2 intends to provide a description of the CityChem extension to the EPISODE model. For application in urban air quality modelling, the EPISODE-CityChem system was developed, which is comprised of the EPISODE model (manuscript part 1) including the CityChem extension and a set of pre-processing utilities and visualization tools. Part 1 provides a description of the physical processes of the Eulerian grid model and a basic description the embedded sub-grid models. The photochemistry on the Eulerian grid is however not included in part 1, since it belongs to the CityChem extension. The description of the EPISODE-CityChem model system will be improved based on the comments of this Reviewer.

2. EPISODE provides a promising set up with sub-grid scale Gausian models embedded into a Eulerian model with relevant photochemistry. As understood from the present manuscript does the CityChem extension offer three improvements compared to the original EPISODE model:

   1. Modification of the sub-grid photochemistry scheme

   2. Modification of the line source emission model

   3. Modifications to the plume rise calculations

   The three improvements are explained and visualised by presenting results from idealised sensitivity studies. In my view do the authors not quite get to the point when the respective improvement is presented and visualized.

The list of the three points given by the Reviewer is a copy of the three items listed in the Introduction (page 3). The mentioned three implementations are in connection with the sub-grid Gaussian models, and are part of the CityChem extension. However, the development of the photochemistry on the Eulerian grid ("grid photochemistry") has to be regarded as additional part of the CityChem development, and is the fourth implementation belonging to CityChem. The manuscript has been revised to arrange the order of the presentation of the improvements consistently: 1. Grid photochemistry, 2. Sub-grid chemistry, 3. Street canyon, 4. Point sources. The Introduction and subsequent sections have been revised to make this more visible to the reader. The order of the list given in the Introduction is followed in section 2 where the components are described and in section 3 where testing of each component is performed and visualized.

In addition, the revision of the study includes a full-year evaluation of the original EPISODE model, allowing for a comparison of the total effect of the CityChem extension on urban air pollutant concentrations in Hamburg.

3. Section 2.1 discusses a number of chemical schemes. From my reading of section 2.1 can I not tell what photochemical module(s) is (are) actually used in EPISODE-CityChem. Section 3.2 compares the performance of the updated photochemical scheme in the EPISODE-CityChem system with the standard PSS assumption. The conclusion (not shown in any graph!) is that the new scheme (EP10-Plume) behaves very similar to the original PSS-scheme. This is an interesting and valuable finding, but does not provide support for introducing an alternative photochemical scheme in the original EPISODE model.

First, the photochemical modules that are actually used in the EPISODE-CityChem simulation are shown in Table 5 (belonging to section 4.1.1). Table 5 displays which photochemistry scheme is used for (a) the photochemistry on the Eulerian grid and (b) the sub-grid modules. Table 5 has been moved to the beginning of section 2 in the revised manuscript and is now Table 1. Second, section 3.2 shows the test of EP10-Plume, the new photochemistry scheme for the sub-grid modules. Again, note that EP10-Plume should not be considered as the "alternative photochemical scheme", since it is only used in the sub-grid component. Figure 4 shows a comparison of EP10-Plume with PSS in the sub-grid model for line sources. The differences are very small and therefore the dashed lines in this plot overlap with the lines from PSS. Figure 4 has been revised and the overlap of lines is now mentioned in the figure caption. We note that the standard EPISODE model does not have a numerical photochemical solver and just uses the PSS assumption on the Eulerian grid and in the sub-grid components. A new subsection 4.3.2 ("Modelling of ozone formation") has been added in section 4 to demonstrate the effect of using advanced photochemistry on the Eulerian grid compared to the photo-stationary state approximation based on an evaluation of the summer season.

4. -I found section 2.2.1 -describing the improvements of the line source emission model particularly difficult to follow.

The text of section 2.2.1 has been revised to explain better the improvements to the line source sub-grid component.

5. -Section 2.2.2 details the calculation of wind speed at the actual plume height, and is a good reference for documenting how winds are calculated in the EPISODE-CityChem model although it is basically a standard similarity approach. As part 1 of the article series is not yet available it is impossible for the reader to know whether the "WMPP" is the sole improvement to the original EPISODE model or if the whole concept of plume rise calculations (Briggs, 1969; 1971; 1975) are also part of the development.

The plume rise formulation according to the Briggs equations is part of the standard EPISODE model and is described in part 1. WMPP leads to a modification of the plume rise from point sources. WMPP is part of the CityChem extension. The referred paragraph in section 2.2.2 (page 10, lines 2-6) has been removed from the manuscript.

6. The set up of the Hamburg domain is illustrated with a figure of the domains (Fig. 6) along with the text in section 4.1. It is still very difficult to understand the whole modelling chain of EPISODE-CityChem (and the other models participating in the comparison). For boundary concentrations of air pollutants there appears to be 2-3 CMAQ nests inside the global FMI APTA chemical transport model. Similarly, is meteorology dynamically downscaled from ERA5 in, at least, five nested TAPM domains. An impressive undertaking, indeed, but rather bulky and inconvenient for most applications.

The Reviewer has correctly interpreted the text in section 4.1. TAPM is used within this setup with two purposes: first to provide the downscaling of meteorology for the urban area and second to perform an alternative calculation of the air quality on urban scale. CMAQ is nested in two steps and the CMAQ simulation results have been used in other context (Karl et al., 2019). A deep nesting of both CMAQ and TAPM, as it was applied in the simulations for this study, is not necessarily required for other research applications of EPISODE-CityChem. The text in section 4.1 has been revised to better describe the global-to-local chemistry transport modelling chain and the downscaling of meteorological data.

Karl, M., Bieser, J., Geyer, B., Matthias, V., Jalkanen, J.-P., Johansson, L., and Fridell, E.: Impact of a nitrogen emission control area (NECA) on the future air quality and nitrogen deposition to seawater in the Baltic Sea region, Atmos. Chem. Phys., 19, 1721-1752, https://doi.org/10.5194/acp-19-1721-2019, 2019.

7. The location of the Hamburg study domain in the Southwest corner of the 4km x 4 km CD04-CMAQ domain does not make sense. The CD04 values at this point will hardly be any different than in the parent CD16 model, as the boundary is only 20-30 km away (less than an hour with a typical wind speed). I do not agree with the authors that: "This is considered to be sufficient to avoid that concentrations in the study domain are affected by domain border effects" (page 18-19, lines 34 and 1, respectively).

The Reviewer is right that results from CD04 in this part of the domain are not significantly different from CD16. We applied a rigid procedure for the nesting, having the CD04 domain as additional nesting level in order to maintain the nesting ratio of four when scaling down from Northern Europe to the urban area of Hamburg. In a less rigid approach, it would have been possible to use BCONs from CD16. The sentence "This is considered to be sufficient to avoid that concentrations in the study domain are affected by domain border effects." has been deleted. Nevertheless, no effects of domain borders appear in the data of CD04, since the concentrations are virtually identical (within 5 %) with the CD16 concentrations for the same geographic extent.

8. Many of the results presented are based on 1-2 week simulations (including, for example, section 4.1.2 and 4.1.3 and the final ~30 rows of section 4.2). I would be reluctant to draw any major conclusions from such limited data series.

We agree with the Reviewer that 1-2 week simulation experiments were not conclusive. The sections 4.1.2, section 4.1.3 and the last 30 rows of section 4.2 have been removed from the

revised manuscript, which also helped to shorten the manuscript. The connected figures and tables have been removed: Figure 7, Figure 11 and Table 7. In the Supplementary Information (SI), the following figures and tables have been deleted: Figure S3, Figure S4, Figure S7, Figure S8, and Table S5.

An exception is the "first experiment" described in section 4.1.2 involving constant boundary conditions (page 20, line 27 to page 21, line 5 in the original manuscript), which demonstrates the propagation of concentrations at the boundaries into the study domain. We decided to move this test to the new Appendix D ("Treatment of boundary concentration data") and to move the associated Figure S2 from the SI as new Figure D1 into this appendix.

9. The description of the "second experiment" involving the TAPM air quality model (P21, L6-17) is difficult to follow. Why only evaluate 14 days? I also lack a brief discussion on why the model results different. Is it due to different emissions or - for CMAQ, the seemingly best model- different meteorology? For the full 12-month evaluation (Figs 8,9,11 and Tables 8-10) is only EPISODE-CityChem compared against observations. Why is not the three-model comparison (Table 7) extended to the full 12-month period?

The "second experiment" on boundary conditions described in Section 4.1.2 has been removed from the revised manuscript, following the previous point of this Reviewer, because the short-term (14 days) evaluation is not conclusive. However, we have carried out a full-year run with the TAPM air quality model (which is a coupled meteorology-chemistry model) during the revision of this work. The TAPM run has been performed with the same horizontal resolution (1 km) as the EPISODE-CityChem run, identical emissions, but 2-D boundary concentrations instead of 3-D boundary conditions from CMAQ. Results from the new full-year run with TAPM have been included in the performance evaluation (Tables 8-10 and Figure 8+9). In addition, a full-year run with the standard EPISODE model (without the CityChem extension) has been used as a third model in the comparison and evaluation.

CMAQ is not further included in the evaluation because CMAQ cannot give realistic concentrations at the traffic sites and the industrial sites. CMAQ is a regional CTM system which does not handle local scale dispersion, i.e. a traffic site and a background site located within the same 4 x 4 km2 grid cell would have the same concentration values. More explanations regarding the comparison with CMAQ are given in the next point.

10. In this section 4.1.2 the high-resolution EPISODE-CityChem is compared with two coarser-resolution models for a two-week simulation over the city of Hamburg. As far as I can judge from the presentation is CMAQ superior to EPISODE-CityChem, which is interesting, since CMAQ is used as boundaries for EPISODE-CityChem. The test is not a good promotion for EPISODE-CityChem.

The two-week comparison between EPISODE-CityChem and CMAQ included only monitoring stations from the urban background. If the traffic stations and industrial stations were included, it would be obvious that CMAQ fails to reproduce concentrations at urban stations that are impacted by the local pollution. A realistic representation of local emissions is complicated by their high the spatial and temporal variability in the urban area. EPISODE-CityChem uses the

local scale emissions to compute the pollutant concentrations in the urban background areas, which are in turn affected by the highly resolved emissions. Therefore, the urban scale model is much more sensitive to an incorrect representation of the local emissions than the regional scale model with coarser resolution. As mentioned in the reply to the two previous points of the Reviewer (specific comments no. 8 and 9), we have removed the test and section 4.1.2 from the manuscript.

Triggered by the comment about the inferior performance at urban background stations, we have carefully inspected the code of the pre-processing utility BCONCC that is used to extract the 3D concentration field at the lateral and vertical boundaries from the CMAQ simulation (section 2.3.1 of the original manuscript). This led to the detection of a bug in the routine that distributes the CMAQ concentrations into the vertical layers (vertical interpolation) of EPISODE-CityChem. The 3D boundary conditions have been re-calculated with the revised BCONCC tool and new runs with EPISODE-CityChem using the re-calculated boundary conditions improved the performance for ozone and particulate matter concentrations. Figure C1 shows the comparison of the correlation coefficient for temporal variation at monitoring stations in Hamburg for a full-year evaluation using the revised boundary conditions. The new results for EPISODE-CityChem are documented in the evaluation section of the revised manuscript.

[Figure]

Figure C1: Comparison of the correlation coefficient for temporal variation (Hamburg, 2012) from EPISODE-CityChem with revised 3-D boundary conditions from CMAQ (red dots), standard EPISODE with revised 3-D boundary conditions from CMAQ (blue dots) and TAPM using 2-D boundary conditions from CMAQ (orange dots) for (a) O3, daily maximum of the 8-hour mean concentration, and (b) PM10, daily mean concentration.

11. The model performance evaluation of EPISODE-CityChem using FAIRMODE DELTA Tool is brief, but difficult to follow (Figure 9 and lines 1-19 on page 26). It could perhaps be dropped (or moved to the Supplement as a stand-alone entity), in favour of a more focussed presentation.

We prefer to keep the evaluation of the model quality objectives based on the FAIRMODE DELTA Tool because this has become the standard evaluation tool for the urban air quality community. We have revised the text belonging to Figure 9 of the original manuscript to better explain the meaning of the target diagrams obtained from the DELTA Tool.

12. The manuscript is rather long but at the same time superficial. I would like to recommend the authors to focus and streamline the presentation.

The manuscript has been shortened by removing the short-term experiments for Hamburg (sections 4.1.2 and 4.1.3). The structure of the manuscript has been changed for a more consistent presentation of the new implementations of the CityChem extension. The performance of EPISODE-CityChem in the full-year simulation for Hamburg is now compared to the performance of EPISODE and of TAPM. The added value of comprehensive treatment of grid photochemistry compared to PSS is illustrated in the revised manuscript in an additional result section 4.3.2 "Modelling of ozone formation".

13. There are several annoying features that distracts the reader from truly appreciating the presentation. For example:
a. As already pointed out is part 1 of the article series, not yet available. Still, the text frequently refers to Hamer et al. for explanations of background and details omitted in the presentation.

We keep the references to part 1 (Hamer et al., 2019) since the part 1 manuscript will be submitted soon.

Hamer, P. D., Slørdal, L. H., Walker, S. E., Sousa-Santos, G., Karl, M.: The urban dispersion model EPISODE: Part I: A Eulerian and subgrid-scale air quality model and its application in Nordic winter conditions, Geosci. Model Dev., in preparation, 2019.

13. b. The manuscript is long and the average reader lose focus after a while. Consider shorten and remove some parts:
-TWOSTEP is unnecessary mentioned and explained several times.
-There is no need, in my opinion to describe, in great detail, the interpolation and file format conversion of the boundary data to the EPISODE model (section 2.3.1).
-What is the rationale for testing the EMEP45, EmChem03-mod and EmChem09-mod in this presentation. Doesn't that belong to the standard EPISODE model (part 1 of the article series?).
-The section of comparing the new and standard K(z) profile (4.1.3) is not suited for this manuscript as the new K(z) method is described in part 1 of the article series. It also distracts the reader from the CityChem extensions -which should be the focus of this manuscript.

The mention of TWOSTEP is not unnecessary because it is a key element of the photochemistry functionality in the CityChem extension. Duplicate definition of TWOSTEP on page 4, line 24-25 has been removed.

The details on the format conversion of boundary condition data from CMAQ in section 2.3.1 and parts of the boundary data description on page 19 have been moved to a new Appendix D. The adaption of CMAQ concentrations at the boundaries is a feature of EPISODE-CityChem that had not been possible in the standard EPISODE.

As stated above in reply to the second comment (specific comment no. 2) of this Reviewer, the three photochemical schemes EMEP45, EmChem03-mod and EmChem09-mod belong to the CityChem extension and are therefore included in this part. We admit that the current presentation was confusing and that the Introduction did not make a clear point about grid photochemistry being a component of the CityChem extension. Although EMEP45 has been outlined in an old technical report (Walker et al., 2003) of NILU, this is the first time that EMEP45 and the newer photochemical schemes are presented in the peer-reviewed literature. Therefore, EMEP45 and the two newer chemistry schemes should be considered as important components of this part 2 paper and of EPISODE-CityChem. The Introduction and section 2 (page 4 of the original manuscript) have been revised to make this clear.

We agree with the Reviewer that the section 4.1.3 of the original manuscript which presents a comparison of the vertical profile with the new urban K(z) and the standard K(z) in EPISODE is better suited for part 1 where the description of the modification of the vertical eddy diffusivity is given. The new urban K(z) method does not belong to the implementations of CityChem extension. Section 4.1.3 and Figure 7 have been removed.

Walker, S.-E., Solberg, S., and Denby, B.: Development and implementation of a simplified EMEP photochemistry scheme for urban areas in EPISODE, Norwegian Institute for Air Research, NILU TR 13/2003, Kjeller, Norway, 2003.

13. c. I don't think it is good practice to discuss results in the main text that are only presented in the Supplement. This happens throughout the manuscript.

We have considered this remark of the Reviewer during the revisions of the manuscript. However, there is a trade-off between the shortening of the manuscript and the inclusion of more supplementary information within the manuscript. Figure S6 has been included in the revised manuscript. Figure S2 has been included in the new Appendix D. Several SI figures have been removed because they belonged to short-term evaluations (Figures S3, S4, S7 and S8 and Table S5).

13. d. The order of how items are presented and discussed is stochastic and confusing for the reader. Check, for example, section 4.2 where the different species are discussed in a seemingly random order, not even following the order in which they are presented in the figures.

Due its unproportioned length, the original section 4.2 has been divided in two parts: the first part (new section 4.2) deals with the evaluation of EPISODE-CityChem (now including standard EPISODE and TAPM) and the second part (new section 4.3 "Atmospheric chemistry in the urban area") deals with aspects of the urban atmospheric chemistry. Section 4.2 has been further divided into several subsections with clear topics. The new subsections are:

The description of the full-year evaluation of EPISODE-CityChem in new subsections 4.2.3 and 4.2.4 will be given in the same order of pollutants, i.e. first NO (only spatial variation) and $NO_2$, then $O_3$, $SO_2$ (only spatial variation), finally $PM_{2.5}$ (only spatial variation) and $PM_{10}$. Therefore, the order of tables 8-10 has been changed: Table 8: $NO_2$, Table 9: $O_3$, Table 10: $PM_{10}$.

The text in section 4.2 on page 26 and 27 gives a description of the atmospheric chemistry of Hamburg based on model results. The text is now in a separate section 4.3.1 "Mapping of annual mean concentrations" and the order of describing chemical compounds is the same as in Figure 10 of the original manuscript. Further, a new subsection 4.3.2 "Modelling ozone formation" has been added in the revised manuscript to compare the effect of advanced photochemistry on the Eulerian grid to the PSS (see reply to specific comment no. 12).

13. e. An overwhelming number of acronyms are introduced and used throughout the manuscript. Acronyms admittedly decrease the length of the presentation but also decrease legibility when these are first introduced and when the reader needs to go back and search for the explanation of a particular term. Would it be possible to put all explanations in a common table, for easy reference?
Some acronyms are not described the first time they are introduced. For example w_sc and L_max on page 8; TAPM on page 9;CMAQ on page 11; ….

We have added a table of the used acronyms in the new Appendix A of the revised manuscript. The definition of variables W_sc and L_max has been added. TAPM and CMAQ are now defined in the beginning of section 2 (page 4 of the original manuscript).

**Minor issues / typos**

1. Page 1, 2nd sentence: "… lower latitudes …". Why open with this in the Abstract? Hamburg is hardly lower latitudes. For most readers Hamburg clearly classifies as belonging to "…northern European cities."

The second sentence in the abstract has been changed:

"The development of the CityChem extension was driven by the need to apply the model in largely populated urban areas with highly complex pollution sources of particulate matter and various gaseous pollutants". In the Introduction (page 2, line 33-34) the statement on lower latitudes has been deleted.

2. P2, L20: "…150 km2 in size…". Unrealistically small domain if you use a 1 x 1 km2 resolution.

Thank you for spotting this mistake. The largest urban area domain extent computed with EPISODE-CityChem until now is 50 km x 50 km. We have replaced the number by 2500 km$^2$.

3. P3, Bullets: Change place/numbering of item 1 and 3. In the following sections you discuss modifications to photochemistry first (Section 2.1), then the street canyon model (Section 2.2.1) and finally the extension that provides realistic winds at actual plume height (section 2.2.2). Please also introduce the three modifications in the same order in the abstract (it is currently photochemistry, plume rise, line source).

There are four major items of the CityChem extension. We change the item list here, in the abstract, in the Introduction and in the beginning of section 2 in order to have a consistent order: 1. Grid photochemistry, 2. Sub-grid chemistry, 3. Street canyon, 4. Point sources.

4. P4, L16: "… fine particulate matter with (PM2.5;…". The sentence is not correct.

Changed.

5. P4, L18-19: "… NO2, the major pollutant in many cities of northern Europe". This statement is opposite to what is said in the introduction (P1, L21-23), where PM and ozone is pointed out as the main air pollution issues in Europe!

The statement in the Introduction (page 1, line 21-23) has been revised to consider NO$_2$ as important pollutant in European cities. The following sentence has been added on page 2, line 6:

"Traffic is a major source of nitrogen oxides (NO$_X$ = NO$_2$ + NO) and highly contributes to the population exposure to ambient NO$_2$ concentrations in urban areas, because these emissions occur close to the ground and are distributed across densely populated areas. Urban emissions of ozone precursors are transported by local/regional air mass flows towards suburban and rural areas, which can be impacted by O$_3$ pollution episodes (Querol et al., 2016)."

Querol, X., Alastuey, A., Reche, C., Orio, A., Pallares, M., Reina, F., Dieguez, J. J., Mantilla, E., Escudero, M., Alonso, L., Gangoiti, G., and Millan, M.: On the origin of the highest ozone episodes in Spain, Sci. Total Environment, 572, 379-389, https://doi.org/10.1016/j.scitotenv.2016.07.193, 2016.

6. P6, L21-22: "… NOx are often below 1 ppbv." This is a too strong sentence. NOx is rarely below 1 ppbv in "rural and sub-urban areas" of northern Europe.

We have deleted the half sentence.

7. P8, Eq 3: sigma_w0 should likely be squared.

Changed.

8. P9, L13-14: "…the average turbulence of the hypotenuse of the trapezium (slant edge towards the opposite street side)." Can this be explained so the un-initiated understands it?.

The cross-section of the recirculation zone is modelled as a trapezium. In Equation (4), $\sigma_{hyp}$ is the average turbulence at the hypotenuse of the trapezium. The following explanation has been added on page 9, line 11:

"The cross-section of the recirculation zone is modelled as a trapezium with the upper length $L_{top}$ and baseline length $L_{base}$. $L_{top}$ is half of the baseline length, where $L_{base}$, is defined as min($L_{rec}$, $L_{max}$). The length of the hypotenuse of the trapezium is calculated as $L_{hyp} = \sqrt{(L_{base}/2)^2 + H_{sc}^2}$ assuming the leeward side edge of the recirculation zone to be the vertical building wall, with the length of the building height. It is further assumed that the slant edge of the recirculation zone towards the opposite street side is not intercepted by buildings."

9.  P9, L15: (L_base/2) should likely be squared.

Changed.

10. P10, L1: Consider renaming the header of this section as it mainly goes through the method of deducing wind speed at height z. The "plume rise model" is only described by referring to Briggs.

The section 2.2.2 has been renamed to "Implementation of the WMPP for point sources".

11. P10, L20: I do not understand the +0.01 term. Can it be explained?

The +0.01 term is due to a change from potential temperature to actual temperature. Theta is potential temperature and T is actual temperature.

12. P12, Eq10: "C_point,s" should likely read "C_point,p".

Changed.

13. P14, L12: "… the ratio 10:1; …". Why not indicate this line in Fig. 2a?

A line with slope 10:1 has been added in Fig. 2a.

14. P15, L22: "Despite a slightly …". This sentence is not complete.

The incomplete sentence has been deleted.

15. P17, L7: What is the difference between "actual plume height" and "final plume height"?

The "actual plume height" refers to the intermediate plume heights that are reached along the plume trajectory until the equilibrium height of the final plume rise is reached. We have replaced "actual plume height" by "plume heights along the plume trajectory".

16. P17, L25: "…roughly proportional to …" -> "…roughly inversely proportional to …" (?).

Thank you for spotting this. Changed.

17. P17, L26: "…40.5 m (neutral) to 32.4 m …". Can you really defend three significant digits?

The effective emissions heights have been changed to two significant digits, here and in Table 3.

18. P19, L18: "…factor of 2.8 …". Why not "…factor of 8 …"?

In this example, a finer resolution leads to a doubling of the number of model steps within one hour. The fact that the computation time increases by a factor of 2.8 and not 8 shows that the computational demand mainly depends on the numerical model time step, while the number of grid cells has a smaller influence on the computation time. The numerical model time step of the EPISODE model is computed dynamically based on the critical time steps associated with the solution of the horizontal advection, vertical advection and diffusion processes.

19. P20, L22: "But considerations …". Strange start of sentence.

The sentence has been rephrased.

20. P23-28: Section 4.2 is much longer than any of the other sections. Can it be divided into subsections for increased legibility?.

The section 4.2 ("Presentation and evaluation of model results") has been divided into several subsection and arranged in accordance with general comments of this Reviewer, see reply to specific comment no. 13d.

21. P23, L3: "…when on the following days." -> "…than on the following days." (?).

Section 4.1.3 has been removed.

22. P24, L1: "26.56". Can you really defend four significant digits?

The value is now given with two significant digits.

23. P24, L11-12. Sentence including "…, modelled and show …" is not correct.

The sentence has been rewritten as follows: "At Hamburg weather mast, modelled versus observed wind directions show good agreement in 10 m and 50 m height (IOA ≥ 0.89) with a Bias in mean wind direction of 16.9º and 6.2º at 10 m and 50 m, respectively."

24. P25, L7: "bias is within +/- 25 ug/m3". From Fig. 8b it is more like +/- 10 ug/m3.

This is correct. The statement on bias of annual mean NO2 was changed to +/- 10 µg/m3 (for most stations).

25. P25, L12: "…summer (JJA) mean…". Summer mean values are not shown in the manuscript.

The sentence has been deleted. The performance of EPISODE-CityChem for summertime ozone is now discussed in the new section 4.3.2 ("Modelling ozone formation") at the end of section 4.

26. P25, L19: "…diurnal cycles…". Not shown in the main manuscript. Avoid discussing details only shown in supplement..

The text on page 25, line 14-26 has been moved to the new section 4.3.2 ("Modelling ozone formation") at the end of section 4.2. Following the specific comment no. 13c of the Reviewer, Fig. S6 has been included in the manuscript in section 4.3.2 as new Figure 10.

27. P25, L27: "… underestimate observed PM2.5 and PM10…". From Fig. 8e,f, these are excellently reproduced!

We assume that this comment refers to page 25, line 27. The new runs with EPISODE-CityChem with revised boundary concentrations (see specific comment no. 10) show that annual mean $PM_{2.5}$ and $PM_{10}$ at most stations are well reproduced and the previous noted underestimation of observed winter peaks of $PM_{2.5}$ and $PM_{10}$ is not apparent in the new run. In the new run, annual mean $PM_{10}$ is slightly overestimated (by 22-24 %) at the traffic stations. Therefore, the sentence on page 25, line 27 has been deleted in the revised manuscript.

28. P27, L5: "… in some distance…" -> "… at some distance…" (?).

Changed.

29. P29, L5: "…EMEP unified model". I bvelieve "unified" (as of combining photochemistry and acid deposition in the same model) has been dropped since several years..

The word "unified" has been deleted.

30. P29, L9,L14: Is there a reason why you spell out "photochemical steady-state" on line 14, but abbreviate to PSS on line 9?.

This has been exchanged (first spelt out, second abbreviated).

31. P30, L23: "…, yet unregulated pollutants, in cities …" -> "…, yet unregulated, pollutants in cities …"..

Changed.

32. P35, L7: "The statistical …". The sentence is not correct.

Changed.

33. P53, panels (d)-(f): Shouldn't the values to the left of 0 on the abscissa be negative?.

The shown Target diagram is correctly reproduced (refer to Fig. 2 in Monteiro et al., 2018). The explanation for the values on the left part of the abscissa is given in Thunis et al. (2012a): Because the centered RMSE is always positive, only the right part of the diagram would be

needed in the Target plot, while the negative X axis section can be used to provide additional information. Thunis et al. chose to give the information whether the centered RMSE related error is dominated by standard deviation or by correlation. In the right quadrant of the Target diagram, the error related to standard deviation dominates the model performance and in the left quadrant the error related to correlation dominates the model performance. The following information has been added on page 26, line 16:

"In the right quadrant of the Target diagram (Fig. 9d–f), the error related to standard deviation dominates the model performance and in the left quadrant the error related to correlation dominates the model performance."

Monteiro, A., Durka, P., Flandorfer, C., Georgieva, E., Guerreiro, C., Kushta, J., Malherbe, L., Maiheu, B., Miranda, A. I., Santos, G., Stocker, J., Trimpeneers, E., Tognet, F., Stortini, M., Wesseling, J., Janssen, S., and Thunis, P.: Strengths and weaknesses of the FAIRMODE benchmarking methodology for the evaluation of air quality models, Air Quality, Atmosphere & Health, published online, https://doi.org/10.1007/s11869-018-0554-8, 2018.

Thunis, P. Pederzoli, A., Pernigotti, D.: Performance criteria to evaluate air quality modeling applications, Atmos. Environ., 79, 476-482, 2012a.

34. P58: Spell out what version of TAPM (D4?) and CMAQ (CD4?) this refers to.

The name of the nests has been added in the table caption and column header of Table 4 in the revised manuscript.

35. P61, Table 7: It takes a while to realise that TAPM also yields air quality results. In the main text TAPM is only mentioned in terms of dynamical downscaling of meteorology.

Table 7 has been removed together with section 4.1.2; see reply to specific comment no. 8. TAPM will be introduced as a reference model for AQ calculation in the revised manuscript. The evaluation results from the TAPM air quality model from a full-year calculation for Hamburg are have been included in Tables 8-10.

36. P61, Table 7: Why doesn't this table list the stations in the same order as in Tabs. 8-10? Why is station 54BL missing in Table 7? Why is Table 7 presenting absolute bias while Tabs 8-10 present relative bias?

Table 7 has been removed; see reply to specific comment no. 8.

37. P61-P62, Tables 8-10: Spell out in the table legend that the presented model performance statistics is for EPISODE-CityChem.

Reply: The model name has been added in the table captions of Tables 8-10. Data from standard EPISODE and TAPM has been included in these tables in the revised manuscript.

**Referee #2**

1. EPISODE is a dispersion model for application in urban settings, including chemical reactions. The manuscript describes the architecture of the EPISODE model very carefully and in detail. Tests on modules (e.g., photochemistry) are performed, and described to very successful in terms of producing reasonable results in agreement with previous model and literature data, and referenced to sample data. The model is applied to the domain Hamburg. It is quite impressive how various sources of data sets are utilized in order to achieve the best possible model output. Eventually, the model fulfills performance objectives for the most important air pollutants and can thus be used for regulatory applications. The authors also propose how the model can be further improved in the future. It is a fine contribution that should be published.

We thank the Reviewer for their assessment of the scope and methodology of the manuscript and the applicability of the EPISODE-CityChem model.

2. There are only a few minor issues this reviewer want to raise:
   a) page 1, line 27: replace "nitrogen oxide" by "nitric oxide"

Thank you. We excuse the mistake in the chemical name of NO. Further, we changed on page 5, line 26, "hydrogen peroxy radical" to "hydroperoxyl radical" ($HO_2$).

2. b) page 7, R8a is not balanced. Do you want to say: HCHO + 2O2 + hv => CO + 2HO2 ?

R8a was unbalanced with respect to oxygen atoms. Oxygen is usually ignored on the reactant side when setting up chemical equations for the numerical ODE solver because the mixing ratio of oxygen in the atmosphere is not affected by this reaction to any significant extent. The photolysis of formaldehyde in the radical channel gives HCHO + hv => H + HCO. The hydrogen atom (H) quickly combines with $O_2$ to form $HO_2$ and the intermediate formyl radical (HCO) reacts very rapidly with $O_2$ to yield $HO_2$ and CO. We correct this equation as written by the reviewer to fulfil the balance for oxygen atoms.

2. c) page 7, R10 is not balanced. Do you want to say: OH + CO + O2 => HO2 + CO2 ?

R10 was unbalanced with respect to oxygen atoms. We correct this equation as written by the reviewer to fulfil the balance for oxygen atoms.

2. d) page 18, line 21: What is meant with "urban albedo and conductivity"? Do you mean heat conductivity of surface material?

Albedo and conductivity are material properties referring to urban surfaces such as concrete/asphalt/roofs. Urban albedo means the surface albedo in cities, i.e. the ability of urban surfaces to reflect solar radiation. Urban surface albedo values in TAPM are based on Oke (1988; table 2 therein). Urban conductivity here means thermal conductivity, which is a measure of the ability of a surface material to conduct or transmit heat. A surface material with

a high thermal conductivity will transfer heat at a higher rate than a material having a low thermal conductivity.

For clarity, we have replaced "urban albedo and conductivity" by "albedo of urban surfaces and thermal conductivity of urban surface materials, e.g. concrete/asphalt/roofs".

Oke T. R.: The urban energy balance, Progress in Physical Geography, 12, 471–508, https://doi.org/10.1177/030913338801200401, 1988.

**List of relevant changes in the ms**

**Relevant text changes:**

The structure of the manuscript has been changed for a more consistent presentation of the new implementations of the CityChem extension. The performance of EPISODE-CityChem in the full-year simulation for Hamburg is now compared to the performance of EPISODE and of TAPM. The added value of comprehensive treatment of grid photochemistry compared to photochemical steady-state (PSS) assumption is illustrated in the revised manuscript in an additional result section 4.3.2 "Modelling of ozone formation".

**1. Introduction**

The Introduction has been revised in accordance with the comment of Referee #1 about the unclear presentation of the improvements relating to the CityChem extension compared to the operational EPISODE model, which is described in Part 1 - still in preparation. The improvements included in the CityChem extension are: 1. Grid photochemistry, 2. Sub-grid chemistry, 3. Street canyon, 4. Point sources. This is now presented as an itemized list in the Introduction and the order of items is followed throughout section 2 where the components are described and in section 3 where testing of each component is performed and visualized.

It has been clarified in the Introduction, that Part 1 provides a description of the physical processes of the Eulerian grid model and a basic description of the embedded sub-grid models. The photochemistry on the Eulerian grid is however not included in Part 1, since it belongs to the CityChem extension.

Following a minor remark of Referee #1, a statement about $NO_2$ as important pollutant in European cities was included:

"Traffic is a major source of nitrogen oxides ($NO_X = NO_2 + NO$) and highly contributes to the population exposure to ambient $NO_2$ concentrations in urban areas, because these emissions occur close to the ground and are distributed across densely populated areas. Urban emissions of ozone precursors are transported by local/regional air mass flows towards suburban and rural areas, which can be impacted by $O_3$ pollution episodes (Querol et al., 2016)."

**2. Section 2 "Development and description of EPISODE-CityChem model extensions"**

We addressed the comment of Referee #1 about the incomplete description of the EPISODE-CityChem model by modification of Figure 1, giving an illustration of the EPISODE model with the components CityChem extension, and a better description of the configuration of EPISODE-CityChem for air quality simulations. We moved Table 5 of the original manuscript to the beginning of section 2 (now Table 1), displaying the photochemistry schemes that are used for (a) the photochemistry on the Eulerian grid and (b) the sub-grid modules.

Following the remark of Referee #2, we have checked the balance of all reaction equations given in the manuscript and corrected the balance of reaction (R.8a) and (R.10).

The text of Sect. 2.2.1 ("Implementation of a simplified street canyon model (SSCM) for line source dispersion") has been revised to explain better the improvements to the line source sub-grid component.

Sect. 2.2.2 was renamed to: "Implementation of the WMPP for point sources". The plume rise formulation according to the Briggs equations is part of the standard EPISODE model and is described in Part 1. The first paragraph in section 2.2.2 was deleted.

Sect. 2.3.1 ("Adapting 3-D boundary conditions from the CMAQ model") which describes the procedure of creating 3-D boundary conditions from a regional air quality simulation with the CMAQ model was shortened and the technical information is now given in the new Appendix D ("Treatment of boundary concentration data").

**3. Section 4.1 "Setup of model experiments for the application for AQ modelling in Hamburg"**

The description of the setup of the simulations with EPISODE-CityChem for air quality in Hamburg in Sect. 4.1 was revised to better explain the global-to-local chemistry transport modelling chain and the downscaling of meteorological data:

"EPISODE-CityChem was run as part of a one-way nested chemistry-transport model chain from global scale to the urban scale. The APTA (Asthma and Allergies in Changing Climate) global re-analysis (Sofiev *et al.*, 2018) of the Finnish Meteorological Institute (FMI) provided the chemical boundary conditions for the European domain. CMAQ v5.0.1 CTM (Byun *et al.*, 1999; Byun and Schere, 2006; Appel *et al.*, 2013) was run with a temporal resolution of one hour over the European domain and an intermediate nested domain over Northern Europe with 64-km and 16-km horizontal resolution, respectively."

CMAQ simulations were driven by the meteorological fields of the COSMO-CLM (COnsortium for SMall-scale MOdeling in CLimate Mode) model version 5.0 (Rockel *et al.,* 2008) for the year 2012 using the ERA-Interim re-analysis of the European Centre for Medium-Range Weather Forecasts (ECMWF) as forcing data (Geyer, 2014).

Within the Northern Europe domain, an inner domain over the Baltic Sea region with 4-km horizontal resolution was nested (Fig. 6a). The 4-km resolved CMAQ simulation of the Baltic Sea region provided the initial and hourly boundary conditions for the chemical concentrations of the Hamburg model domain. The hourly meteorological fields for the study domain Hamburg (30 x 30 km$^2$) were obtained from the inner domain in a nested simulation with TAPM (Hurley *et al.*, 2005) with a 1-km horizontal resolution (D4 in Fig. 6b)."

In accordance with Referee #1, we removed Sect. 4.1.2 ("Description of the experiment to test the CMAQ nesting versus TAPM") and Sect. 4.1.3 ("Experiment to test the new versus old vertical diffusion scheme") and the associated Figure 7 and Table 7. This substantially shortens the manuscript. The "first experiment" described in Sect. 4.1.2 involving constant boundary conditions was moved to the new Appendix D ("Treatment of boundary concentration data") and the associated Figure S2 from the SI was included as new Figure D1 in this appendix.

**4.    Section 4.2 "Presentation and evaluation of model results"**

As recommended by Referee #1, we have divided section 4.2 of the original manuscript into two parts: the first part (new section 4.2) deals with the evaluation of EPISODE-CityChem (now including standard EPISODE and TAPM) and the second part (new section 4.3 "Atmospheric chemistry in the urban area") deals with aspects of the urban atmospheric chemistry.

Section 4.2 has been further divided into several subsections with clear topics. The new subsections are:

> 4.2.1 Setup of the model evaluation and performance analysis
>
> 4.2.2 Evaluation of downscaled meteorological data
>
> 4.2.3 Evaluation of the temporal variation of pollutants
>
> 4.2.4 Evaluation of the spatial variation of pollutants
>
> 4.2.5 Model performance analysis for policy support applications

The description of the full-year evaluation of model results in new sections 4.2.3 and 4.2.4 is given in the same order of pollutants, i.e. first NO (only spatial variation) and $NO_2$, then $O_3$, $SO_2$ (only spatial variation), finally $PM_{2.5}$ (only spatial variation) and $PM_{10}$. Therefore, the order of tables 8-10 has been changed: Table 8: $NO_2$, Table 9: $O_3$, Table 10: $PM_{10}$.

We have carefully inspected the code of the pre-processing utility BCONCC that is used to extract the 3D concentration field at the lateral and vertical boundaries from the CMAQ simulation (section 2.3.1 of the original manuscript). This led to the detection of a bug in the routine that distributes the CMAQ concentrations into the vertical layers (vertical interpolation) of EPISODE-CityChem. The 3D boundary conditions have been re-calculated with the revised BCONCC tool and new runs with EPISODE-CityChem using the re-calculated boundary conditions improved the performance for ozone and particulate matter concentrations.

The performance evaluation of the model was extended by comparing to model results from a full-year run with the standard EPISODE model and the TAPM air quality model. Comparison to results from the standard EPISODE model are used to assess the total effect of the new implementations of the CityChem extension. In the standard EPISODE model, the PSS approximation is used at the receptor points and on the Eulerian grid; the street canyon model and the WMPP module were deactivated. TAPM model is used as a reference model for AQ modelling. CMAQ was not further included in the evaluation because CMAQ cannot give realistic concentrations at the traffic sites and the industrial sites.

The text in sections 4.2.3-4.2.5 now considers air quality model results from EPISODE and TAPM and briefly explains the differences between the models and the possible reasons for different performances of the models.

The evaluation results from the standard EPISODE model and from the TAPM air quality model from a full-year calculation for Hamburg are included in Tables 7-9 of the revised manuscript (i.e. Tables 8-10 of the original manuscript).

The evaluation of the model quality objectives based on the FAIRMODE DELTA Tool in the new section 4.2.5 ("Model performance analysis for policy support applications") has been repeated for the three models included in the full-year evaluation. We revised the text belonging to Figure 9 of the original manuscript to better explain the meaning of the target diagrams obtained from the DELTA Tool.

In order to shorten the manuscript, we deleted the last 30 rows of original section 4.2 and Figure 11 of the original manuscript.

**5.      New Section 4.3 "Atmospheric chemistry in the urban area"**

The text in section 4.2 on page 26 and 27 of the original manuscript gave a description of the atmospheric chemistry of Hamburg based on model results. In the revised manuscript, the text is now in a separate section 4.3.1 "Mapping of annual mean concentrations" and the order of describing chemical compounds is the same as in Figure 10 of the original manuscript. Further, a new section 4.3.2 "Modelling ozone formation" has been added in the revised manuscript to compare the effect of advanced photochemistry on the Eulerian grid to the PSS.

A new figure panel was included (Figure 10 of the revised manuscript) related to new section 4.3.2. The new Figure 10 shows the comparison of ozone formation in summer between the advanced photochemistry scheme EmChem09-mod and the photochemical state assumption for summer mean concentrations and diurnal profiles of $O_3$ and $NO_2$.

**8.      Conclusions**

The Conclusion was aligned with the order of the presentation of the improvements in the CityChem extension: 1. Grid photochemistry, 2. Sub-grid chemistry, 3. Street canyon, 4. Point sources.

The comparison of the performances of EPISODE-CityChem, EPISODE and TAPM for simulation of the air quality in Hamburg is summarized in the Conclusions as follows (page 32, line 6 of the original manuscript):

"The ability to reproduce the temporal variation of major regulated pollutants at AQ monitoring stations in Hamburg was compared to that of the standard EPISODE model and the TAPM AQ model using identical meteorological fields and emissions. EPISODE-CityChem performs better than EPISODE and TAPM for prediction of hourly $NO_2$ concentrations at the traffic stations, which is attributable to the street canyon model. EPISODE-CityChem was in better agreement with observed $O_3$ daily maximum of the 8-h running mean than the other two models. For daily mean $PM_{10}$ at urban background stations, EPISODE-CityChem and EPISODE gave better results than TAPM, largely due to the use of hourly 3-D boundary conditions from CMAQ."

The modelled ozone formation in summer with EPISODE-CityChem is summarized in the Conclusions as follows (page 32, line 10 of the original manuscript):

"The effect of using an advanced photochemical mechanism (EmChem09-mod) compared to the PSS assumption for modelling ozone concentrations and ozone production was investigated in summer (JJA) simulations for Hamburg. Photochemical ozone production was found to take place in the outflow of polluted air from the city, implying that advanced photochemistry is necessary for a more accurate prediction of $O_3$ in the urban background. However, the modelled daily maximum $O_3$ in summer afternoons was ca. 25 % lower than observed at an inner-city urban background station. In addition, the model predicted too high $NO_2$ concentration in the evenings at two urban background stations. Further investigation of the high modelled evening $NO_X$ in summer will require sensitivity analysis of the various source categories contributing to the $NO_X$ levels in the inner city, which remains as a task for future studies."

**Tables:**

Table 1.

The table has been renumbered as Table 2.

Table 2.

The table has been renumbered as Table 3.

Table 3.

The table has been renumbered as Table 4.

Table 4.

The table has been renumbered as Table 5.

Table 5.

The table has been moved to section 2 and is now Table 1.

Table 7.

The table has been removed.

Table 8.

The table now includes results of the statistical performance analysis from standard EPISODE and TAPM for the temporal variation of $O_3$ based on daily maximum of the 8-h running mean concentration.

Table 9.

The table now includes results of the statistical performance analysis from standard EPISODE and TAPM for the temporal variation of $PM_{10}$ based on daily mean concentration.

Table 10.

The table has been renumbered as Table 7. The table now includes results of the statistical performance analysis from standard EPISODE and TAPM for the temporal variation of $NO_2$ based on hourly concentration.

Table A1.

A new table giving a list of acronyms and abbreviations used in the manuscript text was added in new Appendix A.

**Figures:**

Figure 1.

The illustration of the components of the CityChem extension was improved.

Figure 2.

A line of VOC/$NO_X$ ratio 10:1 was added in the ozone isopleth diagram of Figure 2a.

Figure 4.

Figure 4b has been revised to improve the annotation of lines from EP10-Plume and PSS. The overlap of lines is now mentioned in the figure caption.

Figure 7.

The figure has been removed because the description of new vertical eddy diffusivity method, urban K(z) method, and the associated testing of the modification does not belong to the implementations of CityChem extension, as noted by Referee #1.

Figure 8.

The figure has been renumbered as Figure 7.

Figure 9.

The figure has been renumbered as Figure 8.

Figure 10.

The figure has been renumbered as Figure 9.

Figure 11.

The figure panel on concentration time profiles of a 6-day ozone episode has been replaced by a new figure panel showing the comparison of ozone formation in summer the advanced photochemistry scheme EmChem09-mod and the photochemical state assumption for summer mean concentrations and diurnal profiles of $O_3$ and $NO_2$. The figure has been renumbered as Figure 10.

Figure D1.

The new Figure D1 in the new Appendix D illustrates the effect of boundary conditions on the modelled concentrations of $O_3$ and $PM_{2.5}$ in the centre of the model domain of Hamburg.

[revised manuscript text omitted]
_{sc}\sigma_w} \cdot \int_{x_{start}}^{x_{end}} \frac{1}{x + \frac{u_{street}h_0}{\sigma_w}} dx. \tag{2}$$

Where $h_0$ is a constant that accounts for the height of the initial pollutant dispersion ($h_0 = 2$ m is used in SSCM), $\sigma_w$ is the vertical velocity fluctuation due to mechanical turbulence generated by wind and vehicle traffic in the street, and $u_{street}$ is the wind speed at street level, calculated assuming a logarithmic reduction of the wind speed at rooftop towards the bottom of the street. Note that the wind direction at street level in the recirculation zone is mirrored compared to the roof level wind direction. Outside the recirculation zone, the wind direction is the same as at roof level. The vertical velocity fluctuation is calculated as a function of the street level wind speed, and the traffic produced turbulence by the following relationship (Berkowicz et al., 1997):

$$\sigma_w = \sqrt{(\alpha_s u_{street})^2 + (\sigma_{w0})^2} \sqrt{(\alpha_s u_{street})^2 + (\sigma_{w0})^2}, \tag{3}$$

where $\alpha_s$ is a proportionality constant, empirically assigned a value of 0.1, and $\sigma_{w0}$ is the traffic-induced turbulence, in SSCM assigned a value of $0.25$ m s$^{-1}$, typical for traffic on working days between 8 a.m. and 7 p.m. in situations where traffic-induced turbulence dominates (Kastner-Klein et al., 2000; fig. 6 therein).

The integration path for Eq. (2) begins from $x_{start}$ which is defined as the distance from the receptor point where the plume has the same height as the receptor, which is zero in the case that $h_0$ is smaller or equal to the height of the receptor. The upper integration limit is $x_{end}$ defined by tabular values in Ottosen et al. (2015; table 3 therein). The integration is performed along a straight line path against the wind direction. The calculation of the maximum integration path, $L_{max}$, depends on the wind direction with respect to the street axis, $\theta_{street}$, i.e. the angle between the street and the street level wind direction (Ottosen et al., 2015).

The  *recirculation contribution* is computed using a simple box model  assuming equality of the inflow and outflow of the pollutant.

$$C_{screc,s} = \frac{Q_s}{W_{sc}} \cdot \frac{L_{base}}{\sigma_{wt}L_{top} + \sigma_{hyp}L_{hyp}},$$

where $\sigma_{wt}$ is the ventilation velocity of the canyon as given by Hertel and Berkowicz (1989) and $\sigma_{hyp}$ is the average turbulence of the hypotenuse of the trapezium (slant edge towards the opposite street side The cross-section of the recirculation zone is modelled as a trapezium with the upper length $L_{top}$ and baseline length $L_{base}$. $L_{top}$ is half of the baseline length, where $L_{base}$ is defined as $\min(L_{rec}, L_{max})$. The length of the hypotenuse of the trapezium is calculated as

[revised manuscript text omitted]

where $\kappa$ is Von Kármán's constant (0.41), $g$ is the acceleration of gravity (9.81 m s$^{-2}$); $\Delta u$ is the wind speed difference between heights $z_{u2}$ and $z_{u1}$, where $z_{u2}$ is e.g. 10 m, and $z_{u1} = z_{0m}$ where the wind speed is zero, so that $\Delta u = u_{10m} - 0 = u_{10m}$. In the definition of the temperature scale, $\Delta \theta$ is the difference in potential temperature between heights $z_{t2}$ and $z_{t1}$, which are e.g. 10 m and 2 m respectively, so that we have $\Delta \theta = T_{10m} - T_{2m} + 0.01$ , where the +0.01 term is for conversion from potential temperature to actual temperature. In the definition of the Obukhov length, $T_{\text{ref}}$ is a reference temperature, here taken to be the average of $T_{2m}$ and $T_{10m}$.

In Eq. (5), the similarity functions $\varphi_m$ and $\varphi_h$ are defined as follows (Högström, 1996):

[revised manuscript text omitted]

~~The background concentrations are adopted for the grid cells directly adjacent to the grid cells of the model domain (with $nx \times ny$ cells per model layer) and also for the vertical model layer that is on top of the highest model layer. Boundary conditions from CMAQ concentrations are created for the gas-phase chemical compounds: , , , , , , , , , and the individual . Boundary conditions for includes primary aerosol components: elemental carbon (), primary organic aerosol (), sea salt (), and mineral dust; secondary inorganic aerosol () components: sulphate (), ammonium (), nitrate (), and SOA (
[revised manuscript text omitted]

$$\mathrm{MQI}^2 = \frac{\mathrm{Bias}^2}{(\beta\, \mathrm{RMS}_U)^2} + \frac{(\mathrm{SD}_M - \mathrm{SD}_O)^2}{(\beta\, \mathrm{RMS}_U)^2} + \frac{2\, \mathrm{SD}_O - \mathrm{SD}_M (1 - \mathrm{Corr})}{(\beta\, \mathrm{RMS}_U)^2} \ . \tag{E11}$$

From Eq. (E11), the model performance criterion (MPC) for the error of bias, standard deviation and correlation can be derived. The bias MPC is derived from Eq. (E11) assuming $\mathrm{Corr} = 1$ and $\mathrm{SD}_M = \mathrm{SD}_O$, as follows:

[revised manuscript text omitted]